# Noise Augmented Fine Tuning for Mitigating Hallucinations in Large Language Models

## Abstract

Large language models (LLMs) often produce inaccurate or misleading content—*hallucinations*. To address this challenge, we introduce **Noise-Augmented Fine-Tuning (NoiseFiT)**, a novel framework that leverages *adaptive noise injection* based on the signal-to-noise ratio (SNR) to enhance model robustness. Our contribution is threefold. First, NoiseFiT selectively perturbs layers identified as either *high-SNR* (more robust) or *low-SNR* (potentially under-regularized) using a dynamically scaled Gaussian noise. Second, we further propose a *hybrid loss* that combines standard cross-entropy, soft cross-entropy, and consistency regularization to ensure stable and accurate outputs under noisy training conditions. Third, a theoretical analysis proposed shows that adaptive noise injection is both *unbiased* and *variance-preserving*, providing strong guarantees for convergence in expectation. Moreover, empirical results on multiple test and benchmark datasets, demonstrate that NoiseFiT significantly reduces hallucination rates, often *improving* or *matching* baseline performance in key tasks. These findings highlight the promise of noise-driven strategies for achieving robust, trustworthy language modeling without incurring prohibitive computational overhead. We have publicly released the fine-tuning logs, benchmark evaluation artifacts, and source code online at W&B, Hugging Face, and GitHub, respectively, to foster further research, accessibility and reproducibility.

## 1 Introduction

LLMs such as GPT-3 (Brown et al., 2020) and GPT-4 (OpenAI et al., 2024), built upon transformer architectures (Vaswani et al., 2017), have revolutionized the field of natural language processing by achieving state-of-the-art performance on a diverse range of tasks. Despite their impressive capabilities, these models are known to generate content that is often inaccurate or misleading—phenomena broadly referred to as *hallucinations* (Ji et al., 2023; Bang et al., 2023; Niu et al., 2024). The risk of such hallucinations not only arises in specialized domains such as healthcare (Moor et al., 2023) and finance (Wu et al., 2023), where reliability is paramount, but also extends to a variety of more general-purpose benchmarks and tasks such as question answering, underscoring the urgency of developing robust mitigation strategies. Consequently, ensuring the trustworthiness of LLM outputs is critical, making the reduction of hallucinations both a technical and practical imperative for real-world adoption.

Recent research has increasingly focused on noise injection as a means to enhance model robustness. Early work in image restoration demonstrated the efficacy of learning from noisy data (Lehtinen et al., 2018), and this idea has since been adapted for natural language processing. In the context of LLM fine-tuning, noise injection techniques have shown promising results. For instance, methods such as noise perturbation fine-tuning for robust quantization (WANG & Yang, 2025) and the use of noisy embeddings to improve instruction fine-tuning (Jain et al., 2024) illustrate that controlled noise can help models generalize better under diverse and challenging conditions. Additional studies have revealed hidden capabilities of noise injection in reducing overconfidence and mitigating model biases (Tice et al., 2024; Yadav & Singh, 2023), and even enhancing hallucination detection (Liu et al., 2025). Complementary evaluation frameworks, including large-scale benchmarks for hallucination evaluation (Li et al., 2023a) and rigorous instruction-following assessments (Zhou et al., 2023), further motivate the development of noise-based approaches in addressing LLM shortcomings.

In light of these advancements, we propose *Noise-Augmented Fine-Tuning (NoiseFiT)*, a novel framework that integrates adaptive noise injection into the fine-tuning process of LLMs. The key innovation of NoiseFiT lies in its use of adaptive Gaussian noise injection, which is guided by the SNR of internal representations, i.e., the ratio of meaningful information to background noise. By selectively perturbing transformer layers based on the behavior of internal hidden states (Sriramanan et al., 2024), our approach aims to directly enhance robustness against hallucinations while minimizing disruption to overall performance. This strategy is further supported by recent studies demonstrating the benefits of noise regularization and stochastic perturbations in fine-tuning settings (Wang et al., 2023a; Wu et al., 2020; Hua et al., 2023).

The central research questions motivating this work are:

- **RQ1:** *How does adaptive noise injection during fine-tuning help mitigate hallucinations in the outputs of LLMs?*
- **RQ2:** *What is the relationship between the intensity of noise injection and the trade-off between robustness and task performance across diverse applications?*
- **RQ3:** *How does the proposed NoiseFiT framework affect the computational efficiency and scalability of hallucination mitigation?*

The remainder of this paper is organized as follows. Section 2 reviews recent advances in the field and situates our contributions within this context. Section 3 details the NoiseFiT methodology, including the adaptive noise injection strategy. Section 4 reports the experimental setup and results, benchmarking NoiseFiT against state-of-the-art methods. Section 5 presents limitations and discusses the broader implications of our findings. Finally, Section 6 concludes this paper and outlines future research directions.

## 2 RELATED WORK

Existing strategies to reduce hallucinations include retrieval-augmented generation (RAG), which grounds outputs in external knowledge retrieved at different stages of inference (Lewis et al., 2020; Asai et al., 2024; Wang et al., 2025). Reinforcement learning from human feedback (RLHF) (Ouyang et al., 2022) where aligns models with human preferences by training a reward model from curated signals or annotations, then optimizing responses through reinforcement learning. Self-consistency (Wang et al., 2023b) is a decoding strategy built on chain-of-thought prompting (Wei et al., 2022), which improves reasoning reliability by sampling diverse reasoning paths and selecting the most consistent answer. Contrastive decoding (Li et al., 2023b), and adversarial training (Zhu et al., 2024). More recently, CDCR-SFT has been introduced as a supervised fine-tuning framework in which LLMs are trained to construct variable-level directed acyclic graph (DAG) and subsequently reason over them (Li et al., 2025).

While these methods have achieved some success, they often come with increasing computational overhead, adding latency during inference, increased sensitivity to hyperparameter tuning, and limited generalization across varying domains (Zhao et al., 2025; Zellers et al., 2019). These limitations underscore the need for alternative approaches that can robustly improve model performance without incurring prohibitive resource demands.

To address these questions, our methodology introduces a hybrid loss objective that combines standard cross-entropy with soft target regularization and consistency-based penalties computed over multiple noisy forward passes. This formulation not only ensures that the model retains its predictive capabilities under perturbation but also enforces consistency in its outputs—a key requirement for robust performance in safety-critical applications. Furthermore, our mathematical analysis establishes theoretical properties of the adaptive noise injection mechanism, including its unbiasedness and variance-preserving characteristics.

In summary, this paper makes the following contributions:

1. We introduce the NoiseFiT framework that leverages adaptive noise injection based on internal SNR, addressing the critical issue of hallucinations in LLMs **(RQ1)**.
2. We propose a novel hybrid loss function that integrates multiple regularization techniques to ensure robust and consistent model behavior **(RQ2)**.

3. We provide both theoretical analysis and empirical evidence demonstrating that our approach reduces hallucinations while preserving competitive performance across various tasks, while also maintaining computational efficiency and scalability **(RQ1, RQ2, RQ3)**.

Empirical evaluations conducted on multiple benchmarks, such as the GPQA (Rein et al., 2024), MUSR (Sprague et al., 2024), IFEval (Zhou et al., 2023), BBH (Suzgun et al., 2023), MATH (Hendrycks et al., 2021), MMLU-Pro (Wang et al., 2024), HaluEval (Li et al., 2023a), and TruthfulQA multiple-choice (Lin et al., 2022) datasets, validate the effectiveness of NoiseFiT. Our experiments demonstrate a noticeable reduction in hallucinations compared to conventional fine-tuning methods, while maintaining or even improving performance on standard language understanding tasks. These results are consistent with prior findings on the benefits of noise injection in both vision (Lehtinen et al., 2018) and language domains (Jain et al., 2024; Tice et al., 2024; Yadav & Singh, 2023).

## 3 METHODS

Our approach, *Noise-Augmented Fine-Tuning*, is designed to reduce hallucinations in LLMs by integrating adaptive noise injection directly into the fine-tuning process and leveraging a hybrid loss function. Additionally, NoiseFiT selectively injects noise into specific layers based on a SNR criterion, thereby adapting perturbations to target layers with both stable (robust) and unstable (loose) activations, depending on their contribution to model behavior.

### 3.1 DATASET

NoiseFiT was fine-tuned and evaluated using two distinct datasets, each comprising prompt-response pairs. The first of these, which serves as the fine-tuning dataset, includes 832 samples in its training split (with 208 samples in test split, structured similarly to the training), where each sample contains a prompt and its corresponding response. We intentionally designed this dataset to be simple and straightforward, with queries that, in many cases, large language models could be expected to answer easily. To achieve this, we generated the data synthetically using the GROK 3.0 Think, ensuring a broad but basic coverage of topics—from literature and history to geography and science—while maintaining concise prompts and direct responses. Our main motivation for constructing the dataset in this manner was twofold: (1) to demonstrate that our fine-tuning strategy remains effective even when starting from a minimal, uncomplicated dataset; and (2) to show that incorporating additional complexity is not strictly necessary to validate the viability of our approach. By emphasizing simplicity in content and structure, we also reduce potential confounding factors, allowing us to isolate and examine the impact of the fine-tuning methodology itself.

### 3.2 LAYER SELECTION VIA SNR

To determine which layers are suited for noise injection, we perform the following steps:

(a) **Clean and Noisy Forward Passes:** A clean forward pass through the model produces hidden states $\mathbf{h}_\ell^{\text{clean}} \in \mathbb{R}^{B \times L_\ell \times H}$, where $B$ is the batch size, $L_\ell$ the sequence length at layer $\ell$, and $H$ the hidden dimensionality. In parallel, multiple noisy forward passes using adaptive noise injection yield $\mathbf{h}_\ell^{\text{noisy}} \in \mathbb{R}^{B \times L_\ell \times H}$.

(b) **SNR Computation:** The SNR helps identify transformer layers with robust or loose activations. We define:

- **Signal Metric:** The signal $S_\ell$, computed as the mean absolute activation of the clean hidden states, is given by:

$$S_\ell = \frac{1}{B \cdot L_\ell \cdot H} \sum_{b=1}^{B} \sum_{t=1}^{L_\ell} \sum_{i=1}^{H} \left| \left[ \mathbf{h}_\ell^{\text{clean}} \right]_{b,t,i} \right|. \tag{1}$$

- **Noise Metric:** The noise $N_\ell$, estimated as the average absolute difference between noisy and clean activations, is:

$$N_\ell = \frac{1}{B \cdot L_\ell \cdot H} \sum_{b=1}^{B} \sum_{t=1}^{L_\ell} \sum_{i=1}^{H} \left| \left[ \mathbf{h}_\ell^{\text{noisy}} - \mathbf{h}_\ell^{\text{clean}} \right]_{b,t,i} \right|. \tag{2}$$

- **SNR Definition:** The SNR for layer $\ell$ is then:

$$\text{SNR}_\ell = \frac{S_\ell}{N_\ell + \epsilon}, \tag{3}$$

  where $\epsilon > 0$ (e.g., $10^{-6}$) avoids division by zero.

  A higher $S_\ell$ indicates stronger activations in the clean pass, while a lower $N_\ell$ implies less distortion under noise. Thus, higher $\text{SNR}_\ell$ suggests more stable activations under perturbation.

(c) **Layer Selection:** Given the SNR values $\{\text{SNR}_\ell\}_{\ell=1}^{\mathcal{L}}$ (with $\mathcal{L}$ the total number of layers), a fixed number $k$ of layers with the highest or lowest SNR values are selected for noise injection during fine-tuning.

## 3.3 ADAPTIVE NOISE INJECTION

Adaptive noise injection perturbs hidden states by injecting zero-mean Gaussian noise, scaled using robust statistics and model uncertainty. For a hidden state vector $\mathbf{h} \in \mathbb{R}^H$, noise injection is defined as:

$$\tilde{\mathbf{h}} = \mathbf{h} + \boldsymbol{\xi}, \quad \boldsymbol{\xi} \sim \mathcal{N}(\mathbf{0}, \mathbf{I}_H), \tag{4}$$

where $\mathbf{I}_H$ is the $H \times H$ identity matrix.

### 3.3.1 ADAPTIVE SCALING VIA ROBUST STATISTICS AND UNCERTAINTY

Noise is adaptively scaled using the median $\mu_{\text{med}} = \text{median}(\mathbf{h})$ and the Median Absolute Deviation (MAD):

$$\text{MAD}(\mathbf{h}) = \text{median}\left(|\mathbf{h} - \mu_{\text{med}}|\right). \tag{5}$$

**Exponential Weighting:** Define a weighting function that emphasizes deviations from the median:

$$w(\mathbf{h}) = \exp\left(-\beta \frac{|\mathbf{h} - \mu_{\text{med}}|}{\text{MAD}(\mathbf{h}) + \epsilon}\right), \tag{6}$$

where $\beta > 0$ is a hyperparameter controlling the sensitivity to deviations, and $\epsilon$ is a small constant to ensure numerical stability.

**Uncertainty-Based Noise Factor:** To capture model uncertainty, we define a noise factor $\eta$ that dynamically scales the noise magnitude. Two strategies are used to compute the noise factor ($\eta$):

- **Using Logits:** If logits $\mathbf{z} \in \mathbb{R}^{B \times L \times V}$, where $B$ is the batch size, $L$ is the sequence length, and $V$ is the vocabulary size (the number of unique tokens), are available, compute the softmax probabilities. The logits $\mathbf{z}$ are the raw, unnormalized scores output by the model, representing the likelihood of each token in the vocabulary at each sequence position. For each token position $t \in \{1, 2, \ldots, L\}$ and vocabulary token $k \in \{1, 2, \ldots, V\}$, the logit $z_{t,k}$ is the score for token $k$ at position $t$. The softmax probabilities are:

$$p_{t,k} = \frac{\exp(z_{t,k})}{\sum_{j=1}^{V} \exp(z_{t,j})}. \tag{7}$$

  Then, for each token position $t$, the entropy is:

$$\mathcal{H}_t = -\sum_{k=1}^{V} p_{t,k} \log\left(p_{t,k} + \epsilon\right). \tag{8}$$

  The average entropy over the token sequence is:

$$\bar{\mathcal{H}} = \frac{1}{L} \sum_{t=1}^{L} \mathcal{H}_t, \tag{9}$$

  and the noise factor is defined as:

$$\eta = \exp\left(\bar{\mathcal{H}}\right). \tag{10}$$

- **Variance of Hidden States:** In the absence of logits, a pseudo-entropy is computed from the variance of the hidden states:

$$\eta = \exp\left(\frac{\text{Var}(\mathbf{h})}{\mathbb{E}\big[\text{Var}(\mathbf{h})\big] + \epsilon}\right). \tag{11}$$

**Effective Noise Scale.** The final noise scale is computed by integrating base standard deviation $\sigma_{\text{base}}$, a learnable scalar $\alpha$, an external noise gate $g_{\text{noise}} \in [0, 1]$, the MAD, the weighting function, and the uncertainty factor:

$$\sigma_{\text{eff}} = \sigma_{\text{base}} \cdot \alpha \cdot g_{\text{noise}} \cdot \text{MAD}(\mathbf{h}) \cdot w(\mathbf{h}) \cdot \eta. \tag{12}$$

The perturbed hidden state is finally defined as:

$$\tilde{\mathbf{h}} = \mathbf{h} + \sigma_{\text{eff}} \cdot \boldsymbol{\xi}, \quad \boldsymbol{\xi} \sim \mathcal{N}(\mathbf{0}, \mathbf{I}_H). \tag{13}$$

### 3.4 HYBRID LOSS OBJECTIVE WITH CONSISTENCY REGULARIZATION

To ensure robust representation learning under noise, the training objective is augmented with multiple loss components:

#### 3.4.1 CROSS-ENTROPY LOSS ($\mathcal{L}_{\text{CE}}$)

A standard cross-entropy loss is computed on the clean forward pass:

$$\mathcal{L}_{\text{ce}} = -\frac{1}{N} \sum_{t=1}^{N} \log\Big(p\big(y_t \mid \mathbf{h}_t^{\text{clean}}\big)\Big), \tag{14}$$

where $N$ is the number of valid tokens and $y_t$ is the ground-truth token at iteration step $t$.

#### 3.4.2 SOFT CROSS-ENTROPY LOSS ($\mathcal{L}_{\text{SOFT}}$)

Inspired by knowledge distillation, we encourage the noisy model to align with the clean model's softened predictions (via temperature scaling), which provides informative soft targets and boosts calibration (Buciluǎ et al., 2006; Hinton et al., 2015; Guo et al., 2017). To further guide the model under noise, a soft target distribution is computed from the clean logits $\mathbf{z}^{\text{clean}}$ using temperature scaling:

$$p_t^{\text{soft}} = \text{softmax}\Big(\frac{\mathbf{z}_t^{\text{clean}}}{\tau}\Big), \tag{15}$$

where $\tau > 0$ is the temperature. For a noisy forward pass producing logits $\mathbf{z}^{\text{noisy}}$, the soft cross-entropy loss is:

$$\mathcal{L}_{\text{soft}} = \frac{1}{N} \sum_{t=1}^{N} \text{KL}\Big(p_t^{\text{soft}} \,\big\|\, \text{softmax}(\mathbf{z}_t^{\text{noisy}})\Big), \tag{16}$$

with $\text{KL}(\cdot \,\|\, \cdot)$ denoting the Kullback–Leibler divergence.

#### 3.4.3 CONSISTENCY LOSS ($\mathcal{L}_{\text{CONSISTENCY}}$)

To enforce stability across noisy passes, two independent noisy forward passes are performed yielding logits $\mathbf{z}_1^{\text{noisy}}$ and $\mathbf{z}_2^{\text{noisy}}$. The consistency loss is then defined as:

$$\mathcal{L}_{\text{consistency}} = \frac{1}{N} \sum_{t=1}^{N} \text{KL}\Big(\text{softmax}(\mathbf{z}_{1,t}^{\text{noisy}}) \,\big\|\, \text{softmax}(\mathbf{z}_{2,t}^{\text{noisy}})\Big). \tag{17}$$

#### 3.4.4 HYBRID LOSS AND FINAL TRAINING OBJECTIVE

The hybrid loss combines the clean and soft cross-entropy losses:

$$\mathcal{L}_{\text{hybrid}} = \lambda_{\text{ce}} \cdot \mathcal{L}_{\text{ce}} + (1 - \lambda_{\text{ce}}) \cdot \mathcal{L}_{\text{soft}}, \tag{18}$$

where $\lambda_{\text{ce}} \in [0, 1]$ balances the two objectives. The final training loss, incorporating the consistency regularization, is:

$$\mathcal{L}_{\text{final}} = \mathcal{L}_{\text{hybrid}} + \lambda_{\text{consistency}} \cdot \mathcal{L}_{\text{consistency}}, \tag{19}$$

with $\lambda_{\text{consistency}}$ governing the weight of the consistency loss.

# 4 EXPERIMENTS AND RESULTS

## 4.1 EXPERIMENTAL SETUP

We conducted experiments to assess the effectiveness of *NoiseFiT*, implemented using `PyTorch`'s multiprocessing on four Tesla V-100 GPUs (each equipped with 32 GB of GPU memory) (Varrette et al., 2022). Our method incorporates adaptive noise injection, a hybrid loss function, and parameter-efficient fine-tuning (PEFT) using Low-Rank Adaptation (LoRA). We used pre-trained causal language models as base models, varying across architectures including LLaMA (Grattafiori et al., 2024), Qwen (Bai et al., 2023), Gemma (Team, 2025), and Mistral (Jiang et al., 2023). The fine-tuning dataset was structured to include user prompts and assistant responses in a conversational format.

The fine-tuning process leverages the following key steps. First, the trainer selects layers for noise injection based on their SNR. The number of layers selected—set to 3 for most architectures, and to 3, 6, 12, and all layers in the case of larger model Mistral-7B—corresponded to those exhibiting the highest or lowest SNR values. These were identified using forward-pass statistics computed from both clean and noise-injected hidden states. Then, zero-mean Gaussian noise was injected into the hidden states of the selected layers. The noise was adaptively scaled based on hidden state statistics, using base standard deviations of 0.001, 0.01, and 0.1, further modulated by layer-specific scaling factors.

Finally, the training objective combines three components including a standard cross-entropy (CE) loss, a soft cross-entropy loss whose soft targets are computed from the *clean (noiseless)* logits via temperature scaling (teacher), and a consistency loss between two independently noise-injected *student* passes; see Eqs. (15)–(16). We used $\lambda_{ce} = 0.5$ and $\lambda_{consistency} = 0.1$ across all of the experiments. LoRA was applied with a rank of 8, targeting `q_proj` and `v_proj` modules, with an alpha of 16 and dropout of 0.05. In addition to the above settings, we set batch size of 4 per device, with 4 gradient accumulation steps, learning rate to $5 \times 10^{-5}$ (with a cosine scheduler, Appendix, Figure A.3). We used Paged AdamW in 32-bit (Loshchilov & Hutter, 2019), with mixed precision (FP16) and gradient clipping at 1.0 as the optimizer by setting number of epochs to 5, with a maximum of 1000 steps. Training histories were logged using Weights & Biases (Appendix, Figures A.1 and A.2) (Biewald, 2020).

## 4.2 RESULTS

### 4.2.1 LEADERBOARD BENCHMARKS AND HALLUCINATION EVALUATIONS

Table 1 summarizes the performance of various model configurations (extended evaluation results for Mistral-7B are provided in Table D.1), derived from the leaderboard evaluation task benchmarks (Gao et al., 2024), supplemented by the hallucination evaluation results using HaluEval (Li et al., 2023a) and TruthfulQA multiple-choice (TfQA-MC) (Lin et al., 2022) datasets. Each model family is evaluated by varying:

(i) **#Layers:** The number of layers selected for adaptive noise injection (where applicable), with `All` denoting full-layer injection and `BaseFiT` indicating a fine-tuning with no noise setup (we used cross-entropy loss for fine-tuning).

(ii) **STD:** The base standard deviation for noise was typically chosen from the set $\{0.01, 0.1\}$. Increasing this value results in stronger perturbations.

(iii) **SNR:** *Highest* layers first (favoring more robust activations) vs. *Lowest* layers first (targeting weaker activations), or `N/A` (e.g., when no targeted noise injection is performed).

**Impact of Noise Levels:** As demonstrated in Table 1, injecting noise at various levels (`STD`=0.01, 0.1, or 0.3) can confer notable performance advantages across multiple tasks and model families. Although higher noise levels (e.g., `STD`=0.3) are sometimes associated with greater instability, moderate levels (`STD`=0.01 or 0.1) frequently yield improvements in domains such as `Math` or `BBH`. For instance, Llama-1B exhibits enhanced `Math` accuracy (0.17) under `STD`=0.1 when targeting layers with high SNR, signifying that carefully calibrated noise can strengthen certain forms of reasoning.

**Layer Selection via SNR:** Results for *Highest* vs. *Lowest* SNR noise injected layers reveal that directing noise to specific layers can accentuate its benefits. For example, Mistral-7B achieves its highest `BBH` score (45.84) when noise is restricted to three *Lowest*-SNR layers, while Llama-1B attains superior `Math` performance (0.17) by injecting noise into the *Highest*-SNR layers. These outcomes highlight the importance of selectively targeting layers based on SNR profiles, indicating that the optimal approach may vary according to both the model architecture and the specific task objectives. Appendix G provides insights via SNR curves and five auxiliary metrics: sparsity, logit entropy, gradient cosine, L2 norm, and attention entropy.

Table 1: Benchmark and hallucination results across models, #Layers, STD, and SNR. Mistral-7B attains 52.34 on HaluEval with our approach, compared to 48.20 in CDCR-SFT (Li et al., 2025).

| Model | #Layers | STD | SNR | MMLU-Pro | BBH | GPQA | Math | IFEval | MUSR | TfQA-MC | HaluEval |
|---|---|---|---|---|---|---|---|---|---|---|---|
| Llama3.2-1B | 3 | 0.01 | Highest | **12.28** | 31.48 | 24.22 | 0.00 | 7.58 | 32.46 | 33.33 | 33.18 |
| | 3 | 0.01 | Lowest | 12.05 | **31.91** | **24.90** | 0.07 | 7.02 | 32.07 | 29.68 | 29.07 |
| | BaseFiT | N/A | N/A | 11.86 | 31.90 | 24.84 | 0.07 | **8.69** | 32.21 | 29.82 | 28.86 |
| | 3 | 0.1 | Highest | 11.79 | 31.72 | 23.99 | **0.17** | 7.95 | **33.54** | **33.77** | **33.67** |
| | 3 | 0.1 | Lowest | 11.67 | 31.05 | 24.56 | 0.05 | 6.47 | 32.75 | 30.99 | 28.13 |
| Llama3.2-3B | 3 | 0.01 | Lowest | **25.85** | 38.56 | 25.91 | 0.80 | 9.43 | 34.05 | 38.16 | 40.16 |
| | BaseFiT | N/A | N/A | 25.81 | 38.27 | 25.41 | 1.31 | 9.98 | 34.59 | 37.87 | 39.77 |
| | 3 | 0.1 | Highest | 25.52 | **39.38** | 27.83 | 1.32 | 8.87 | 33.79 | 38.74 | 41.24 |
| | 3 | 0.01 | Highest | 25.29 | 38.64 | 26.49 | **1.52** | 9.80 | **35.65** | **40.35** | **42.60** |
| | 3 | 0.1 | Lowest | 25.27 | 38.39 | **28.08** | 1.04 | **10.91** | 34.06 | 39.33 | 41.86 |
| Qwen2.5-.5B | 3 | 0.01 | Highest | **18.75** | 31.84 | **28.36** | 0.45 | 14.60 | 34.20 | 33.48 | 28.34 |
| | 3 | 0.01 | Lowest | 18.56 | 31.45 | 25.13 | 0.58 | 13.31 | **35.40** | 31.58 | 27.42 |
| | 3 | 0.1 | Lowest | 18.51 | 31.80 | 25.07 | 0.42 | 13.68 | 34.59 | 33.04 | 28.79 |
| | 3 | 0.1 | Highest | 18.29 | 31.77 | 28.08 | 0.68 | 12.20 | 34.05 | **34.65** | **29.12** |
| | BaseFiT | N/A | N/A | 17.43 | **32.17** | 26.89 | **0.70** | **15.16** | 34.59 | 31.58 | 27.08 |
| Gemma3-1B | BaseFiT | N/A | N/A | **14.92** | 35.11 | 28.00 | 4.74 | 37.52 | 32.75 | 21.35 | 24.37 |
| | 3 | 0.001 | Lowest | 14.85 | 34.25 | 27.88 | 4.40 | 34.57 | **33.95** | 21.49 | 25.08 |
| | 3 | 0.1 | Highest | 13.63 | **35.14** | 27.51 | 2.15 | 29.21 | 31.01 | 19.88 | 21.62 |
| | 3 | 0.01 | Lowest | 14.59 | 34.54 | 26.94 | 4.45 | 38.08 | 33.02 | 21.64 | 25.18 |
| | 3 | 0.01 | Highest | 14.37 | 34.84 | **28.39** | **5.08** | **39.37** | 33.41 | **21.78** | **25.39** |
| Mistral-7B | 6 | 0.1 | Highest | **30.28** | 44.63 | 29.46 | 2.69 | 13.31 | 39.51 | 38.20 | 51.40 |
| | BaseFiT | N/A | N/A | 30.01 | 44.34 | 29.29 | 2.97 | 11.46 | 38.84 | 37.62 | 47.60 |
| | 3 | 0.1 | Lowest | 29.97 | **45.84** | 28.07 | 3.01 | **13.49** | 39.89 | 39.41 | 50.72 |
| | 6 | 0.01 | Lowest | 29.74 | 45.08 | 29.66 | **3.43** | 11.83 | **40.18** | **41.24** | **52.34** |
| | 3 | 0.3 | Lowest | 29.53 | 44.53 | **30.54** | 2.75 | 10.91 | 39.51 | 37.26 | 49.40 |
| **Mean Top-5 Δ (%)** relative to BaseFiT | | | | +5.74↑ | +1.94↑ | +6.84↑ | +36.58↑ | +10.27↑ | +3.40↑ | +10.18↑ | +11.42↑ |

**BaseFiT vs Noise-Injected Runs:** Comparisons with the `BaseFiT` baseline (fine-tuning without noise) underscore the capacity of noise injection to surpass baseline results in multiple settings. For instance, Qwen-0.5B with `STD`=0.01 in the high-SNR configuration outperforms `BaseFiT` on `MMLU-Pro` (18.75 vs. 17.43) and `GPQA` (28.36 vs. 26.89). Similarly, Gemma-1B realizes substantial gains in majority of the tasks under targeted noise conditions. These findings demonstrate that noise-injected configurations can frequently exceed baseline performance when the noise parameters and layer selections are carefully optimized.

**Task-Specific Observations:** Across a diverse set of evaluation benchmarks, the impact of noise injection varies by task type and model architecture, but notable patterns emerge. In several cases, injecting moderate levels of noise appears to improve performance, suggesting it may act as a form of regularization or stimulus for deeper reasoning:

- **Math:** Table 1 shows that moderate noise (`STD`=0.01 or 0.1) can substantially improve Math accuracy. For example, Llama-1B's score rises from 0.05 to 0.17 under `STD`=0.1 (highest-SNR layers), and Gemma-1B reaches 5.08 (vs. 4.74) under `STD`=0.01 (highest SNR). These improvements suggest that carefully tuned noise benefits numerical reasoning.

- **BBH and MMLU-Pro:** These broader language understanding benchmarks often show moderate fluctuation with noise, yet select configurations demonstrate that noise can push performance above the baseline. In Llama-3B, for example, `BBH` rises to 39.38 under `STD`=0.1 (highest SNR), exceeding the BaseFiT score of 38.27. On MMLU-Pro, for example, Qwen-0.5B at `STD`=0.01 (highest SNR) rises to 18.75 from a baseline of 17.43. These results confirm that noise, particularly at moderate levels, can be harnessed to refine performance in language understanding tasks.

- **GPQA:** Detailed inspection of the GPQA results shows consistent gains under targeted noise strategies. Llama-3B moves from 25.41 (BaseFiT) to 27.83 (`STD=0.1`, Highest SNR), while Qwen-0.5B increases from 26.89 (BaseFiT) to 28.36 (`STD=0.01`, Highest SNR). Gemma-1B also achieves its top GPQA score (28.39) with `STD=0.01` in the Highest-SNR layers, surpassing the base 28.00. Notably, Mistral-7B records 30.54 (`STD=0.3`, three Lowest-SNR layers), indicating that, with the right noise level and layer selection, graduate-level question-answering performance can be enhanced across various model families.

- **IFEval and MUSR:** Noise can also yield performance improvements in following instructions (IFEval) and multistep soft reasoning (MUSR). In Gemma-1B, IFEval rises from 37.52 (BaseFiT) to 39.37 (`STD=0.01`, Highest SNR), and MUSR increases from 32.75 to 33.41 under the same setting. Likewise, Mistral-7B achieves up to 13.49 on IFEval (`STD=0.1`, 3 Lowest-SNR layers) compared to 11.46 at BaseFiT, and elevates MUSR from 38.84 (BaseFiT) to 40.18 (`STD=0.01`, 6 Lowest-SNR layers).

**Consistency Across Model Families:** Despite variability in architecture and scale (Llama, Qwen, Gemma, Mistral), several consistent observations emerge:

- **Efficacy of moderate noise across model-task configurations:** While high noise magnitudes (e.g., `STD=0.3`) can induce instability in performance, moderate perturbation levels (`STD=0.01` or 0.1) frequently yield consistent gains across a range of tasks and architectures.

- **Targeted perturbation via layer-wise selection:** Constraining noise application to specific layer subsets—such as those with the *highest* or *lowest* signal-to-noise ratios—enables more precise control over performance modulation, highlighting the utility of structurally selective noise injection.

- **Augmenting `BaseFiT` with stochastic refinement:** Although `BaseFiT` establishes a robust baseline, many noise-augmented configurations achieve comparable or superior results on specific benchmarks, suggesting that noise injection can function as an effective complement or enhancement to traditional fine-tuning methodologies.

### 4.2.2 TEST DATASET

We evaluated our fine-tuning approach using a test set of 208 unique prompts. We employed the same models as the base, incorporating a PEFT adapter. The entire generation procedure was accelerated across multiple GPUs. We employed GROK 3.0 Think (xAI, 2025) to assess the hallucination performance in the generated responses (online supplementary material).

The results illustrate the effects of noise injection on the performance of various models across multiple categories (Appendix, Tables E.2-E.6). A consistent trend observed across models is that noise-injection under various scenarios, often outperform their respective base models, suggesting that controlled noise can improve the models' ability to produce less hallucinated responses and handle diverse inputs, potentially by mimicking real-world data variability.

## 5 DISCUSSION AND LIMITATIONS

Across four model families, multiple leaderboards and hallucination evaluation datasets, `NoiseFiT` routinely matches or outperforms standard fine-tuning (`BaseFiT`) while reducing hallucinations. Moderate perturbation magnitudes are consistently effective, whereas large magnitudes can be unstable (Table 1). Extended ablations on Mistral-7B show that *targeted* injection into a small subset of layers (3–6) dominates *all-layers* injection (Table D.1). Beyond point estimates, distributional analysis with the Epps–Singleton test confirms statistically reliable shifts in hallucination scores for noisy variants versus base models (§E.2).

Appendix G reveals complementary trends that clarifies mechanics of SNR injection. In larger models such as Mistral-7B, *low-SNR* layers deeper in the stack behave as pattern amplifiers; injecting noise there increases gradient diversity, elevates attention entropy slightly, and curbs memorization-driven errors—aligning with the finding that *lowest-SNR* targeting works best for Mistral-7B. In smaller models, *high-SNR* layers are already robust; adding moderate noise functions as gentle regularization without destabilizing training, which matches the gains we observe with *highest-SNR* targeting.

Across families, we also see (i) higher sparsity and (ii) modestly higher attention entropy under NoiseFiT, both consistent with better generalization and less over-confident false assertions.

The training objective blends: (a) standard CE for task fidelity, (b) *soft-target* cross-entropy that distills from the *clean* pass (teacher) at temperature $\tau$, and (c) a consistency term across two independently noised passes. Together, these terms encourage calibrated logits and stability to perturbations, which concretely reduces the model's tendency to *run away* from weak evidence. Our theory (Appendix B) supports this picture. Appendix C summarizes our heuristics as a practical recipe for NoiseFiT.

**Limitations.**

- We primarily compare against BaseFiT. Methods like RAG, RLHF, self-consistency, and con-trastive decoding optimize orthogonal axes (retrieval, reward shaping, inference-time ensembling). A controlled, apples-to-apples comparison is non-trivial; we therefore frame NoiseFiT as *comple-mentary*. (We provide preliminary truthfulness gains on TruthfulQA/HaluEval in Table 1.)

- Our core fine-tuning set is compact and partly synthetic; and hallucinations on the custom test set are assessed by an LLM judge and corroborated through human evaluation. To mitigate bias we report public-benchmark results (TfQA-MC, HaluEval) and provide distributional tests (§E.2).

- Although the heuristics above work well, **per-task tuning is encouraged** based on our findings. In practice, gains are sensitive to the choice of $k$ perturbed layers, the base noise scale and ramp; mis-setting these can yield non-monotonic behavior. The SNR bootstrap that ranks layers from most stable to most unstable relies on brief noisy forward passes and can be brittle under domain shift or when validation sets are small; longer pilots or robust smoothing/averaging may be required.

- The SNR-based targeting rationale is supported by auxiliary metrics (Appendix G), but we do not claim a full causal account. Understanding functional roles of *low-SNR* vs. *high-SNR* layers across architectures is an active direction.

Overall, NoiseFiT is a simple, training-time technique that improves factual reliability without inference-time cost, but it benefits from small, targeted sweeps and broader evaluations to fully establish external validity.

## 6 CONCLUSION AND FUTURE WORK

We introduced *NoiseFiT*, an adaptive, SNR-guided noise-augmented fine-tuning framework with a hybrid objective. Empirically, across LLaMA, Qwen, Gemma, and Mistral, moderate noise injected into a small, SNR-selected subset of layers yields reliable improvements on truthfulness and reasoning leaderboards (Table 1), with extended Mistral-7B ablations favoring 3–6 targeted layers (Table D.1). Notably, on hallucination-focused benchmarks, NoiseFiT achieves average gains of 3.72% on TfQA-MC and 4.70% on HaluEval across models.

Overall, we find that *targeted* layer-wise noise outperforms blanket perturbation; *moderate* magni-tudes (STD 0.01–0.1) are consistently best; the optimal layer choice is *model dependent*—larger models benefit from injecting into lowest-SNR (less stable) layers, whereas smaller models benefit from highest-SNR (more robust) layers; and the gains stem chiefly from improved calibration and perturbation stability rather than raw accuracy alone. NoiseFiT is training-time only, adds no test-time latency, and is compatible with retrieval (RAG), RLHF, self-consistency, and decoding-time defenses. We view it as a lightweight regularizer that improves the base model's factual reliability before any downstream alignment or retrieval stack is applied.

We see two immediate directions: (1) an open-source auto-tuner that (a) bootstraps SNR with a short warm-up, (b) selects $k$ via validation hallucination proxies, and (c) schedules noise ramps; (2) testing BitFit and AdaLoRA (Ben Zaken et al., 2022; Zhang et al., 2023), and composing NoiseFiT with RAG/RLHF at scale.

By turning a small amount of *targeted* stochasticity into a principled regularizer, NoiseFiT improves robustness and reduces hallucinations with minimal engineering burden. We hope the simple recipe, theoretical guarantees, and practical heuristics make it a useful building block for reliable LLM adaptation.

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

# A   NoiseFiT Algorithm

Algorithm A.1 summarizes the operational mechanics of the NoiseFiT framework. It outlines the key steps involved in our approach—from data and model preparation, performing clean and noisy forward passes, and computing the SNR for each transformer layer, to select layers for adaptive noise injection and finally calculating the hybrid loss components for backpropagation. This high-level structure serves as a blueprint for implementing our NoiseFiT algorithm.

---

**Algorithm A.1** NoiseFiT Algorithm

---

1: **Input:** Training data $\mathcal{D}$, pretrained model $\mathcal{M}$, number of layers $k$, hyperparameters $\lambda_{\text{ce}}$, $\lambda_{\text{consistency}}$, $\tau$, etc.
2: **Output:** Fine-tuned model $\mathcal{M}^*$
3: **Step 1: Data and Model Preparation**
4:     Load dataset and format each sample (prompt, response).
5:     Initialize tokenizer and model $\mathcal{M}$.
6: **Step 2: Clean Forward Pass**
7:     For a batch, run a clean forward pass to obtain hidden states $\mathbf{h}_\ell^{\text{clean}} \in \mathbb{R}^{B \times L_\ell \times H}$.
8: **Step 3: Noisy Forward Passes**
9:     Enable noise hooks for the forward passes.
10:    Run multiple forward passes with adaptive noise injection to obtain $\mathbf{h}_\ell^{\text{noisy}} \in \mathbb{R}^{B \times L_\ell \times H}$.
11: **Step 4: SNR Computation (for all of the layers $\ell$)**
12:    Compute signal: $S_\ell = \frac{1}{B \cdot L_\ell \cdot H} \sum_{b,t,i} \left| \left[\mathbf{h}_\ell^{\text{clean}}\right]_{b,t,i} \right|$.
13:    Compute noise: $N_\ell = \frac{1}{B \cdot L_\ell \cdot H} \sum_{b,t,i} \left| \left[\mathbf{h}_\ell^{\text{noisy}} - \mathbf{h}_\ell^{\text{clean}}\right]_{b,t,i} \right|$.
14:    Compute SNR: $\text{SNR}_\ell = \frac{S_\ell}{N_\ell + \epsilon}$.
15: **Step 5: Layer Selection**
16:    Select $k$ layers with the highest (or lowest) $\text{SNR}_\ell$ values for noise injection.
17: **Step 6: Loss Computation**
18:    **(a) Cross-Entropy Loss:** $\mathcal{L}_{\text{ce}} = -\frac{1}{N} \sum_{t=1}^{N} \log\big(p(y_t \mid \mathbf{h}_t^{\text{clean}})\big)$.
19:    **(b) Soft Cross-Entropy Loss:** $\mathcal{L}_{\text{soft}} = \frac{1}{N} \sum_{t=1}^{N} \text{KL}\Big(p_t^{\text{soft}} \,\big\|\, \text{softmax}(\mathbf{z}_t^{\text{noisy}})\Big)$, where $p_t^{\text{soft}} = \text{softmax}\Big(\frac{\mathbf{z}_t^{\text{clean}}}{\tau}\Big)$.
20:    **(c) Consistency Loss:** $\mathcal{L}_{\text{consistency}} = \frac{1}{N} \sum_{t=1}^{N} \text{KL}\Big(\text{softmax}(\mathbf{z}_{1,t}^{\text{noisy}}) \,\big\|\, \text{softmax}(\mathbf{z}_{2,t}^{\text{noisy}})\Big)$.
21: **Step 7: Final Loss and Backpropagation**
22:    Compute hybrid loss: $\mathcal{L}_{\text{hybrid}} = \lambda_{\text{ce}} \mathcal{L}_{\text{ce}} + (1 - \lambda_{\text{ce}}) \mathcal{L}_{\text{soft}}$.
23:    Compute overall loss: $\mathcal{L}_{\text{final}} = \mathcal{L}_{\text{hybrid}} + \lambda_{\text{consistency}} \mathcal{L}_{\text{consistency}}$.
24:    Backpropagate $\mathcal{L}_{\text{final}}$ and update model parameters.

---

# B   NoiseFiT Mathematical Foundations

In this section, we introduce the theoretical underpinnings of our NoiseFiT framework. We begin by outlining the core assumptions for unbiased noise injection and describe how these assumptions inform the variance-preserving characteristics of our approach. In particular, we provide high-level insights into why adaptive noise regularization improves generalization and stability, setting the stage for the formal lemmas and theorems that follow.

## B.1   Unbiased Noise Injection and Variance Preservation

### B.1.1   Zero-Mean Noise

**Lemma B.1.** *Let $\boldsymbol{\xi}$ be an $n$-dimensional random vector distributed as*

$$\boldsymbol{\xi} \sim \mathcal{N}(\mathbf{0}, \mathbf{I}),$$

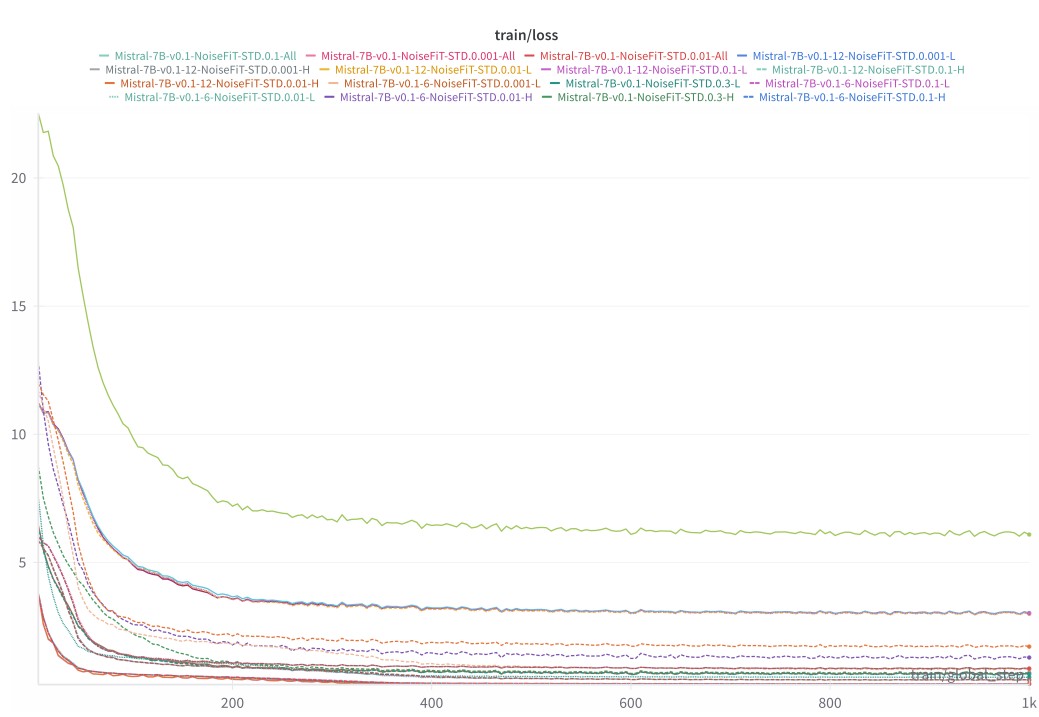

Figure A.1: NoiseFiT training loss history per model, noise injection STD and layer selection strategy. Available in interactive mode online at W&B.

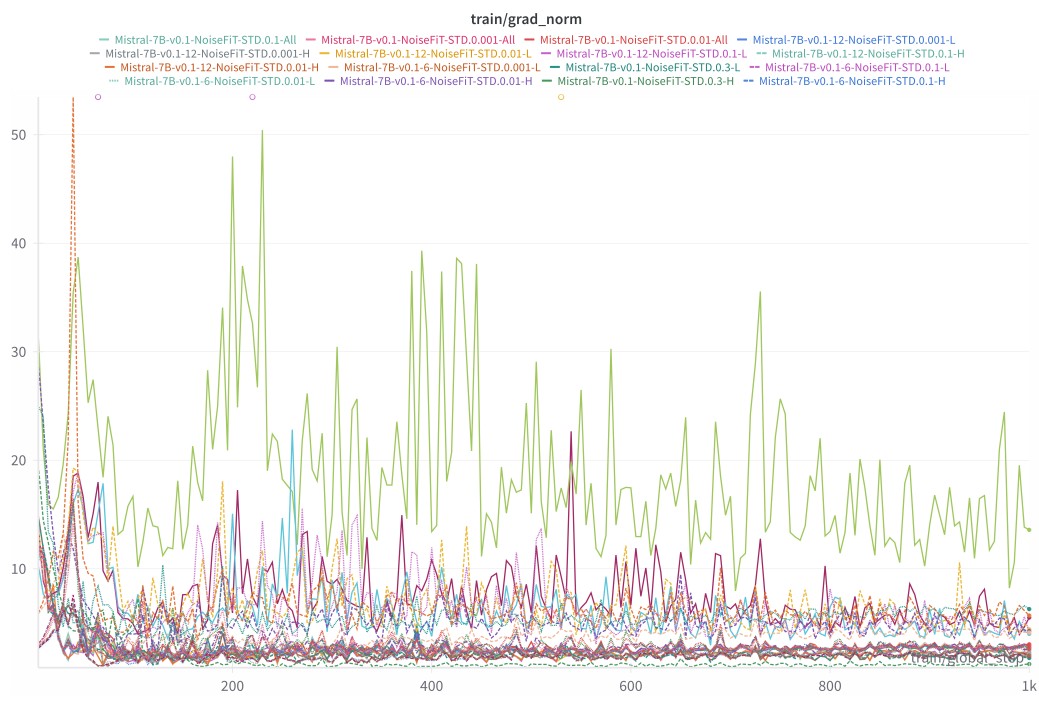

Figure A.2: NoiseFiT training gradients norm history per fine-tuning step for different models, noise injection STD and layer selection strategies. Available in interactive mode online at W&B.

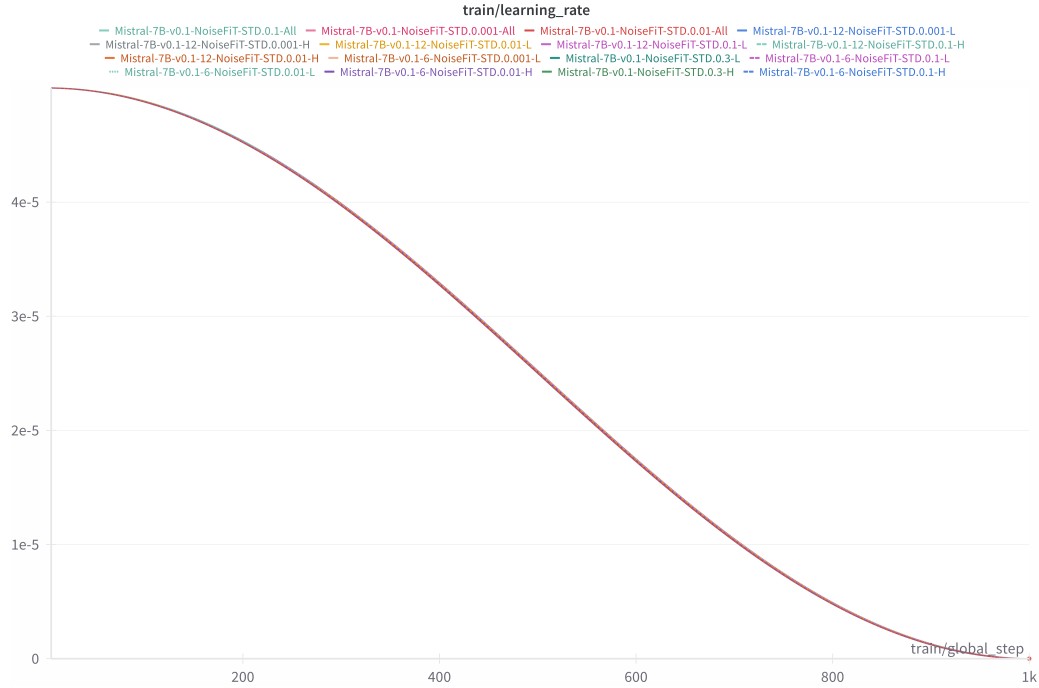

Figure A.3: NoiseFiT training learning rate scheduling strategy across all experiments. Available in interactive mode online at W&B.

*that is, each component $\xi_i$ of $\boldsymbol{\xi}$ is an independent standard normal random variable with mean 0 and variance 1. Then the expectation of $\boldsymbol{\xi}$ is the zero vector:*

$$\mathbb{E}[\boldsymbol{\xi}] = \mathbf{0}.$$

*Proof.* A random vector $\boldsymbol{\xi} \in \mathbb{R}^n$ has the distribution $\mathcal{N}(\mathbf{0}, \mathbf{I})$ if and only if each component $\xi_i$ (for $i = 1, 2, \ldots, n$) is distributed according to a one-dimensional standard normal distribution $\mathcal{N}(0, 1)$, and all components are mutually independent. The joint PDF of $\boldsymbol{\xi}$ can be written as

$$f_{\boldsymbol{\xi}}(\mathbf{x}) = \frac{1}{\sqrt{(2\pi)^n}} \exp\left(-\tfrac{1}{2}\|\mathbf{x}\|^2\right),$$

where $\|\mathbf{x}\|^2 = \sum_{i=1}^n x_i^2$. This density is *spherically symmetric* around the origin $\mathbf{0} \in \mathbb{R}^n$. The expectation of $\boldsymbol{\xi}$ is the vector of expectations of its components:

$$\mathbb{E}[\boldsymbol{\xi}] = \big(\mathbb{E}[\xi_1], \mathbb{E}[\xi_2], \ldots, \mathbb{E}[\xi_n]\big).$$

Equivalently, we can write

$$\mathbb{E}[\boldsymbol{\xi}] = \int_{\mathbb{R}^n} \mathbf{x}\, f_{\boldsymbol{\xi}}(\mathbf{x})\, d\mathbf{x}.$$

Since $\xi_i \sim \mathcal{N}(0, 1)$ for each $i$, we know by the definition of the standard normal distribution that

$$\mathbb{E}[\xi_i] = 0, \quad \text{for each } i = 1, 2, \ldots, n.$$

Hence, immediately we have

$$\mathbb{E}[\boldsymbol{\xi}] = \big(0, 0, \ldots, 0\big).$$

We can also see this from the integral form. For each component $\xi_i$,

$$\mathbb{E}[\xi_i] = \int_{-\infty}^{\infty} x_i \left( \int_{\mathbb{R}^{n-1}} f_{\boldsymbol{\xi}}(x_1, \ldots, x_{i-1}, x_i, x_{i+1}, \ldots, x_n)\, dx_1 \cdots dx_{i-1} dx_{i+1} \cdots dx_n \right) dx_i.$$

Because $f_{\boldsymbol{\xi}}(\mathbf{x})$ is an even function in each $x_i$ due to its form $\exp(-\frac{1}{2}\|\mathbf{x}\|^2)$ and the domain of integration is symmetric ($\mathbb{R}^n$), the integral of $x_i f_{\boldsymbol{\xi}}(\mathbf{x})$ over $\mathbb{R}^n$ is zero. This confirms $\mathbb{E}[\xi_i] = 0$ for every $i$. By combining the component-wise results, it follows that

$$\mathbb{E}[\boldsymbol{\xi}] \;=\; \big(\mathbb{E}[\xi_1], \mathbb{E}[\xi_2], \ldots, \mathbb{E}[\xi_n]\big) \;=\; \big(0, 0, \ldots, 0\big) \;=\; \mathbf{0}.$$

This completes the proof. $\qquad\square$

### B.1.2 Unbiasedness of Noisy Representations

**Theorem B.1.** *Let $\mathbf{h} \in \mathbb{R}^n$ be a deterministic hidden state and consider its noisy counterpart*

$$\tilde{\mathbf{h}} \;=\; \mathbf{h} \;+\; \sigma_{\mathit{eff}} \cdot \boldsymbol{\xi},$$

*where $\sigma_{\mathit{eff}} \in \mathbb{R}$ is a deterministic noise scale. Then,*

$$\mathbb{E}[\tilde{\mathbf{h}}] \;=\; \mathbf{h}.$$

*Proof.* We wish to show that the expectation of the noisy hidden state $\tilde{\mathbf{h}}$, which is formed by adding Gaussian noise scaled by $\sigma_{\text{eff}}$, remains equal to the original deterministic hidden state $\mathbf{h}$. Mathematically, we want to prove $\mathbb{E}[\tilde{\mathbf{h}}] = \mathbf{h}$. By definition, we have

$$\tilde{\mathbf{h}} \;=\; \mathbf{h} \;+\; \sigma_{\text{eff}} \cdot \boldsymbol{\xi}.$$

Since $\mathbf{h}$ and $\sigma_{\text{eff}}$ are deterministic, the only source of randomness in $\tilde{\mathbf{h}}$ is $\boldsymbol{\xi}$. One of the key properties we use is that expectation is a *linear operator*, which means:

$$\mathbb{E}[\, a\mathbf{X} + b\mathbf{Y} \,] \;=\; a\,\mathbb{E}[\mathbf{X}] + b\,\mathbb{E}[\mathbf{Y}],$$

for any random vectors $\mathbf{X}, \mathbf{Y}$ and scalars (deterministic constants) $a, b \in \mathbb{R}$. Applying this to $\tilde{\mathbf{h}} = \mathbf{h} + \sigma_{\text{eff}}\boldsymbol{\xi}$, we obtain:

$$\mathbb{E}[\tilde{\mathbf{h}}] \;=\; \mathbb{E}[\mathbf{h} + \sigma_{\text{eff}}\boldsymbol{\xi}] \;=\; \mathbb{E}[\mathbf{h}] \;+\; \mathbb{E}[\sigma_{\text{eff}}\boldsymbol{\xi}].$$

Since $\mathbf{h}$ is *not* random, its expectation is simply:

$$\mathbb{E}[\mathbf{h}] \;=\; \mathbf{h}.$$

Conceptually, viewing $\mathbf{h}$ as fixed means that integrating (or summing) over its distribution does not introduce any randomness. Next, consider $\mathbb{E}[\sigma_{\text{eff}}\boldsymbol{\xi}]$. Because $\sigma_{\text{eff}}$ is a constant (deterministic with respect to the random vector $\boldsymbol{\xi}$), it factors out of the expectation:

$$\mathbb{E}[\sigma_{\text{eff}}\boldsymbol{\xi}] \;=\; \sigma_{\text{eff}}\,\mathbb{E}[\boldsymbol{\xi}].$$

This step relies again on the linearity of expectation and the property that constants can be pulled out of expectation. By Lemma B.1 (Zero-Mean Noise), we know that $\mathbb{E}[\boldsymbol{\xi}] = \mathbf{0}$. Substituting this result, we get:

$$\mathbb{E}[\sigma_{\text{eff}}\boldsymbol{\xi}] = \sigma_{\text{eff}} \cdot \mathbf{0} = \mathbf{0}.$$

Putting all the above together:

$$\mathbb{E}[\tilde{\mathbf{h}}] = \mathbb{E}[\mathbf{h}] + \mathbb{E}[\sigma_{\text{eff}}\boldsymbol{\xi}] = \mathbf{h} + \mathbf{0} = \mathbf{h}.$$

Hence the noisy representation $\tilde{\mathbf{h}}$ is *unbiased*, completing the proof. $\qquad\square$

### B.1.3 Variance Preservation

**Lemma B.2.** *Assume that $\mathbf{h}$ and $\boldsymbol{\xi}$ are independent. Recalling the definitions from Theorem B.1, the covariance of $\tilde{\mathbf{h}}$ is given by*

$$\mathrm{Cov}[\tilde{\mathbf{h}}] \;=\; \mathrm{Cov}[\mathbf{h}] \;+\; \sigma_{\mathit{eff}}^2\,\mathbf{I}.$$

*Proof.* For an $n$-dimensional random vector $\mathbf{X}$, the covariance matrix is

$$\mathrm{Cov}[\mathbf{X}] \;=\; \mathbb{E}\big[(\mathbf{X} - \mathbb{E}[\mathbf{X}])(\mathbf{X} - \mathbb{E}[\mathbf{X}])^\top\big].$$

Covariance is a bilinear operator, meaning that if $\mathbf{X}$ and $\mathbf{Y}$ are random vectors, then

$$\mathrm{Cov}[\mathbf{X} + \mathbf{Y}] \;=\; \mathrm{Cov}[\mathbf{X}] \;+\; \mathrm{Cov}[\mathbf{Y}] \;+\; 2\,\mathrm{Cov}[\mathbf{X}, \mathbf{Y}].$$

Since $\tilde{\mathbf{h}} = \mathbf{h} + \sigma_{\mathrm{eff}}\,\boldsymbol{\xi}$, we write

$$\mathrm{Cov}[\tilde{\mathbf{h}}] \;=\; \mathrm{Cov}[\mathbf{h} + \sigma_{\mathrm{eff}}\,\boldsymbol{\xi}] \;=\; \mathrm{Cov}[\mathbf{h}] \;+\; \mathrm{Cov}[\sigma_{\mathrm{eff}}\,\boldsymbol{\xi}] \;+\; 2\,\mathrm{Cov}\big[\mathbf{h},\, \sigma_{\mathrm{eff}}\,\boldsymbol{\xi}\big].$$

This follows directly from the bilinear expansion of covariance. We are given that $\mathbf{h}$ and $\boldsymbol{\xi}$ are independent. By definition, if two random vectors $\mathbf{A}$ and $\mathbf{B}$ are independent, then $\mathbb{E}[\mathbf{A}\mathbf{B}^\top] = \mathbb{E}[\mathbf{A}]\mathbb{E}[\mathbf{B}]^\top$. It follows that

$$\mathrm{Cov}\big[\mathbf{h},\, \boldsymbol{\xi}\big] \;=\; \mathbb{E}\big[(\mathbf{h} - \mathbb{E}[\mathbf{h}])(\boldsymbol{\xi} - \mathbb{E}[\boldsymbol{\xi}])^\top\big] \;=\; \mathbf{0},$$

because $\mathbb{E}[\boldsymbol{\xi}] = \mathbf{0}$ and $\mathbf{h}, \boldsymbol{\xi}$ are independent. Therefore,

$$\mathrm{Cov}\big[\mathbf{h}, \sigma_{\mathrm{eff}}\,\boldsymbol{\xi}\big] \;=\; \sigma_{\mathrm{eff}}\,\mathrm{Cov}[\mathbf{h}, \boldsymbol{\xi}] \;=\; \sigma_{\mathrm{eff}} \cdot \mathbf{0} \;=\; \mathbf{0}.$$

Hence, the cross-covariance term vanishes. Next, we analyze $\mathrm{Cov}[\sigma_{\mathrm{eff}}\,\boldsymbol{\xi}]$. If $\boldsymbol{\xi} \sim \mathcal{N}(\mathbf{0}, \mathbf{I})$, its covariance is $\mathbf{I}$. For any deterministic scalar $\alpha$, scaling a random vector $\mathbf{X}$ by $\alpha$ scales its covariance matrix by $\alpha^2$. Hence,

$$\mathrm{Cov}[\sigma_{\mathrm{eff}}\,\boldsymbol{\xi}] \;=\; \sigma_{\mathrm{eff}}^2\,\mathrm{Cov}[\boldsymbol{\xi}] \;=\; \sigma_{\mathrm{eff}}^2\,\mathbf{I}.$$

Putting everything together, we get

$$\mathrm{Cov}[\tilde{\mathbf{h}}] \;=\; \mathrm{Cov}[\mathbf{h}] \;+\; \sigma_{\mathrm{eff}}^2\,\mathbf{I} \;+\; 2 \cdot \mathbf{0} \;=\; \mathrm{Cov}[\mathbf{h}] \;+\; \sigma_{\mathrm{eff}}^2\,\mathbf{I}.$$

Thus, adding independent Gaussian noise with variance $\sigma_{\mathrm{eff}}^2$ to the random vector $\mathbf{h}$ increases its covariance by $\sigma_{\mathrm{eff}}^2\mathbf{I}$, preserving the original variances plus a constant isotropic inflation. $\qquad\square$

## B.2 LIPSCHITZ CONTINUITY OF THE ADAPTIVE NOISE INJECTION

**Lemma B.3.** *Let $\mathbf{h} \in \mathbb{R}^n$ lie in a bounded set (so there exists some $\Omega > 0$ with $\|\mathbf{h}\| \leq \Omega$ for all relevant $\mathbf{h}$). Suppose the noise-scale function $\sigma_{eff}(\mathbf{h})$ is Lipschitz continuous with constant $L_\sigma > 0$; i.e.,*

$$\big|\sigma_{eff}(\mathbf{h}_1) \;-\; \sigma_{eff}(\mathbf{h}_2)\big| \;\leq\; L_\sigma\,\|\mathbf{h}_1 - \mathbf{h}_2\| \quad \forall\,\mathbf{h}_1, \mathbf{h}_2.$$

*Define the mapping (for a fixed realization of $\boldsymbol{\xi}$)*

$$\mathcal{T}(\mathbf{h}) \;=\; \mathbf{h} \;+\; \sigma_{eff}(\mathbf{h})\,\boldsymbol{\xi}.$$

*Then $\mathcal{T}(\mathbf{h})$ is Lipschitz continuous almost surely in $\boldsymbol{\xi}$.*

*Proof.* We need to show that there exists a (random) constant $L$ such that

$$\|\mathcal{T}(\mathbf{h}_1) - \mathcal{T}(\mathbf{h}_2)\| \;\leq\; L\,\|\mathbf{h}_1 - \mathbf{h}_2\|,$$

for all $\mathbf{h}_1$ and $\mathbf{h}_2$ in our domain, except on an event of probability zero (hence the phrase *almost surely*). Recall

$$\mathcal{T}(\mathbf{h}) = \mathbf{h} + \sigma_{\mathrm{eff}}(\mathbf{h})\,\boldsymbol{\xi}.$$

While $\mathbf{h}$ is a variable in $\mathbb{R}^n$, $\boldsymbol{\xi}$ is a random vector. Once $\boldsymbol{\xi}$ is fixed, $\mathcal{T}$ becomes a deterministic function of $\mathbf{h}$. For any two points $\mathbf{h}_1, \mathbf{h}_2 \in \mathbb{R}^n$, consider:

$$\mathcal{T}(\mathbf{h}_1) - \mathcal{T}(\mathbf{h}_2) \;=\; \big(\mathbf{h}_1 + \sigma_{\mathrm{eff}}(\mathbf{h}_1)\,\boldsymbol{\xi}\big) - \big(\mathbf{h}_2 + \sigma_{\mathrm{eff}}(\mathbf{h}_2)\,\boldsymbol{\xi}\big) \;=\; (\mathbf{h}_1 - \mathbf{h}_2) + \big(\sigma_{\mathrm{eff}}(\mathbf{h}_1) - \sigma_{\mathrm{eff}}(\mathbf{h}_2)\big)\boldsymbol{\xi}.$$

Hence, by the triangle inequality,

$$\|\mathcal{T}(\mathbf{h}_1) - \mathcal{T}(\mathbf{h}_2)\| \;\leq\; \|\mathbf{h}_1 - \mathbf{h}_2\| \;+\; \big|\sigma_{\mathrm{eff}}(\mathbf{h}_1) - \sigma_{\mathrm{eff}}(\mathbf{h}_2)\big|\,\|\boldsymbol{\xi}\|.$$

Since $\sigma_{\mathrm{eff}}(\mathbf{h})$ is Lipschitz with constant $L_\sigma$, we have

$$\big|\sigma_{\mathrm{eff}}(\mathbf{h}_1) - \sigma_{\mathrm{eff}}(\mathbf{h}_2)\big| \;\leq\; L_\sigma\,\|\mathbf{h}_1 - \mathbf{h}_2\|.$$

Therefore,
$$\|\mathcal{T}(\mathbf{h}_1) - \mathcal{T}(\mathbf{h}_2)\| \;\leq\; \|\mathbf{h}_1 - \mathbf{h}_2\| \;+\; L_\sigma \, \|\mathbf{h}_1 - \mathbf{h}_2\| \, \|\boldsymbol{\xi}\| \;=\; \big(1 + L_\sigma \|\boldsymbol{\xi}\|\big) \, \|\mathbf{h}_1 - \mathbf{h}_2\|.$$
For a fixed realization of $\boldsymbol{\xi}$, define
$$L(\boldsymbol{\xi}) \;=\; 1 + L_\sigma \, \|\boldsymbol{\xi}\|.$$
Clearly, $L(\boldsymbol{\xi})$ is a (finite) constant whenever $\boldsymbol{\xi}$ is given. Thus $\mathcal{T}$ is Lipschitz continuous with Lipschitz constant $L(\boldsymbol{\xi})$. Since $\|\boldsymbol{\xi}\|$ might be unbounded theoretically, the mapping $\mathcal{T}$ may have different Lipschitz constants for different realizations of $\boldsymbol{\xi}$. However, $\boldsymbol{\xi}$ is almost surely finite (i.e., the probability that $\|\boldsymbol{\xi}\|$ is infinite is zero). Hence, $\mathcal{T}$ is almost surely Lipschitz continuous with constant $1 + L_\sigma \|\boldsymbol{\xi}\|$. This completes the proof. $\qquad\square$

### B.3 STABILITY OF HYBRID LOSS GRADIENTS

**Lemma B.4.** *Let $f(\mathbf{h})$ denote the (scalar or vector) output of a neural network as a differentiable function of the hidden state $\mathbf{h} \in \mathbb{R}^n$. Suppose:*

1. *The noise injection is unbiased (Theorem B.1), i.e., $\mathbb{E}[\boldsymbol{\xi}] = \mathbf{0}$.*

2. *The noise variance is bounded by $\sigma^2_{\text{eff}}$.*

3. *The function $f$ is sufficiently smooth (i.e., it has Lipschitz-continuous gradients or can be approximated by its first-order Taylor expansion with bounded higher-order terms).*

*Recalling the definition of noisy hidden state $\tilde{\mathbf{h}}$ from Theorem B.1, then, the difference between the gradients computed on the clean and the noisy hidden states is $\mathcal{O}(\sigma_{\text{eff}})$. Moreover, as $\sigma_{\text{eff}} \to 0$, the gradients converge.*

*Proof.* We consider the loss function $f(\mathbf{h})$, which is differentiable with respect to the hidden state $\mathbf{h}$. We write $\nabla f(\mathbf{h})$ for the gradient of $f$ evaluated at $\mathbf{h}$. We introduce the noisy state $\tilde{\mathbf{h}} = \mathbf{h} + \sigma_{\text{eff}}\boldsymbol{\xi}$ and seek to understand how $\nabla f(\tilde{\mathbf{h}})$ differs from $\nabla f(\mathbf{h})$ for small $\sigma_{\text{eff}}$. For a small perturbation $\Delta \equiv \sigma_{\text{eff}}\, \boldsymbol{\xi}$ around $\mathbf{h}$, the output $f(\mathbf{h} + \Delta)$ can be approximated by the first-order Taylor expansion:
$$f(\mathbf{h} + \Delta) \;\approx\; f(\mathbf{h}) \;+\; \nabla f(\mathbf{h})^\top \Delta \;+\; \underbrace{R(\mathbf{h}, \Delta)}_{\text{higher-order remainder}}.$$

If $f$ is $\mathcal{C}^2$ (twice continuously differentiable) and/or has Lipschitz-continuous gradients, the remainder $R(\mathbf{h}, \Delta)$ is of order $\|\Delta\|^2$. Concretely,
$$R(\mathbf{h}, \Delta) \;=\; \mathcal{O}\big(\|\Delta\|^2\big) \;=\; \mathcal{O}\big(\sigma^2_{\text{eff}}\big),$$
since $\|\Delta\| = \mathcal{O}(\sigma_{\text{eff}})$. Given $\mathbb{E}[\boldsymbol{\xi}] = \mathbf{0}$ (unbiased noise) and $\text{Var}(\boldsymbol{\xi}) = \mathbf{I}$ (each component has unit variance), the perturbation $\Delta = \sigma_{\text{eff}}\boldsymbol{\xi}$ has zero mean and bounded second moment $\mathbb{E}\big[\|\Delta\|^2\big] = n\,\sigma^2_{\text{eff}}$. This ensures that: 1. $\mathbb{E}[\Delta] = \mathbf{0}$, 2. $\Delta$ is $\mathcal{O}(\sigma_{\text{eff}})$ in norm, on average or with high probability (e.g., by concentration inequalities). The gradient difference of interest is
$$\nabla f(\tilde{\mathbf{h}}) \;-\; \nabla f(\mathbf{h}) \;=\; \nabla f(\mathbf{h} + \Delta) \;-\; \nabla f(\mathbf{h}).$$
Under standard smoothness conditions (e.g., $f$ having an $L$-Lipschitz gradient), we have:
$$\big\|\nabla f(\mathbf{h} + \Delta) - \nabla f(\mathbf{h})\big\| \;\leq\; L \, \|\Delta\| \;=\; \mathcal{O}(\sigma_{\text{eff}}),$$
where $L$ is the Lipschitz constant of $\nabla f$. Equivalently, if one uses a second-order expansion of $f$, the difference in gradients can be bounded by the magnitude of $\Delta$. Either viewpoint shows that the discrepancy is controlled by $\sigma_{\text{eff}}$. Since the difference in gradients is at most proportional to $\|\Delta\| \sim \sigma_{\text{eff}}$, letting $\sigma_{\text{eff}} \to 0$ forces $\Delta \to \mathbf{0}$ and therefore
$$\lim_{\sigma_{\text{eff}} \to 0} \big\|\nabla f(\mathbf{h} + \Delta) - \nabla f(\mathbf{h})\big\| \;\leq\; \lim_{\sigma_{\text{eff}} \to 0} L \, \|\Delta\| \;=\; L \cdot 0 \;=\; 0.$$
Hence,
$$\lim_{\sigma_{\text{eff}} \to 0} \big\|\nabla f(\mathbf{h} + \Delta) - \nabla f(\mathbf{h})\big\| \;=\; 0,$$
Thus, for very small noise levels, the gradient computed on the noisy hidden state becomes arbitrarily close to the gradient computed on the clean hidden state. In other words, the gradients *converge*. We conclude that under the assumptions of unbiasedness, bounded noise variance, and smoothness of $f$, the difference between the gradients evaluated at the clean and noisy states is of order $\sigma_{\text{eff}}$. Hence, in the limit $\sigma_{\text{eff}} \to 0$, the gradient discrepancy vanishes. $\qquad\square$

### B.4 ROBUSTNESS OF CONSISTENCY LOSS

**Lemma B.5.** *Let $\mathbf{z}^{(1)}, \mathbf{z}^{(2)} \in \mathbb{R}^K$ be two logit vectors of dimension $K$. Define the consistency loss between their associated softmax outputs by*

$$\mathcal{L}_{consistency} = \text{KL}\Big(\text{softmax}(\mathbf{z}^{(1)}) \,\|\, \text{softmax}(\mathbf{z}^{(2)})\Big).$$

*Then $\mathcal{L}_{consistency}$ is minimized if and only if*

$$\text{softmax}(\mathbf{z}^{(1)}) = \text{softmax}(\mathbf{z}^{(2)}).$$

*Proof.* For a vector $\mathbf{z} = (z_1, \ldots, z_K) \in \mathbb{R}^K$, the softmax function is defined component-wise by

$$\text{softmax}(\mathbf{z})_i = \frac{e^{z_i}}{\sum_{j=1}^{K} e^{z_j}} \quad \text{for } i = 1, 2, \ldots, K.$$

This ensures that each component $\text{softmax}(\mathbf{z})_i$ is non-negative and $\sum_{i=1}^{K} \text{softmax}(\mathbf{z})_i = 1$. Therefore, $\text{softmax}(\mathbf{z})$ is a valid probability distribution over the $K$ outcomes. Consider two discrete probability distributions $\mathbf{p} = (p_1, \ldots, p_K)$ and $\mathbf{q} = (q_1, \ldots, q_K)$, both lying in the probability simplex ($p_i, q_i \geq 0$ and $\sum_i p_i = \sum_i q_i = 1$). The Kullback–Leibler (KL) divergence from $\mathbf{q}$ to $\mathbf{p}$ is defined as

$$\text{KL}(\mathbf{p} \,\|\, \mathbf{q}) = \sum_{i=1}^{K} p_i \log\Big(\frac{p_i}{q_i}\Big),$$

where by convention $0 \log(0/q) = 0$ and $p \log(p/0) = \infty$ for $p > 0$. A key property of KL divergence is its non-negativity:

$$\text{KL}(\mathbf{p} \,\|\, \mathbf{q}) \geq 0,$$

with equality if and only if $\mathbf{p} = \mathbf{q}$ (i.e., $p_i = q_i$ for each $i$). In our setup, we let

$$\mathbf{p} = \text{softmax}(\mathbf{z}^{(1)}) \quad \text{and} \quad \mathbf{q} = \text{softmax}(\mathbf{z}^{(2)}).$$

Then the consistency loss is exactly

$$\mathcal{L}_{\text{consistency}} = \text{KL}\Big(\text{softmax}(\mathbf{z}^{(1)}) \,\|\, \text{softmax}(\mathbf{z}^{(2)})\Big) = \sum_{i=1}^{K} \text{softmax}(\mathbf{z}^{(1)})_i \, \log\Big(\frac{\text{softmax}(\mathbf{z}^{(1)})_i}{\text{softmax}(\mathbf{z}^{(2)})_i}\Big).$$

From the fundamental property of KL divergence, we know that

$$\text{KL}(\mathbf{p} \,\|\, \mathbf{q}) \geq 0, \quad \text{with equality if and only if } \mathbf{p} = \mathbf{q}.$$

Translating this to our softmax distributions, we get

$$\text{KL}\Big(\text{softmax}(\mathbf{z}^{(1)}), \, \text{softmax}(\mathbf{z}^{(2)})\Big) \geq 0,$$

and it is equal to 0 precisely when

$$\text{softmax}(\mathbf{z}^{(1)}) = \text{softmax}(\mathbf{z}^{(2)}).$$

Hence, the consistency loss $\mathcal{L}_{\text{consistency}}$ achieves its global minimum of 0 if and only if

$$\text{softmax}(\mathbf{z}^{(1)}) = \text{softmax}(\mathbf{z}^{(2)}),$$

as required. This completes the proof. $\qquad\square$

### B.5 BOUND ON THE FINAL LOSS DUE TO NOISE

We now derive a simple upper bound showing how the presence of adaptive noise injection affects the final training loss. Consider the final loss $\mathcal{L}_{\text{final}}$ in Equation equation 19, which we write abstractly as a function of the model parameters $\Theta$:

$$\mathcal{L}_{\text{final}}(\Theta) = \underbrace{\lambda_{\text{ce}} \, \mathcal{L}_{\text{ce}}(\Theta) + (1 - \lambda_{\text{ce}}) \, \mathcal{L}_{\text{soft}}(\Theta)}_{\mathcal{L}_{\text{hybrid}}(\Theta)} + \lambda_{\text{consistency}} \, \mathcal{L}_{\text{consistency}}(\Theta).$$

Because each term in $\mathcal{L}_{\text{final}}$ (cross-entropy, KL divergence, etc.) is $\beta$-smooth (Nesterov, 2005) with respect to the logits, and the logits themselves are Lipschitz continuous with respect to the hidden states $\mathbf{h}$ (assuming bounded weight matrices), we can show that random perturbations in $\mathbf{h}$ of size $\|\Delta\|$ shift the loss by at most $O(\|\Delta\|)$.

**Theorem B.2.** *Let $\Delta = \tilde{\mathbf{h}} - \mathbf{h}$ be the per-token perturbation introduced by noise injection. Suppose $\Delta$ has zero mean and bounded second moment such that*

$$\mathbb{E}[\|\Delta\|^2] \;\leq\; \sigma_{\max}^2.$$

*Then, for sufficiently smooth loss components, the expected deviation in $\mathcal{L}_{final}$ satisfies*

$$\left| \mathbb{E}\left[\mathcal{L}_{final}(\Theta + \Delta)\right] \;-\; \mathcal{L}_{final}(\Theta) \right| \;\leq\; C\,\sigma_{\max}^2,$$

*for some constant $C > 0$ depending on the network's Lipschitz constants and the smoothness parameters of the loss.*

*Proof.* For simplicity of notation, write $\mathcal{L}_{\text{final}}(\Theta)$ as a function that ultimately depends on the hidden states $\mathbf{h}$. In typical neural network architectures, the hidden states $\mathbf{h}$ themselves depend on subsets of $\Theta$ (e.g., weights and biases), so a perturbation $\Delta$ to $\mathbf{h}$ can be seen as an effective perturbation to the logits or subsequent layers. Formally, let $\mathbf{z}(\mathbf{h}, \Theta)$ denote the logits. Then $\mathcal{L}_{\text{final}}(\Theta)$ depends on $\mathbf{z}(\mathbf{h}, \Theta)$. We assume that $\mathbf{z}(\mathbf{h}, \Theta)$ is $L$-Lipschitz in $\mathbf{h}$. That is, there exists some constant $L > 0$ such that

$$\left\| \mathbf{z}(\mathbf{h}_1, \Theta) - \mathbf{z}(\mathbf{h}_2, \Theta) \right\| \;\leq\; L\,\|\mathbf{h}_1 - \mathbf{h}_2\|,$$

for any $\mathbf{h}_1, \mathbf{h}_2$. This typically follows from bounding the norm of weight matrices and using standard results on Lipschitz continuity of affine and activation transformations. Each term in $\mathcal{L}_{\text{final}}$ (e.g., cross-entropy, soft loss, or KL term) is $\beta$-smooth with respect to its input logits $\mathbf{z}$. Concretely, this means the gradient of $\mathcal{L}_{\text{final}}$ with respect to $\mathbf{z}$ is $\beta$-Lipschitz. Equivalently, the second derivative (or Hessian) of $\mathcal{L}_{\text{final}}$ w.r.t. $\mathbf{z}$ is bounded by $\beta$ in norm:

$$\left\| \nabla_{\mathbf{z}}^2 \mathcal{L}_{\text{final}}(\mathbf{z}) \right\| \;\leq\; \beta.$$

Therefore, under small perturbations to $\mathbf{z}$, the change in $\mathcal{L}_{\text{final}}$ is $\mathcal{O}(\|\Delta_{\mathbf{z}}\|^2)$, where $\Delta_{\mathbf{z}}$ is the corresponding change in logits. Given $\Delta = \tilde{\mathbf{h}} - \mathbf{h}$, the corresponding change in the logits is approximately

$$\Delta_{\mathbf{z}} \;\approx\; \mathbf{z}(\mathbf{h} + \Delta, \Theta) \;-\; \mathbf{z}(\mathbf{h}, \Theta).$$

By Lipschitz continuity in $\mathbf{h}$, we have

$$\left\| \Delta_{\mathbf{z}} \right\| \;\leq\; L\,\|\Delta\|.$$

Then, if $\|\Delta\|$ is small, we can write a first-order Taylor expansion for $\mathcal{L}_{\text{final}}$ around the unperturbed logits $\mathbf{z}(\mathbf{h}, \Theta)$, yielding an extra second-order remainder term on the order of $\|\Delta_{\mathbf{z}}\|^2$. Let $\mathcal{L}_{\text{final}}(\mathbf{z})$ denote the final loss viewed as a function of $\mathbf{z}$. Under a small change $\Delta_{\mathbf{z}}$, we have

$$\mathcal{L}_{\text{final}}\left(\mathbf{z} + \Delta_{\mathbf{z}}\right) \;=\; \mathcal{L}_{\text{final}}(\mathbf{z}) \;+\; \nabla_{\mathbf{z}}\mathcal{L}_{\text{final}}(\mathbf{z})^\top \Delta_{\mathbf{z}} \;+\; \underbrace{R\left(\mathbf{z}, \Delta_{\mathbf{z}}\right)}_{\text{second-order term}}.$$

With $\beta$-smoothness, $R(\mathbf{z}, \Delta_{\mathbf{z}}) = \mathcal{O}\left(\|\Delta_{\mathbf{z}}\|^2\right)$. Since the noise $\Delta$ has $\mathbb{E}[\Delta] = \mathbf{0}$ and $\mathbb{E}[\|\Delta\|^2] \leq \sigma_{\max}^2$, we focus on bounding the expected magnitude of the remainder term. By combining Lipschitz continuity of the logits with $\beta$-smoothness of $\mathcal{L}_{\text{final}}$, one obtains:

$$\left| \mathbb{E}\left[\mathcal{L}_{\text{final}}\left(\mathbf{z} + \Delta_{\mathbf{z}}\right)\right] \;-\; \mathcal{L}_{\text{final}}(\mathbf{z}) \right| \;\leq\; \mathbb{E}\left[\left| R\left(\mathbf{z}, \Delta_{\mathbf{z}}\right)\right|\right] \;=\; \mathcal{O}\left(\mathbb{E}\left[\|\Delta_{\mathbf{z}}\|^2\right]\right).$$

Since $\|\Delta_{\mathbf{z}}\| \leq L\,\|\Delta\|$, we have $\|\Delta_{\mathbf{z}}\|^2 \leq L^2\|\Delta\|^2$. Taking expectations,

$$\mathbb{E}\left[\|\Delta_{\mathbf{z}}\|^2\right] \;\leq\; L^2\,\mathbb{E}\left[\|\Delta\|^2\right] \;\leq\; L^2\,\sigma_{\max}^2.$$

Hence the overall change is

$$\left| \mathbb{E}\left[\mathcal{L}_{\text{final}}\left(\mathbf{z} + \Delta_{\mathbf{z}}\right)\right] \;-\; \mathcal{L}_{\text{final}}(\mathbf{z}) \right| \;\leq\; C\,\sigma_{\max}^2,$$

where $C$ encapsulates constants like $L^2$, $\beta$, and possibly other network-dependent factors. This shows that the expected difference in $\mathcal{L}_{\text{final}}$ under perturbation $\Delta$ with bounded second moment $\sigma_{\max}^2$ remains upper-bounded by a term proportional to $\sigma_{\max}^2$. Thus, moderate noise levels do not drastically increase the final loss, aligning with empirical observations that adaptive noise injection remains stable in training. $\square$

## B.6 CONVERGENCE IN EXPECTATION

Next, we show that under standard assumptions, NoiseFiT converges in expectation to a local minimum even in the presence of adaptive noise.

**Theorem B.3.** *Let $\Theta^* \in \mathbb{R}^d$ be a local minimum of the final loss $\mathcal{L}_{final}(\Theta)$. Suppose the following conditions hold:*

*(a)* $\mathcal{L}_{final}(\Theta)$ *is continuously differentiable on $\mathbb{R}^d$ and bounded below, i.e., $\inf_\Theta \mathcal{L}_{final}(\Theta) > -\infty$.*

*(b) The gradient $\nabla \mathcal{L}_{final}(\Theta)$ is $L$-Lipschitz continuous with respect to $\Theta$. Formally,*

$$\|\nabla \mathcal{L}_{final}(\Theta_1) - \nabla \mathcal{L}_{final}(\Theta_2)\| \leq L \|\Theta_1 - \Theta_2\|,$$

*for all $\Theta_1, \Theta_2 \in \mathbb{R}^d$.*

*(c) The adaptive noise injection yields* unbiased *hidden states in expectation (Theorem B.1) with bounded variance. Concretely, at each iteration $t$, the hidden-state noise $\Delta_t$ satisfies $\mathbb{E}[\Delta_t] = \mathbf{0}$ and $\mathbb{E}[\|\Delta_t\|^2] \leq \sigma_{\max}^2$ for some $\sigma_{\max} > 0$.*

*Then, performing stochastic gradient descent (or any standard first-order optimizer) on the noised hidden states converges to $\Theta^*$ in expectation. In other words, if $\Theta_t$ denotes the parameters at iteration $t$,*

$$\lim_{t \to \infty} \mathbb{E}\big[\|\nabla \mathcal{L}_{final}(\Theta_t)\|\big] = 0.$$

*Proof.* We outline the main ideas, referencing standard results from stochastic optimization (Bottou et al., 2018). At iteration $t$, let $\mathbf{h}_t$ be the hidden states (a function of $\Theta_t$) and let $\tilde{\mathbf{h}}_t = \mathbf{h}_t + \Delta_t$ be the noised hidden states, where $\Delta_t$ is the adaptive noise added at iteration $t$. By assumption (c), we have

$$\mathbb{E}[\Delta_t] = \mathbf{0}, \quad \mathbb{E}[\|\Delta_t\|^2] \leq \sigma_{\max}^2.$$

The gradient of $\mathcal{L}_{\text{final}}$ with respect to $\Theta$ can be approximated by backpropagation through $\tilde{\mathbf{h}}_t$, leading to an update of the form:

$$\Theta_{t+1} = \Theta_t - \alpha_t \widehat{\nabla \mathcal{L}_{\text{final}}}(\Theta_t, \tilde{\mathbf{h}}_t),$$

where $\alpha_t$ is the step size at iteration $t$. Because the noise injection is unbiased in expectation (Theorem B.1), the difference between $\tilde{\mathbf{h}}_t$ and $\mathbf{h}_t$ introduces no systematic bias into the gradient. Effectively, $\widehat{\nabla \mathcal{L}_{\text{final}}}(\Theta_t, \tilde{\mathbf{h}}_t)$ can be seen as a *stochastic* gradient estimator of $\nabla \mathcal{L}_{\text{final}}(\Theta_t)$. While it may have increased variance due to noise, the expectation of this estimator still aligns with the true gradient (up to standard stochastic sampling noise). Formally, one can write:

$$\mathbb{E}\Big[\widehat{\nabla \mathcal{L}_{\text{final}}}(\Theta_t, \tilde{\mathbf{h}}_t) \mid \Theta_t\Big] = \nabla \mathcal{L}_{\text{final}}(\Theta_t),$$

provided the only randomness comes from $\Delta_t$ (and possibly mini-batch subsampling), both of which are classical scenarios in stochastic gradient methods. Under assumption (c), the second moment of $\Delta_t$ is bounded by $\sigma_{\max}^2$, which implies that the gradient estimator has bounded variance. Specifically, one can show:

$$\mathbb{E}\Big[\big\|\widehat{\nabla \mathcal{L}_{\text{final}}}(\Theta_t, \tilde{\mathbf{h}}_t) - \nabla \mathcal{L}_{\text{final}}(\Theta_t)\big\|^2 \mid \Theta_t\Big] \leq \sigma_g^2,$$

for some constant $\sigma_g^2 > 0$ that depends on $\sigma_{\max}^2$ and network/Lipschitz constants (see also the discussion in Section B.5 for how noise affects loss gradients). The convergence in expectation for stochastic gradient-type methods requires:

- $\mathcal{L}_{\text{final}}$ is lower-bounded and differentiable,

- $\nabla \mathcal{L}_{\text{final}}(\Theta)$ is $L$-Lipschitz,

- The gradient estimator is unbiased with bounded variance,

- A suitable step-size ($\alpha_t$) decay schedule, such as $\alpha_t = \frac{1}{\sqrt{t}}$ or $\alpha_t = \frac{1}{t}$.

Under these conditions, classical results in stochastic optimization (Bottou et al., 2018)) guarantee that $\nabla \mathcal{L}_{\text{final}}(\Theta_t)$ converges to $\mathbf{0}$ in expectation, which implies $\Theta_t$ converges to a stationary (or local minimum) point of $\mathcal{L}_{\text{final}}$. Finally, assumption (c) and our earlier result that the gradient difference induced by noise is $\mathcal{O}(\|\Delta_t\|)$ (see Lemma B.4) implies

$$\left\| \nabla \mathcal{L}_{\text{final}}(\Theta_t, \tilde{\mathbf{h}}_t) - \nabla \mathcal{L}_{\text{final}}(\Theta_t, \mathbf{h}_t) \right\| = \mathcal{O}(\|\Delta_t\|),$$

and since $\mathbb{E}[\|\Delta_t\|^2] \le \sigma_{\max}^2$, this does not disrupt the overall convergence analysis. The difference diminishes for small noise and remains bounded for moderate noise, preserving the standard stochastic gradient convergence arguments. Hence, all the conditions of a standard convergence theorem for stochastic gradient methods are satisfied: (1) $\mathcal{L}_{\text{final}}$ is smooth and bounded below, (2) its gradient is Lipschitz, (3) the noised gradient is unbiased with bounded variance, (4) the step size can be chosen to decay appropriately. Therefore, the standard result

$$\lim_{t \to \infty} \mathbb{E}\left[ \|\nabla \mathcal{L}_{\text{final}}(\Theta_t)\| \right] = 0$$

holds, indicating convergence in expectation to a local minimum (or stationary point) $\Theta^*$. This completes the proof. $\qquad\square$

## C  PRACTICAL RECIPE FOR NOISEFIT

To make *NoiseFiT* usable without extensive sweeps, we distill the strongest settings that consistently reproduced the trends in Table 1.

1. **Warm-up SNR bootstrap:** run a short pilot forward passes with uniform, small noise to estimate per-layer SNR curves from clean vs. noisy passes; rank layers by stability (high SNR) / instability (low SNR) and then freeze the $k$ target layers for injection.

2. for large models, start with $k=3$ *lowest*-SNR layers; for small models, start with $k=3$ *highest*-SNR layers. If the validation hallucination rate plateaus, adjust $k$ by $\pm 3$.

3. use $\mathrm{STD} = 0.01$ as a safe default; consider $0.1$ on larger models if validation remains stable. Apply a short linear ramp-up over the first 10–20% of steps.

**Runtime overhead.** NoiseFiT adds negligible training-time cost beyond a second (noisy) forward pass. On V100 GPUs we observed slightly higher memory but similar or lower utilization/power than BASEFIT (Table F.1), which is acceptable for multi-GPU fine-tuning.

## D  EXTENDED EVALUATION FOR MISTRAL-7B

To assess how our method scales to a relatively large-parameter model (**Mistral-7B**), we conduct *extended ablations* over (i) the # layers to which we apply noise injection and (ii) the magnitude of the injected noise (**STD**), alongside two SNR settings (*Highest/Lowest*). This study complements the cross-model results as shown in Table 1. Table D.1 shows that selective noise injection (3–12 layers) generally outperforms injecting into *all* layers, suggesting that broader perturbation is not always beneficial at larger scale.

## E  TEST PERFORMANCE ANALYSIS

This section presents detailed experimental results for NoiseFiT in mitigating hallucinations of LLMs based on the test dataset. The evaluated models include Llama-3.2-1B, Llama-3.2-3B, Gemma-3-1B, Qwen2.5-0.5B, and Mistral-7B-v0.1. For each model, performance is assessed across 17 distinct categories of prompts, encompassing a total of 208 prompts, under multiple configurations: the base model, the base model with fine-tuning (denoted BaseFiT), and several noise-injected variants using NoiseFiT. These NoiseFiT configurations vary by the number of layers affected, the standard deviation (STD) of the injected noise (e.g., 0.001, 0.01, 0.1), and the signal-to-noise ratio (SNR), where 'L' denotes the lowest SNR (highest noise relative to signal were selected for noise injection) and 'H' denotes the highest SNR (lowest noise relative to signal were selected for noise injection). Performance metrics are averaged across five runs per prompt to ensure statistical reliability (online supplementary material).

Table D.1: Extended leaderboard benchmark and hallucination evaluation results for Mistral-7B model across different #Layers, STD, and SNR

| Model | #Layers | STD | SNR | MMLU-Pro | BBH | GPQA | Math | IFEval | MUSR | TfQA-MC | HaluEval |
|---|---|---|---|---|---|---|---|---|---|---|---|
| | 6 | 0.1 | Highest | **30.28** | 44.63 | 29.46 | 2.69 | 13.31 | 39.51 | 38.20 | 51.40 |
| | 12 | 0.001 | Highest | 30.14 | 44.32 | 29.01 | 2.49 | 13.68 | 39.37 | 36.64 | 47.13 |
| | All | 0.01 | N/A | 30.14 | 45.12 | 29.17 | 2.70 | 13.86 | 40.98 | 35.12 | 46.96 |
| | BaseFiT | N/A | N/A | 30.01 | 44.34 | 29.29 | 2.97 | 11.46 | 38.84 | 37.62 | 47.60 |
| | 12 | 0.1 | Lowest | 30.01 | 43.59 | 28.86 | 2.52 | 11.65 | **41.49** | 40.63 | 51.46 |
| | 12 | 0.01 | Lowest | 30.00 | 43.74 | 28.83 | 3.05 | 13.68 | 39.50 | 38.44 | 49.37 |
| Mistral7B-v0.1 | 3 | 0.1 | Lowest | 29.97 | **45.84** | 28.07 | 3.01 | 13.49 | 39.89 | 39.41 | 50.72 |
| | 3 | 0.01 | Lowest | 29.95 | 45.47 | 25.16 | 3.35 | 13.68 | 39.48 | 36.24 | 46.88 |
| | 3 | 0.01 | Highest | 29.95 | 45.56 | 26.09 | 2.63 | 10.91 | 39.11 | 37.02 | 47.13 |
| | 12 | 0.01 | Highest | 29.93 | 44.05 | 28.86 | 3.30 | 14.05 | 39.77 | 37.14 | 47.39 |
| | 12 | 0.1 | Highest | 29.93 | 44.37 | 28.90 | 2.71 | 13.31 | 40.30 | 39.84 | 50.44 |
| | 3 | 0.1 | Highest | 29.75 | 44.62 | 28.36 | 3.10 | 10.17 | 37.65 | 38.54 | 49.71 |
| | 6 | 0.01 | Lowest | 29.74 | 45.08 | 29.66 | **3.43** | 11.83 | 40.18 | **41.24** | **52.34** |
| | 6 | 0.01 | Highest | 29.72 | 44.67 | 28.74 | 2.78 | 13.68 | 38.97 | 37.05 | 47.17 |
| | 6 | 0.1 | Lowest | 29.67 | 44.53 | 30.04 | 3.19 | 12.20 | 38.71 | 37.72 | 47.85 |
| | 6 | 0.001 | Lowest | 29.59 | 45.61 | 30.18 | 2.26 | 12.75 | 39.64 | 35.43 | 47.12 |
| | All | 0.001 | N/A | 29.57 | 44.51 | 28.74 | 2.26 | 11.65 | 38.58 | 34.71 | 46.58 |
| | 3 | 0.3 | Highest | 29.56 | 44.65 | 29.67 | 3.22 | 12.38 | 39.38 | 37.46 | 48.21 |
| | 3 | 0.3 | Lowest | 29.53 | 44.53 | **30.54** | 2.75 | 10.91 | 39.51 | 37.26 | 49.40 |
| | 12 | 0.001 | Lowest | 29.24 | 45.16 | 28.48 | 1.69 | **14.42** | 41.23 | 38.33 | 50.19 |
| | All | 0.1 | N/A | 29.00 | 44.81 | 29.69 | 2.45 | 11.65 | 40.31 | 34.35 | 49.91 |

**Prompt Formatting and Generation:** To generate model responses, we formatted each user prompt with specific delimiters (`<|im_start|>user ... <|im_end|>`) followed by the assistant token. We used the generation configuration demonstrated in Table E.1.

Table E.1: Generation configuration hyperparameters.

| | Max. New Tokens | Temperature | Top-$p$ | Top-$k$ | Rep. Penalty |
|---|---|---|---|---|---|
| **Value** | 50 | 0.5 | 0.9 | 40 | 1.2 |

This setup allowed us to obtain diverse responses while mitigating overly repetitive outputs. Each local process repeated the generation step for five rounds, independently producing slightly varied outputs for each prompt.

The tables in this appendix (Tables E.2 to E.6) provide category-wise performance scores alongside overall performance metrics for each model and configuration. This enables a comprehensive evaluation of how NoiseFiT mitigates hallucinations across different tasks and setups.

### E.1 ANALYSIS OF RESULTS

The results in this appendix highlight the effectiveness of NoiseFiT in mitigating hallucinations in LLMs, demonstrating both general trends across models and specific insights tailored to this task. Below, we analyze these findings, with an in-depth focus on the Mistral-7B-v0.1 model due to its comprehensive set of noise injection configurations.

**General Performance Trends Across Models:** Fine-tuning the base models (BaseFiT) generally improves performance over the untrained base models, serving as a foundational step in reducing hallucinations by better aligning the model with the training data. For Llama-3.2-1B, overall performance increases from 48.6% to 54.0%; for Llama-3.2-3B, from 60.0% to 66.4%; for Qwen2.5-0.5B, from 26.4% to 28.8%; and for Mistral-7B-v0.1, from 70.6% to 77.2%. However, Gemma-3-1B shows a decline from 50.6% to 47.6% with BaseFiT, suggesting that standard fine-tuning alone may not always mitigate hallucinations effectively and could even exacerbate them in some cases.

NoiseFiT, designed specifically to tackle hallucinations, frequently enhances performance beyond BaseFiT, particularly in categories prone to factual inaccuracies. For Llama-3.2-3B, the best NoiseFiT variant (3 layers, STD 0.01, highest SNR) achieves 70.2%, surpassing BaseFiT's 66.4%. Qwen2.5-0.5B improves significantly from 28.8% (BaseFiT) to 36.6% (3 layers, STD 0.1, highest SNR). In

Gemma-3-1B, NoiseFiT recovers performance to 54.6% (3 layers, STD 0.1, highest SNR) from BaseFiT's 47.6%, exceeding the base model's 50.6%. These improvements indicate that NoiseFiT's noise injection enhances the model's ability to generalize, reducing its tendency to hallucinate by regularizing its learned representations.

**Mistral-7B-v0.1:**   The Mistral-7B-v0.1 model, with its 7 billion parameters, provides a robust case study for evaluating NoiseFiT's impact on hallucination mitigation, as it was tested with noise applied to 3 layers, 12 layers, or all layers, across various STDs and SNRs. The key findings are as follows:

- **Optimal Number of Layers for Noise Injection:** Injecting noise into fewer layers (specifically 3 layers) consistently outperforms configurations with noise applied to 12 layers or all layers in mitigating hallucinations. The highest overall performance of 78.4% is achieved with 3 layers, STD 0.1, and lowest SNR, compared to 77.2% for BaseFiT. In contrast, the best 12-layer configuration (STD 0.1, lowest SNR) yields 76.4%, and the best all-layer configuration (STD 0.1) also reaches 76.4%. This suggests that targeting a small, critical subset of layers with noise injection enhances the model's ability to distinguish correct from incorrect information, effectively reducing hallucinations, while broader noise application may disrupt learned hidden states excessively.

- **Impact of Noise Level and SNR:** Within the 3-layer configurations, higher noise levels (STD 0.1) paired with lower SNR (more noise relative to signal) outperform lower noise levels or higher SNR settings. For example, 3 layers with STD 0.1 and lowest SNR achieves 78.4%, while the same STD with highest SNR yields 77.4%, and STD 0.01 with lowest and highest SNR scores 76.5% and 77.3%, respectively. This indicates that substantial noise, when carefully applied, acts as a strong regularizer in Mistral-7B-v0.1, reducing overconfidence in incorrect outputs and thus mitigating hallucinations. The preference for lower SNR underscores the benefit of higher noise intensity in this context.

- **Category-wise Performance Variations:** NoiseFiT significantly improves performance in categories where hallucinations are particularly prevalent. In "Medical (Disease Causes)," performance reaches 100.0% across multiple 3-layer configurations (e.g., STD 0.1, lowest SNR), up from 93.3% in BaseFiT. "Scientific Discoveries" improves from 81.2% to 88.2% (3 layers, STD 0.01, lowest SNR), "Who Invented" from 82.1% to 85.2% (3 layers, STD 0.1, highest SNR), and "Sports (Famous Players)" from 74.7% to 92.0% (3 layers, STD 0.01, lowest SNR). These gains highlight NoiseFiT's effectiveness in enhancing factual accuracy and reducing hallucinations in knowledge-intensive tasks. However, categories like "Animals" (BaseFiT: 34.1%, best NoiseFiT: 43.6% with 12 layers, STD 0.001, highest SNR, still below base's 63.5%) and "Art (Painting Subjects)" (BaseFiT: 32.2%, best NoiseFiT: 40.0% with 12 layers, STD 0.1, highest or lowest SNR, below base's 42.2%) show persistent challenges, indicating that NoiseFiT may not fully mitigate hallucinations in tasks requiring nuanced or context-sensitive understanding.

- **Comparison with Other Models:** Unlike smaller models like Llama-3.2-1B (best: STD 0.1, highest SNR, 55.8%) or Qwen2.5-0.5B (best: STD 0.1, highest SNR, 36.6%), where higher SNR (less noise) often performs better, Mistral-7B-v0.1 favors lower SNR (more noise) in its optimal configuration. This difference likely reflects Mistral's larger capacity, allowing it to benefit from higher noise levels as a stronger regularizer against hallucinations, whereas smaller models may be more sensitive to noise, requiring lower levels to maintain stability.

- **Layer Selection Implications:** The superior performance of the 3-layer configuration suggests an optimal subset exists—possibly layers critical with high variance. Broader noise application (12 layers or all layers) reduces effectiveness (e.g., 12 layers, STD 0.001, lowest SNR: 77.0%; all layers, STD 0.001: 74.4%), emphasizing the importance of layer selection strategy for noise injection in mitigating hallucination.

These findings demonstrate that for Mistral-7B-v0.1, injecting significant noise (STD 0.1, lowest SNR) into a small, targeted set of layers (3 layers) optimizes performance, slightly surpassing BaseFiT and outperforming broader noise applications in reducing hallucinations. The category-wise analysis reveals substantial benefits in factual, knowledge-based tasks, though challenges persist in areas like "Animals" and "Art," suggesting limitations in NoiseFiT's applicability across those domains for our specific test dataset.

In conclusion, NoiseFiT proves to be a promising technique for mitigating hallucinations in LLMs, particularly in knowledge-intensive categories, by leveraging noise injection to enhance robustness and reduce overconfidence in incorrect outputs. However, its effectiveness varies across tasks and

models, necessitating task-specific optimization of noise injection parameters and further research to address remaining challenges in certain domains.

Table E.2: Category-wise performance of Llama-3.2-1B configurations. Performance is averaged across 5 runs per prompt (208 prompts total). For noise-injected cases (3 layers), the first value is the standard deviation (STD), with 'L' indicating Lowest SNR and 'H' indicating Highest SNR.

| Category | **Llama-3.2-1B** (3 layers if noise injected) | | | | | |
|---|---|---|---|---|---|---|
| | Base | BaseFiT | 0.1, L | **0.1, H** | 0.01, L | 0.01, H |
| Medical (Disease Causes) | 76.6 | 50.0 | 53.4 | 60.0 | 56.6 | 63.4 |
| Geography – Landmarks | 65.4 | 69.0 | 81.8 | 85.4 | 80.0 | 81.8 |
| Geography – Capitals | 66.6 | 75.0 | 50.0 | 75.0 | 98.4 | 80.0 |
| Geography – Currency | 34.6 | 66.6 | 65.4 | 74.6 | 77.4 | 74.6 |
| Geography – Landmark Locations | 53.4 | 100.0 | 100.0 | 91.6 | 91.6 | 100.0 |
| Language | 80.0 | 100.0 | 100.0 | 100.0 | 100.0 | 100.0 |
| History (Year Events) | 54.6 | 91.0 | 100.0 | 100.0 | 87.2 | 85.4 |
| History (When Events) | 48.4 | 98.4 | 100.0 | 96.6 | 96.6 | 93.4 |
| Inventions | 40.0 | 25.0 | 41.2 | 40.0 | 38.8 | 22.6 |
| Animals | 14.2 | 34.2 | 29.4 | 36.4 | 22.4 | 24.8 |
| Music/Composers | 43.4 | 53.4 | 46.6 | 50.0 | 50.0 | 60.0 |
| Scientific Discoveries | 57.6 | 36.4 | 36.4 | 41.2 | 35.2 | 29.4 |
| Who Invented | 45.2 | 32.6 | 42.2 | 33.6 | 35.8 | 37.8 |
| Sports (Famous Players) | 74.6 | 30.6 | 26.6 | 29.4 | 30.6 | 32.0 |
| Art (Painting Subjects) | 22.2 | 13.4 | 12.2 | 10.0 | 13.4 | 12.2 |
| Literature | 60.0 | 70.6 | 60.0 | 64.2 | 67.4 | 53.6 |
| Miscellaneous | 80.0 | 100.0 | 100.0 | 100.0 | 100.0 | 100.0 |
| **Overall** | 48.6 | 54.0 | 53.4 | **55.8** | 55.4 | 52.4 |

Table E.3: Category-wise performance of Llama-3.2-3B configurations. Performance is averaged across 208 prompts total. For noise-injected cases (3 layers), the first value is the standard deviation (STD), with 'L' indicating Lowest SNR and 'H' indicating Highest SNR.

| Category | **Llama-3.2-3B** (3 layers if noise injected) | | | | | |
|---|---|---|---|---|---|---|
| | Base | BaseFiT | 0.1, L | 0.1, H | 0.01, L | **0.01, H** |
| Medical (Disease Causes) | 73.4 | 63.4 | 80.0 | 63.4 | 70.0 | 80.0 |
| Geography – Landmarks | 92.8 | 100.0 | 98.4 | 94.6 | 96.7 | 85.4 |
| Geography – Capitals | 78.3 | 91.6 | 85.0 | 91.6 | 91.6 | 91.7 |
| Geography – Currency | 84.0 | 100.0 | 98.6 | 100.0 | 98.7 | 100.0 |
| Geography – Landmark Locations | 86.7 | 96.6 | 100.0 | 100.0 | 100.0 | 100.0 |
| Language | 80.0 | 100.0 | 100.0 | 100.0 | 100.0 | 100.0 |
| History (Year Events) | 85.5 | 90.9 | 63.6 | 81.8 | 89.1 | 96.4 |
| History (When Events) | 70.0 | 91.6 | 91.6 | 91.6 | 91.6 | 98.3 |
| Inventions | 33.8 | 48.8 | 47.6 | 50.0 | 43.8 | 37.5 |
| Animals | 27.1 | 22.4 | 11.8 | 15.2 | 16.5 | 32.9 |
| Music/Composers | 76.7 | 60.0 | 66.6 | 50.0 | 63.3 | 63.3 |
| Scientific Discoveries | 49.4 | 48.2 | 49.4 | 58.8 | 52.9 | 55.3 |
| Who Invented | 51.6 | 66.4 | 75.8 | 83.2 | 82.1 | 74.7 |
| Sports (Famous Players) | 10.7 | 49.4 | 54.6 | 57.4 | 53.3 | 60.0 |
| Art (Painting Subjects) | 45.6 | 25.6 | 16.6 | 21.2 | 16.7 | 20.0 |
| Literature | 83.2 | 77.8 | 81.0 | 84.2 | 83.2 | 93.7 |
| Miscellaneous | 100.0 | 100.0 | 100.0 | 100.0 | 100.0 | 100.0 |
| **Overall** | 60.0 | 66.4 | 65.6 | 68.2 | 68.0 | **70.2** |

Table E.4: Category-wise performance of Gemma-3-1B configurations. Performance is averaged across 208 prompts total. For noise-injected cases (3 layers), the first value is the standard deviation (STD), with 'L' indicating Lowest SNR and 'H' indicating Highest SNR.

| Category | **Gemma-3-1B-it** (3 layers if noise injected) | | | | | |
| --- | --- | --- | --- | --- | --- | --- |
| | Base | BaseFiT | **0.1, H** | 0.01, H | 0.001, L | 0.01, L |
| Medical (disease causes) | 43.4 | 76.7 | 60.0 | 70.0 | 66.6 | 70.0 |
| Miscellaneous | 80.0 | 100.0 | 100.0 | 20.0 | 100.0 | 80.0 |
| Geography – Landmarks | 43.6 | 85.5 | 92.8 | 80.0 | 83.6 | 76.4 |
| Geography – Capitals | 78.4 | 100.0 | 100.0 | 100.0 | 100.0 | 100.0 |
| Geography – Currency | 73.4 | 93.3 | 86.6 | 97.4 | 96.0 | 92.0 |
| Language | 80.0 | 100.0 | 100.0 | 100.0 | 100.0 | 100.0 |
| History (Year events) | 83.6 | 81.8 | 81.8 | 81.8 | 91.0 | 80.0 |
| History (When events) | 76.6 | 41.7 | 76.6 | 81.6 | 60.0 | 40.0 |
| Inventions | 55.0 | 32.5 | 40.0 | 37.6 | 31.2 | 27.6 |
| Geography – Landmark Locations | 70.0 | 66.7 | 100.0 | 100.0 | 100.0 | 20.0 |
| Animals | 31.8 | 7.1 | 21.2 | 10.6 | 9.4 | 9.4 |
| Music/Composers | 26.6 | 23.3 | 26.6 | 26.6 | 40.0 | 33.4 |
| Scientific Discoveries | 42.4 | 36.5 | 29.4 | 27.0 | 24.8 | 24.8 |
| Who Invented | 65.2 | 45.3 | 63.2 | 57.8 | 50.6 | 53.6 |
| Sports (Famous Players) | 26.6 | 30.7 | 32.0 | 28.0 | 26.6 | 30.6 |
| Art (Painting Subjects) | 7.8 | 8.9 | 7.8 | 16.6 | 14.4 | 7.8 |
| Literature | 44.2 | 31.6 | 40.0 | 35.8 | 31.6 | 34.8 |
| **Overall** | 50.6 | 47.6 | **54.6** | 53.2 | 51.0 | 43.8 |

Table E.5: Category-wise performance of Qwen2.5-0.5B configurations. Performance is averaged across 208 prompts total. For noise-injected cases (3 layers), the first value is the standard deviation (STD), with 'L' indicating Lowest SNR and 'H' indicating Highest SNR.

| Category | **Qwen2.5-0.5B** (3 layers if noise injected) | | | | | |
| --- | --- | --- | --- | --- | --- | --- |
| | Base | BaseFiT | 0.1, L | **0.1, H** | 0.01, L | 0.01, H |
| Medical (disease causes) | 66.7 | 76.6 | 80.0 | 80.0 | 86.6 | 73.4 |
| Miscellaneous | 60.0 | 100.0 | 100.0 | 100.0 | 20.0 | 100.0 |
| Geography – Landmarks | 21.8 | 31.0 | 16.4 | 40.0 | 18.2 | 36.4 |
| Geography – Capitals | 31.7 | 75.0 | 85.0 | 86.6 | 76.6 | 71.6 |
| Geography – Currency | 38.7 | 50.6 | 74.6 | 77.4 | 69.4 | 78.6 |
| Language | 80.0 | 100.0 | 20.0 | 100.0 | 80.0 | 100.0 |
| History (Year events) | 43.6 | 45.4 | 63.6 | 58.2 | 51.0 | 61.8 |
| History (When events) | 38.3 | 68.4 | 73.4 | 73.4 | 61.6 | 66.6 |
| Inventions | 23.8 | 5.0 | 11.2 | 10.0 | 11.2 | 7.6 |
| Geography – Landmark Locations | 60.0 | 93.4 | 88.4 | 93.4 | 81.6 | 80.0 |
| Animals | 17.6 | 5.8 | 3.6 | 18.8 | 9.4 | 7.0 |
| Music/Composers | 0.0 | 0.0 | 0.0 | 0.0 | 0.0 | 0.0 |
| Scientific Discoveries | 28.2 | 11.8 | 11.8 | 13.0 | 7.0 | 14.2 |
| Who Invented | 21.0 | 10.6 | 11.6 | 7.4 | 11.6 | 8.4 |
| Sports (Famous Players) | 9.3 | 6.6 | 9.4 | 17.4 | 9.4 | 20.0 |
| Art (Painting Subjects) | 3.3 | 3.4 | 1.2 | 3.4 | 2.2 | 1.2 |
| Literature | 16.8 | 8.4 | 26.4 | 25.2 | 20.0 | 20.0 |
| **Overall** | 26.4 | 28.8 | 33.0 | **36.6** | 30.2 | 33.0 |

Table E.6: Category-wise performance of Mistral-7B-v0.1 configurations. Performance is typically averaged across 208 prompts. For noise-injected cases, column labels show #Layers, (STD, SNR); with 'L' indicating Lowest SNR and 'H' indicating Highest SNR.

| Category | Base | BaseFiT | 3 (0.1, L) | 3 (0.1, H) | 3 (0.01, L) | 3 (0.01, H) | l2 (0.001, L) | l2 (0.001, H) | l2 (0.01, L) | l2 (0.01, H) | l2 (0.1, L) | l2 (0.1, H) | All (0.001) | All (0.01) | All (0.1) |
|---|---|---|---|---|---|---|---|---|---|---|---|---|---|---|---|
| | | | | | | | Mistral-7B-v0.1 | | | | | | | | |
| Medical (Disease Causes) | 90.0 | 93.3 | 100.0 | 96.6 | 100.0 | 100.0 | 83.4 | 83.4 | 83.4 | 83.4 | 83.4 | 83.4 | 83.4 | 83.4 | 83.4 |
| Miscellaneous | 80.0 | 100.0 | 100.0 | 100.0 | 100.0 | 100.0 | 100.0 | 100.0 | 100.0 | 100.0 | 100.0 | 100.0 | 100.0 | 100.0 | 100.0 |
| Geography – Landmarks | 90.9 | 96.4 | 89.1 | 94.6 | 94.6 | 98.2 | 98.2 | 100.0 | 96.6 | 96.4 | 72.8 | 100.0 | 100.0 | 100.0 | 78.2 |
| Geography – Capitals | 93.3 | 100.0 | 100.0 | 100.0 | 100.0 | 100.0 | 100.0 | 100.0 | 100.0 | 100.0 | 100.0 | 100.0 | 100.0 | 100.0 | 100.0 |
| Geography – Currency | 81.3 | 100.0 | 100.0 | 100.0 | 98.6 | 100.0 | 100.0 | 100.0 | 100.0 | 100.0 | 100.0 | 100.0 | 100.0 | 100.0 | 100.0 |
| Language | 80.0 | 100.0 | 100.0 | 100.0 | 100.0 | 100.0 | 100.0 | 100.0 | 100.0 | 100.0 | 100.0 | 100.0 | 100.0 | 100.0 | 100.0 |
| History (Year Events) | 89.1 | 96.4 | 90.9 | 91.0 | 91.0 | 90.9 | 92.8 | 87.2 | 89.0 | 92.8 | 98.2 | 96.4 | 100.0 | 100.0 | 98.2 |
| History (When Events) | 91.7 | 100.0 | 100.0 | 100.0 | 100.0 | 100.0 | 100.0 | 100.0 | 100.0 | 100.0 | 100.0 | 100.0 | 100.0 | 100.0 | 100.0 |
| Inventions | 37.5 | 58.8 | 53.8 | 56.2 | 50.0 | 52.5 | 58.8 | 67.6 | 58.8 | 56.2 | 55.0 | 43.8 | 45.0 | 47.6 | 56.2 |
| Geography – Landmark Locations | 100.0 | 100.0 | 100.0 | 100.0 | 100.0 | 100.0 | 100.0 | 100.0 | 100.0 | 100.0 | 100.0 | 100.0 | 100.0 | 100.0 | 100.0 |
| Animals | 63.5 | 34.1 | 38.8 | 25.8 | 29.4 | 36.5 | 35.2 | 43.6 | 35.2 | 33.0 | 34.2 | 30.6 | 24.8 | 31.8 | 34.2 |
| Music/Composers | 73.3 | 80.0 | 66.7 | 70.0 | 66.6 | 66.7 | 83.4 | 66.6 | 83.4 | 83.4 | 83.4 | 83.4 | 83.4 | 83.4 | 83.4 |
| Scientific Discoveries | 57.6 | 81.2 | 82.4 | 83.6 | 88.2 | 81.2 | 76.4 | 76.4 | 76.4 | 76.4 | 74.2 | 82.4 | 75.2 | 65.8 | 76.4 |
| Who Invented | 66.3 | 82.1 | 82.1 | 85.2 | 84.2 | 78.9 | 81.0 | 69.4 | 73.6 | 73.6 | 76.8 | 73.6 | 79.0 | 73.6 | 77.8 |
| Sports (Famous Players) | 48.0 | 74.7 | 90.7 | 89.4 | 92.0 | 77.3 | 82.6 | 81.4 | 77.4 | 69.4 | 85.4 | 77.4 | 80.0 | 88.0 | 82.6 |
| Art (Painting Subjects) | 42.2 | 32.2 | 37.8 | 28.8 | 33.4 | 30.0 | 31.2 | 23.4 | 33.4 | 38.8 | 40.0 | 40.0 | 27.8 | 38.8 | 38.8 |
| Literature | 81.1 | 75.8 | 78.9 | 68.4 | 82.2 | 76.8 | 71.6 | 73.6 | 75.8 | 63.2 | 80.0 | 67.4 | 70.6 | 75.8 | 70.6 |
| **Overall** | 70.6 | 77.2 | **78.4** | 77.4 | 76.5 | 77.3 | 77.0 | 75.6 | 75.8 | 74.4 | 76.4 | 75.2 | 74.4 | 75.8 | 76.4 |

## E.2 STATISTICAL ANALYSIS

A hallucination score measures the degree to which a language model generates hallucinated outputs. Figure E.1 (panels a–d) and Figure E.2 display the mean hallucination scores (with standard error interval for the mean) for each noisy fine-tuned variant alongside its base variants. To assess whether noisy fine-tuning leads to statistically meaningful changes in the distribution of hallucination scores (beyond what can be inferred from mean and error-bar overlap alone), we employ the *Epps–Singleton two-sample test* (Epps & Singleton, 1986; Goerg & Kaiser, 2009). This test is a nonparametric method for comparing two continuous distributions without assuming normality or equal variances—properties. Moreover, by examining the full empirical distribution rather than only its first two moments (mean and variance), the Epps–Singleton test can detect shifts in shape, tail behavior, or modality that might be missed by simpler tests.

All pairwise comparisons against the base model were performed in Python using SciPy's `epps_singleton_2samp` function, with Holm's method applied to control the familywise error rate at $\alpha = 0.05$. The detailed results for each model family appear in Tables E.7–E.11. Across all models, the adjusted p-values are consistently low (e.g., 0.000 to 0.018), and the "Significant" column is uniformly True, leading to the rejection of $H_0$, indicating that the distributions of hallucination scores for the noisy fine-tuned models differ significantly from those of their base variants. The significant results across nearly all experiments confirm that adding noise during fine-tuning isn't a trivial change—it alters how the model generates outputs, as reflected in the hallucination scores. This suggests that noise acts as a regularizer, helping the model avoid overfitting to the training data and improving its ability to handle new inputs without hallucinating.

Table E.7: Statistical comparison between noisy fine-tuned models and the base variants of Llama-3.2-1B.

| Experiment | Test used | Statistic | P-value raw | P-value adjusted | Significant | Interpretation |
|---|---|---|---|---|---|---|
| Noisy1 | Epps-Singleton | 93.975 | 0.000 | 0.000 | True | Reject $H_0 \to$ distributions differ |
| Noisy2 | Epps-Singleton | 52.584 | 0.000 | 0.000 | True | Reject $H_0 \to$ distributions differ |
| Noisy3 | Epps-Singleton | 60.025 | 0.000 | 0.000 | True | Reject $H_0 \to$ distributions differ |
| Noisy4 | Epps-Singleton | 53.175 | 0.000 | 0.000 | True | Reject $H_0 \to$ distributions differ |

Table E.8: Statistical comparison between noisy fine-tuned models and the base variants of Llama-3.2-3B.

| Experiment | Test used | Statistic | P-value raw | P-value adjusted | Significant | Interpretation |
|---|---|---|---|---|---|---|
| Noisy1 | Epps-Singleton | 46.320 | 0.000 | 0.000 | True | Reject $H_0 \to$ distributions differ |
| Noisy2 | Epps-Singleton | 57.663 | 0.000 | 0.000 | True | Reject $H_0 \to$ distributions differ |
| Noisy3 | Epps-Singleton | 60.415 | 0.000 | 0.000 | True | Reject $H_0 \to$ distributions differ |
| Noisy4 | Epps-Singleton | 62.553 | 0.000 | 0.000 | True | Reject $H_0 \to$ distributions differ |

Table E.9: Statistical comparison between noisy fine-tuned models and the base variants of Qwen2.5-0.5B.

| Experiment | Test used | Statistic | P-value raw | P-value adjusted | Significant | Interpretation |
|---|---|---|---|---|---|---|
| Noisy1 | Epps-Singleton | 19.060 | 0.001 | 0.002 | True | Reject $H_0 \to$ distributions differ |
| Noisy2 | Epps-Singleton | 26.945 | 0.000 | 0.000 | True | Reject $H_0 \to$ distributions differ |
| Noisy3 | Epps-Singleton | 11.742 | 0.019 | 0.019 | True | Reject $H_0 \to$ distributions differ |
| Noisy4 | Epps-Singleton | 33.286 | 0.000 | 0.000 | True | Reject $H_0 \to$ distributions differ |

Table E.10: Statistical comparison between noisy fine-tuned models and the base variants of Gemma-3-1B-it.

| Experiment | Test used | Statistic | P-value raw | P-value adjusted | Significant | Interpretation |
|---|---|---|---|---|---|---|
| Noisy1 | Epps-Singleton | 19.498 | 0.001 | 0.002 | True | Reject $H_0 \to$ distributions differ |
| Noisy2 | Epps-Singleton | 23.610 | 0.000 | 0.000 | True | Reject $H_0 \to$ distributions differ |
| Noisy3 | Epps-Singleton | 7.663 | 0.105 | 0.105 | False | Fail to reject $H_0 \to$ no evidence of difference |
| Noisy4 | Epps-Singleton | 18.095 | 0.001 | 0.002 | True | Reject $H_0 \to$ distributions differ |

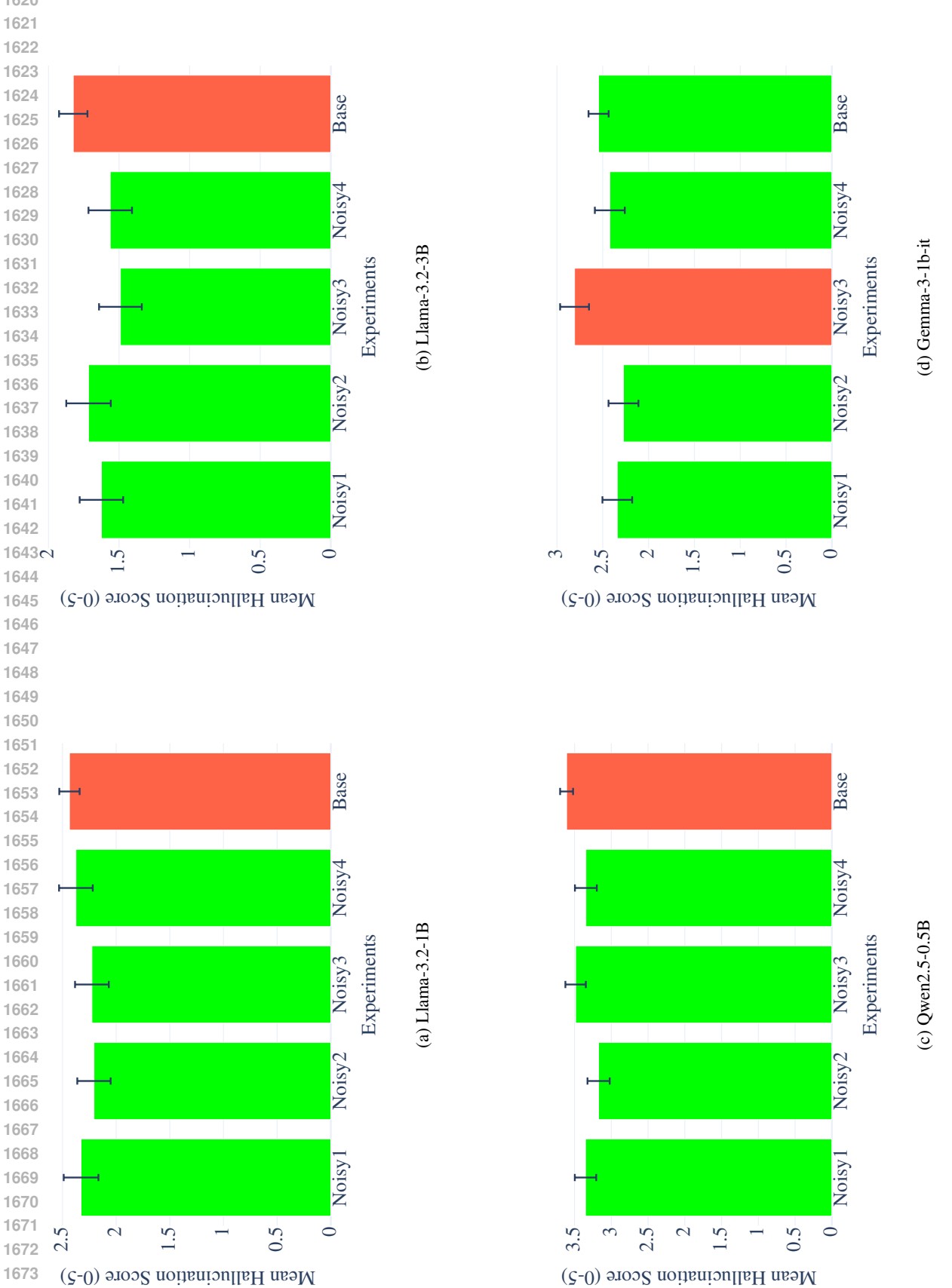

Figure E.1: Error bars for the mean hallucination scores across the models and experiments

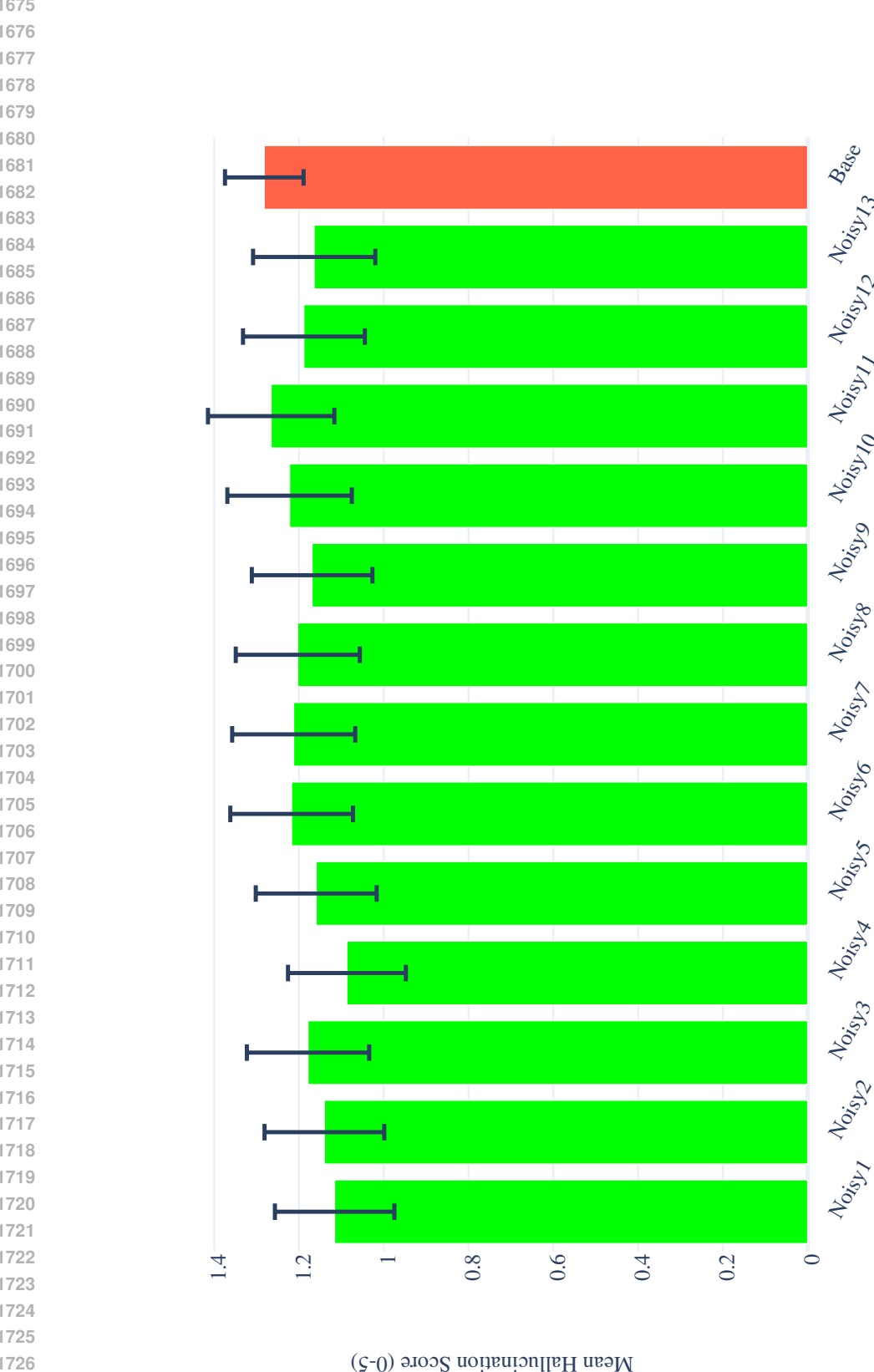

Figure E.2: Error bars for the mean hallucination scores across the experiments for Mistral-7B-V0.1.

Table E.11: Statistical comparison between noisy fine-tuned models and the base variants of Mistral-7B-v0.1.

| Experiment | Test used | Statistic | P-value raw | P-value adjusted | Significant | Interpretation |
|---|---|---|---|---|---|---|
| Noisy1 | Epps-Singleton | 63.822 | 0.000 | 0.000 | True | Reject $H_0 \rightarrow$ distributions differ |
| Noisy2 | Epps-Singleton | 49.946 | 0.000 | 0.000 | True | Reject $H_0 \rightarrow$ distributions differ |
| Noisy3 | Epps-Singleton | 63.714 | 0.000 | 0.000 | True | Reject $H_0 \rightarrow$ distributions differ |
| Noisy4 | Epps-Singleton | 50.750 | 0.000 | 0.000 | True | Reject $H_0 \rightarrow$ distributions differ |
| Noisy5 | Epps-Singleton | 54.740 | 0.000 | 0.000 | True | Reject $H_0 \rightarrow$ distributions differ |
| Noisy6 | Epps-Singleton | 54.225 | 0.000 | 0.000 | True | Reject $H_0 \rightarrow$ distributions differ |
| Noisy7 | Epps-Singleton | 63.442 | 0.000 | 0.000 | True | Reject $H_0 \rightarrow$ distributions differ |
| Noisy8 | Epps-Singleton | 83.037 | 0.000 | 0.000 | True | Reject $H_0 \rightarrow$ distributions differ |
| Noisy9 | Epps-Singleton | 57.399 | 0.000 | 0.000 | True | Reject $H_0 \rightarrow$ distributions differ |
| Noisy10 | Epps-Singleton | 77.249 | 0.000 | 0.000 | True | Reject $H_0 \rightarrow$ distributions differ |
| Noisy11 | Epps-Singleton | 85.994 | 0.000 | 0.000 | True | Reject $H_0 \rightarrow$ distributions differ |
| Noisy12 | Epps-Singleton | 60.581 | 0.000 | 0.000 | True | Reject $H_0 \rightarrow$ distributions differ |
| Noisy13 | Epps-Singleton | 69.129 | 0.000 | 0.000 | True | Reject $H_0 \rightarrow$ distributions differ |

## F  COMPUTATIONAL EFFICIENCY AND SCALABILITY ANALYSIS

To evaluate the computational efficiency and scalability of the proposed NoiseFiT framework compared to the common fine-tuning (BaseFiT), we analyzed a series of GPU performance metrics recorded during the experiments (Figures F.1 and F.2). The metrics under consideration include:

- **GPU Memory Allocated (%)** – Indicates the percentage of total GPU memory used.

- **GPU Power Usage (%)** – Reflects the power consumption during model training.

- **GPU Temperature** (°C) – Monitors the thermal performance of the GPU.

- **Time Spent Accessing Memory (%)** – Measures the relative time the GPU spent in memory operations.

- **GPU Utilization (%)** – Captures the overall usage of the GPU computational resources.

For each metric, we computed the mean and standard deviation over multiple experimental runs. Table F.1 summarizes the performance for both BaseFiT (Base) and NoiseFiT configurations. The results indicate that the NoiseFiT framework exhibits a mixed performance profile across the evaluated GPU metrics:

- **Memory and Power Efficiency:** While NoiseFiT requires a higher GPU memory allocation (61.3% vs. 35.5%), it achieves reduced power usage (64.0% vs. 67.3%). This suggests that, despite the increased memory demand, NoiseFiT benefits from lower energy consumption during training.

- **Thermal Performance and Memory Operations:** The GPU temperature and the time spent accessing memory are marginally elevated in the NoiseFiT setup (58.6°C and 50.7%, respectively) compared to BaseFiT (57.8°C and 49.0%). These slight differences indicate that thermal management and memory operation times remain largely comparable between the two approaches.

- **Overall Utilization:** The slightly lower overall GPU utilization observed with NoiseFiT (75.5% vs. 77.2%) implies that similar or improved performance may be achieved with a reduced computational load, which is beneficial for scalability.

In summary, the performance trade-offs observed with the NoiseFiT suggest a viable balance between computational efficiency and resource allocation. Although NoiseFiT demands higher memory usage and shows marginal increases in thermal metrics, its reduced power consumption and overall GPU utilization indicate that it can mitigate hallucinations while decreasing the computational overhead associated with training. These benefits are especially critical when scaling large language models in resource-constrained environments, thereby enhancing both the practicality and the environmental sustainability of deploying such systems.

Table F.1: Summary of GPU performance metrics statistics comparing Base Fine Tuning (BaseFiT) and Noisy Fine Tuning (NoiseFiT) workflows. Values represent the mean $\pm$ standard deviation across multiple runs.

| Metric | BaseFiT | NoiseFiT |
|---|---|---|
| GPU Memory Allocated (%) | $35.5 \pm 0.0$ | $61.3 \pm 0.5$ |
| GPU Power Usage (%) | $67.3 \pm 14.0$ | $64.0 \pm 16.1$ |
| GPU Temperature (°C) | $57.8 \pm 2.2$ | $58.6 \pm 1.2$ |
| Time Spent Accessing Memory (%) | $49.0 \pm 13.5$ | $50.7 \pm 19.4$ |
| GPU Utilization (%) | $77.2 \pm 20.6$ | $75.5 \pm 18.3$ |

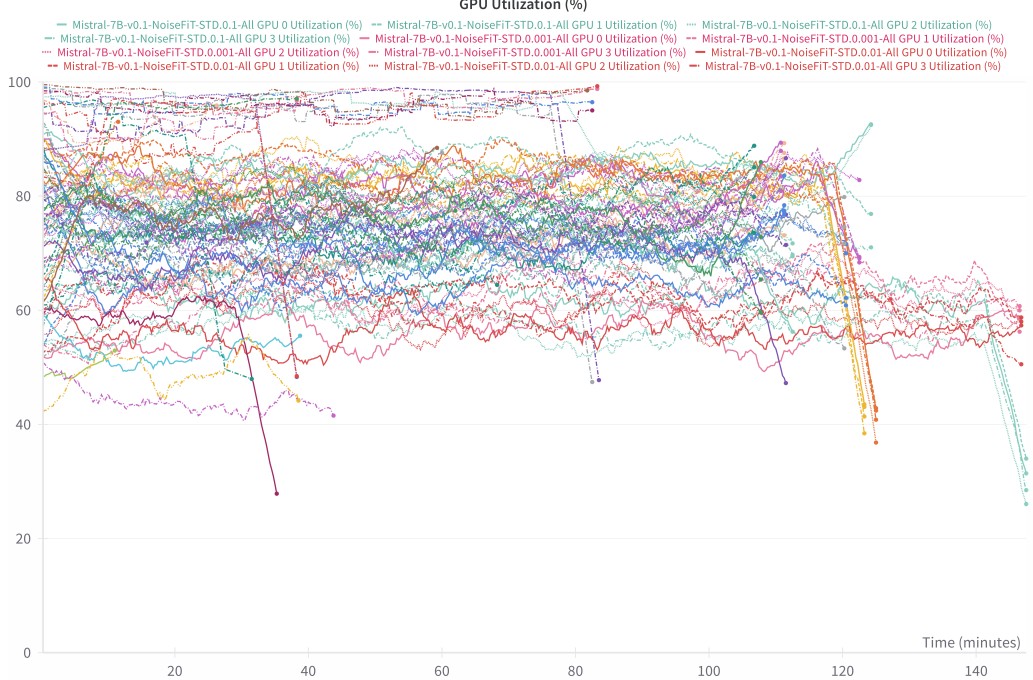

Figure F.1: NoiseFiT training GPU utilization history for different models, noise injection STD and layer selection strategies. Available in interactive mode online at W&B.

## G ANALYSIS OF LAYER-WISE METRICS

In this section, we analyze the layer-wise insights for the models. First, we provide an analysis of the SNR trends per layer across multiple noise standard deviation (STD) values in the five models (Figures G.1- G.5). Then, we provide an analysis of the metrics including sparsity, variance, logit entropy, attention entropy, mean L2 norm, and rank of the hidden states across layers.

For Llama-3.2-1B, Llama-3.2-3B, and Mistral-7B-v0.1, SNR increases with layer index for all noise STD values. Conversely, gemma-3-1b-it shows a unique decreasing SNR trend across layers, with the decline more pronounced at lower STD values, indicating greater noise sensitivity in deeper layers. Qwen2.5-0.5B presents a mixed trend: SNR remains stable for lower STD values but declines for higher STD values, reflecting varying noise tolerance. Across all models, higher STD values consistently yield lower SNR. The diversity in trends suggests that the model's architecture plays a crucial role in how noise injection influences the fine-tuning.

Here we briefly define the metrics used:

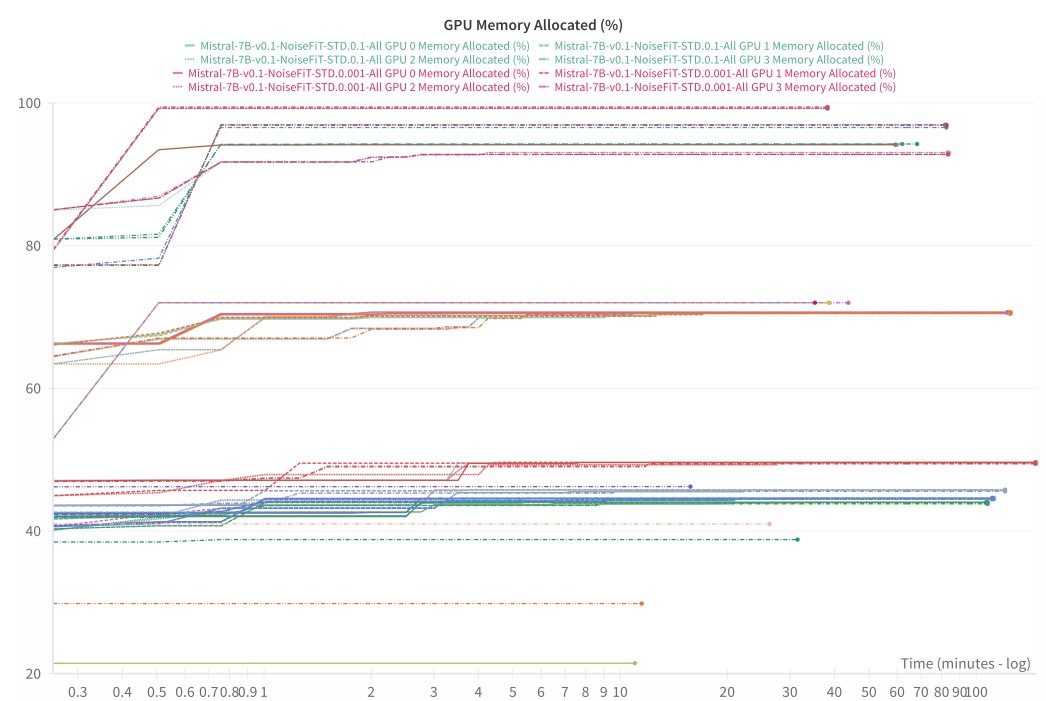

Figure F.2: NoiseFiT training GPU memory allocation history for different models, noise injection STD and layer selection strategies. Available in interactive mode online at W&B.

- **Sparsity**: The proportion of zero or near-zero values in the hidden states, indicating how many features are inactive. Higher sparsity suggests a focus on fewer, potentially more robust features.

- **Variance**: The spread of hidden state activations. Higher variance may indicate greater expressiveness, while lower variance suggests stability.

- **Logit Entropy**: Measures uncertainty in the model's output predictions. Lower entropy reflects higher confidence, while higher entropy indicates more uncertainty.

- **Attention Entropy**: Assesses the distribution of attention weights. Lower entropy implies concentrated attention on specific tokens, while higher entropy suggests more uniform attention.

- **Mean L2 Norm**: The magnitude of hidden state activations. Larger norms indicate stronger activations, while smaller norms suggest subdued activations.

- **Rank**: The effective rank of hidden states, reflecting the dimensionality of information processed. Higher rank suggests more complex representations.

## G.1 LLAMA-3.2-1B

- **Sparsity (Fig. G.6a)**: The noisy variants show higher sparsity across most layers, especially in the middle layers, compared to BaseFiT and the Base model. This suggests that noise promotes sparser, potentially more robust representations. BaseFiT exhibits lower sparsity, indicating reliance on more features, while the Base model maintains moderate sparsity compared to other models.

- **Variance (Fig. G.6b)**: Base model displays higher variance, reflecting more diverse activations. BaseFiT and noisy variants show lower variance, suggesting more stable activations.

- **Logit Entropy (Fig. G.6c)**: Noisy variants exhibit lower logit entropy median and higher logit entropy variance, which may improve calibration. Base model shows moderate entropy relatively higher median with less variance.

- **Attention Entropy (Fig. G.6d)**: Noisy variants have higher attention entropy, implying more distributed attention across tokens. The Base model's entropy increases gradually across layers.

- **Mean L2 Norm (Fig. G.6e)**: Noisy variants have lower mean L2 norms, especially in deeper layers, suggesting smaller activations. Base model exhibits higher norms, indicating stronger activations.

- **Rank (Fig. G.6f)**: The rank is lower for base model, particularly in later layers, suggesting compressed representations. The noisy variants show consistently higher rank.

## G.2 LLAMA-3.2-3B

- **Sparsity (Fig. G.7a)**: Noisy variants exhibit fluctuating higher sparsity, especially in middle and later layers.

- **Variance (Fig. G.7b)**: Noisy variants have higher variance. The Base model shows the lowest variance.

- **Logit Entropy (Fig. G.7c)**: Noisy variants display lower logit entropy, indicating greater confidence, while Base model has higher entropy.

- **Attention Entropy (Fig. G.7d)**: Noisy variants have slightly higher attention entropy, especially in deeper layers, suggesting distributed attention. Base model's entropy decreases gradually relative to other models.

- **Mean L2 Norm (Fig. G.7e)**: Mean L2 norms exhibit relatively similar pattern across all models.

- **Rank (Fig. G.7f)**: Ranks exhibit relatively similar pattern across all models.

## G.3 QWEN2.5-0.5B

- **Sparsity (Fig. G.8a)**: Noisy variants show higher sparsity, particularly in earlier layers, compared to the Base model.

- **Variance (Fig. G.8b)**: Base model exhibits relatively higher variance in the middle layers with the BaseFiT and noisy variants maintaining lower variance across these layers.

- **Logit Entropy (Fig. G.8c)**: Noisy variants have show higher output logit entropy median and variance, indicating less confidence. Base model shows lower entropy.

- **Attention Entropy (Fig. G.8d)**: Mean attention entropy exhibits similar pattern across all models.

- **Mean L2 Norm (Fig. G.8e)**: Mean L2 norms exhibit relatively similar pattern across all models.

- **Rank (Fig. G.8f)**: Ranks exhibit relatively similar pattern across all models.

## G.4 GEMMA-3-1B-IT

- **Sparsity (Fig. G.9a)**: Sparsity exhibits relatively similar pattern across all models.

- **Variance (Fig. G.9b)**: Variance exhibits relatively similar pattern across all models.

- **Logit Entropy (Fig. G.9c)**: Noisy variants display relatively higher logit entropy median and higher variance, while Base model shows lower entropy median and variance.

- **Attention Entropy (Fig. G.9d)**: Attention entropy exhibits relatively similar pattern across all models.

- **Mean L2 Norm (Fig. G.9e)**: Logit entropy exhibits relatively similar pattern across all models with the noisy variants exhibiting relatively higher mean L2 norms in deeper layers.

- **Rank (Fig. G.9f)**: Rank exhibits relatively similar pattern across all models with the noisy variants exhibiting relatively higher rank in deeper layers.

### G.5 MISTRAL-7B-V0.1

Based on the layer-wise metrics for Mistral-7B-v0.1, we observe the following trends:

- **Sparsity (Fig. G.10a)**: Sparsity exhibits similar pattern across all models.
- **Variance (Fig. G.10b)**: Variance exhibits similar pattern across all models.
- **Logit Entropy (Fig. G.10c)**: The noisy variants show lower logit entropy, suggesting high confidence in predictions. Base model displays higher entropy.
- **Attention Entropy (Fig. G.10d)**: Attention entropy exhibits relatively similar pattern across all models with the noisy variants exhibiting relatively higher entropy in deeper layers.
- **Mean L2 Norm (Fig. G.10e)**: The Mean L2 norms exhibit similar pattern across all models.
- **Rank (Fig. G.10f)**: Ranks exhibit relatively similar pattern across all models.

### G.6 SYNTHESIS OF THE FINDINGS

Across all five models, consistent patterns emerge:

- **Sparsity**: Noisy variants exhibit higher sparsity compared to their base counterparts, particularly in specific layers. In LLaMA-3.2-1B, noisy variants show elevated sparsity in middle layers (Fig. G.6a), while in LLaMA-3.2-3B, this increase is more pronounced in middle and later layers (Fig. G.7a). Similarly, Qwen2.5-0.5B displays higher sparsity in earlier layers for noisy variants (Fig. G.8a). This pattern suggests that noise injection may encourage sparser representations, potentially enhancing robustness by focusing on fewer, critical features. However, in Gemma-3-1b-it and Mistral-7B-v0.1, sparsity remains relatively consistent across all variants (Figs. G.9a and G.10a), indicating that the impact of noise on sparsity may be architecture-dependent.

- **Variance**: In LLaMA-3.2-1B and Qwen2.5-0.5B, noisy variants tend to have lower variance compared to the base models, particularly noticeable in middle layers for Qwen2.5-0.5B (Figs. G.6b and G.8b), suggesting more stable activations. In contrast, LLaMA-3.2-3B shows higher variance in noisy variants (Fig. G.7b), indicating greater activation diversity. Gemma-3-1b-it and Mistral-7B-v0.1 exhibit similar variance patterns across all variants (Figs. G.9b and G.10b), highlighting that the effect of noise on activation spread is not uniform and likely influenced by model size or structure.

- **Logit Entropy**: In LLaMA-3.2-1B, noisy variants have a lower median but higher variance in logit entropy (Fig. G.6c), potentially indicating better calibration. LLaMA-3.2-3B and Mistral-7B-v0.1 show lower logit entropy in noisy variants (Figs. G.7c and G.10c), suggesting increased prediction confidence. Conversely, Qwen2.5-0.5B and Gemma-3-1b-it exhibit higher median and variance in logit entropy for noisy variants (Figs. G.8c and G.9c), pointing to greater uncertainty.

- **Attention Entropy**: Attention entropy tends to increase in noisy variants across multiple models. LLaMA-3.2-1B shows higher attention entropy in noisy variants (Fig. G.6d), while LLaMA-3.2-3B and Mistral-7B-v0.1 exhibit slightly higher entropy in deeper layers (Figs. G.7d and G.10d). This trend suggests that noise promotes more distributed attention across tokens, possibly improving contextual awareness. In Qwen2.5-0.5B and Gemma-3-1b-it, attention entropy patterns are largely similar across variants (Figs. G.8d and G.9d), indicating less pronounced effects in these models.

- **Mean L2 Norm**: The mean L2 norm generally shows consistent patterns across variants in most models, with some exceptions. In LLaMA-3.2-1B, noisy variants have lower mean L2 norms, especially in deeper layers (Fig. G.6e), suggesting subdued activations, whereas Gemma-3-1b-it displays higher norms in noisy variants in deeper layers (Fig. G.9e). LLaMA-3.2-3B, Qwen2.5-0.5B, and Mistral-7B-v0.1 exhibit similar norm patterns across all variants (Figs. G.7e, G.8e, and G.10e), suggesting that noise impact on activation magnitude varies by model.

- **Rank**: LLaMA-3.2-1B's noisy variants maintain a higher rank, particularly in later layers (Fig. G.6f), indicating more complex representations. In Gemma-3-1b-it, noisy variants also

show higher rank in deeper layers (Fig. G.9f), while LLaMA-3.2-3B, Qwen2.5-0.5B, and Mistral-7B-v0.1 display similar rank patterns across variants (Figs. G.7f, G.8f, and G.10f). This suggests that noise may enhance representational dimensionality in some models.

These findings indicate that noise injection influences the internal representations of the models in various ways. Increased sparsity and attention entropy are relatively consistent effects, while variance, logit entropy, and mean L2 norm exhibit model-specific responses, highlighting the role of architecture in noisy fine-tuning. Our findings demonstrate that NoiseFiT can effectively alter layer-wise hidden states characteristics of language models for mitigating hallucinations. The increased sparsity and attention entropy in noisy variants align with goals of reducing overfitting and enhancing generalization. However, the mixed effects on variance and logit entropy emphasize the complexity of noise's impact and the need for careful calibration.

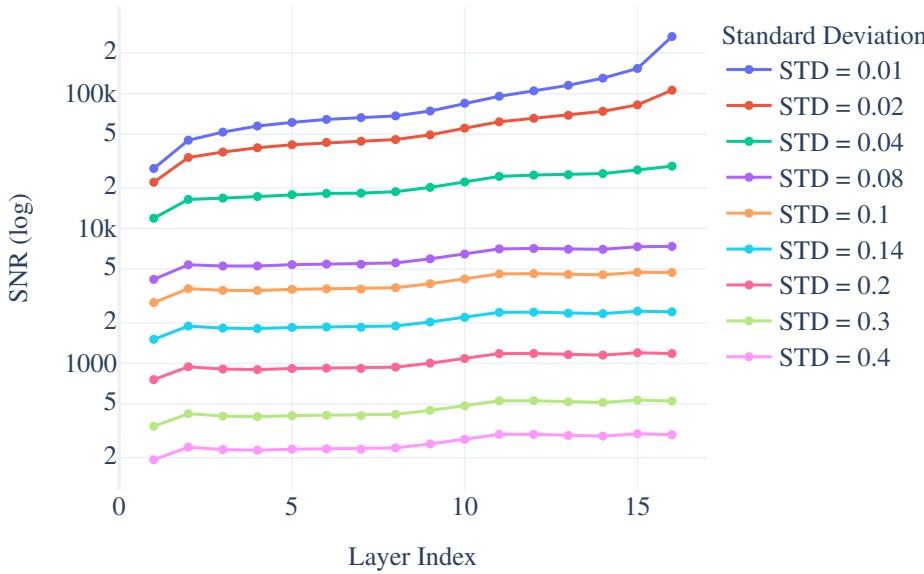

Figure G.1: Layerwise SNR for Llama-3.2-1B across different noise standard deviation values

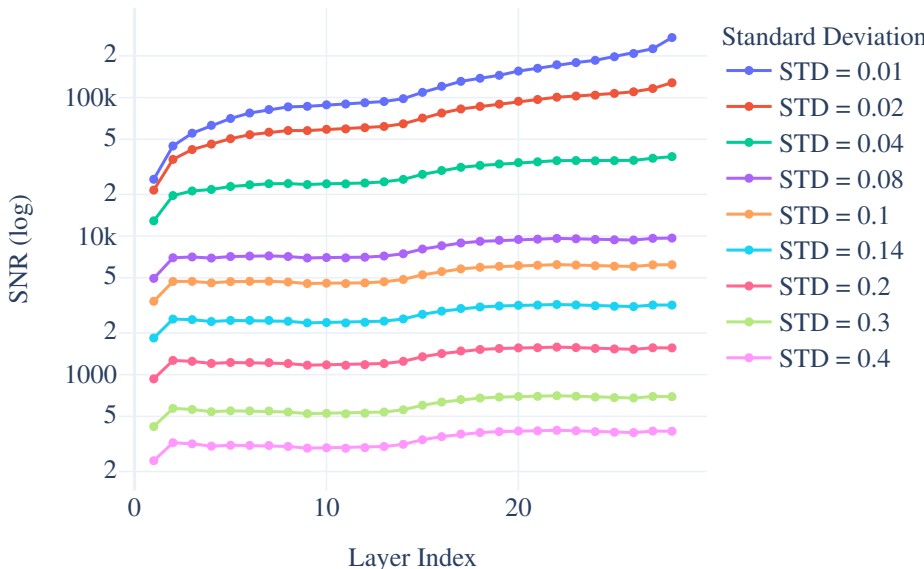

Figure G.2: Layerwise SNR for Llama-3.2-3B across different noise standard deviation values

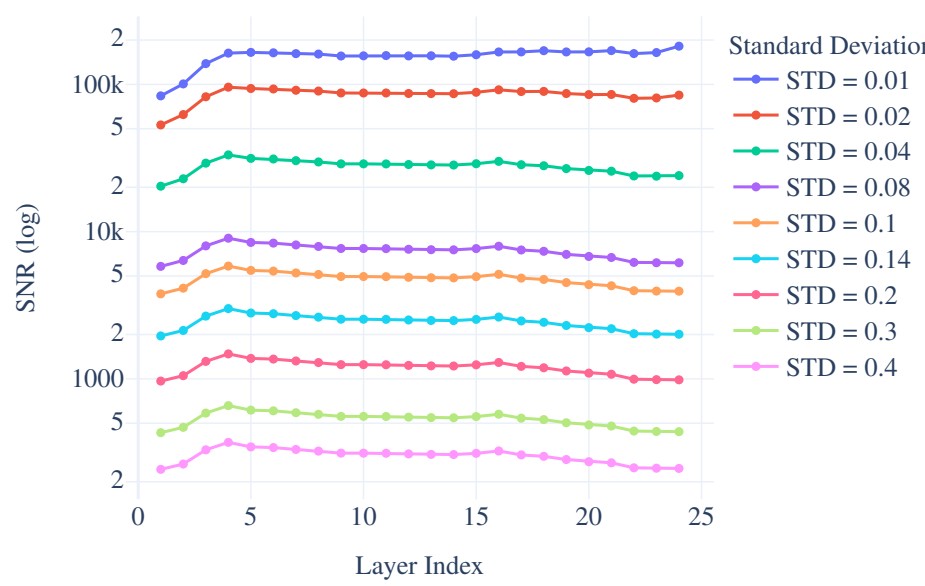

Figure G.3: Layerwise SNR for Qwen2.5-0.5B across different noise standard deviation values

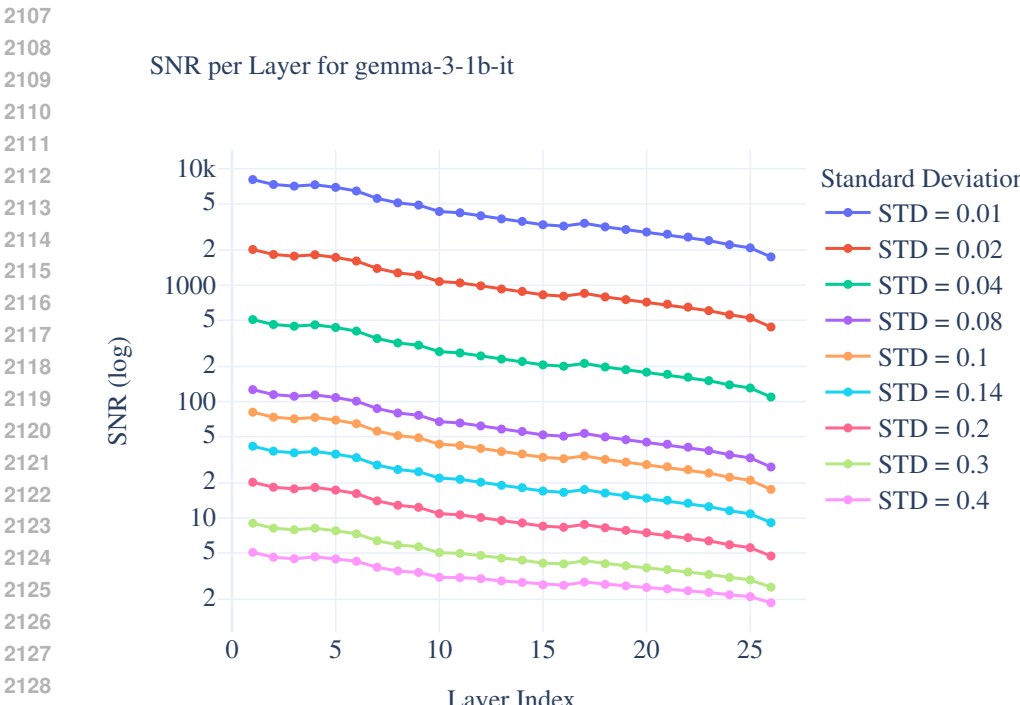

Figure G.4: Layerwise SNR for gemma-3-1b-it across different noise standard deviation values

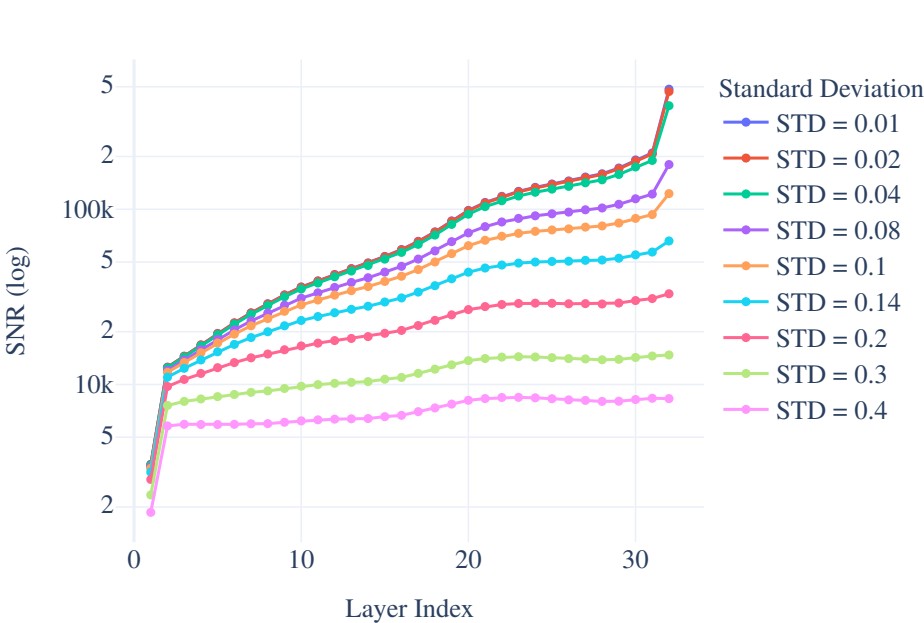

Figure G.5: Layerwise SNR for Mistral-7B-v0.1 across different noise standard deviation values

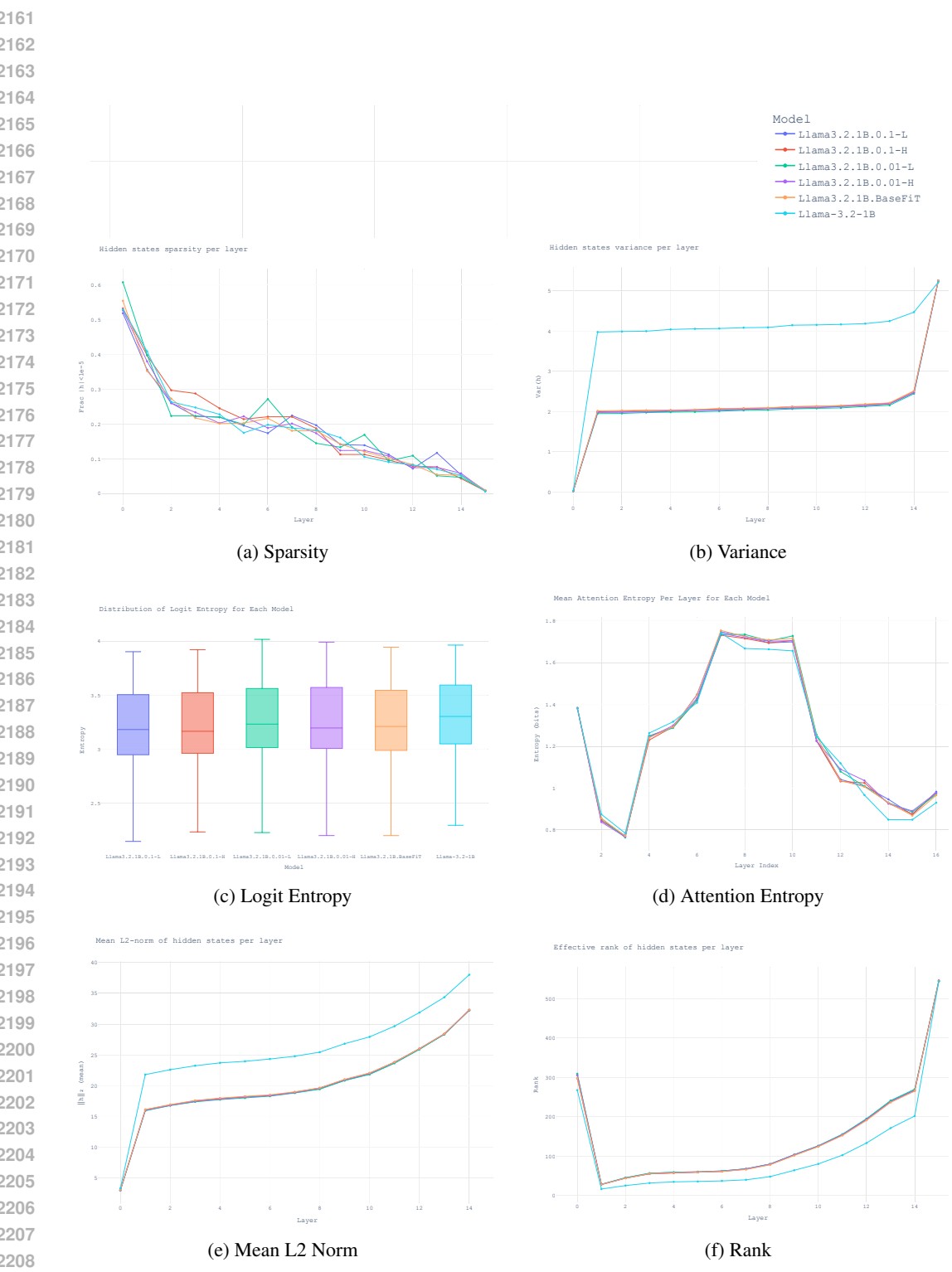

Figure G.6: Layerwise metrics for LLaMA-3.2-1B

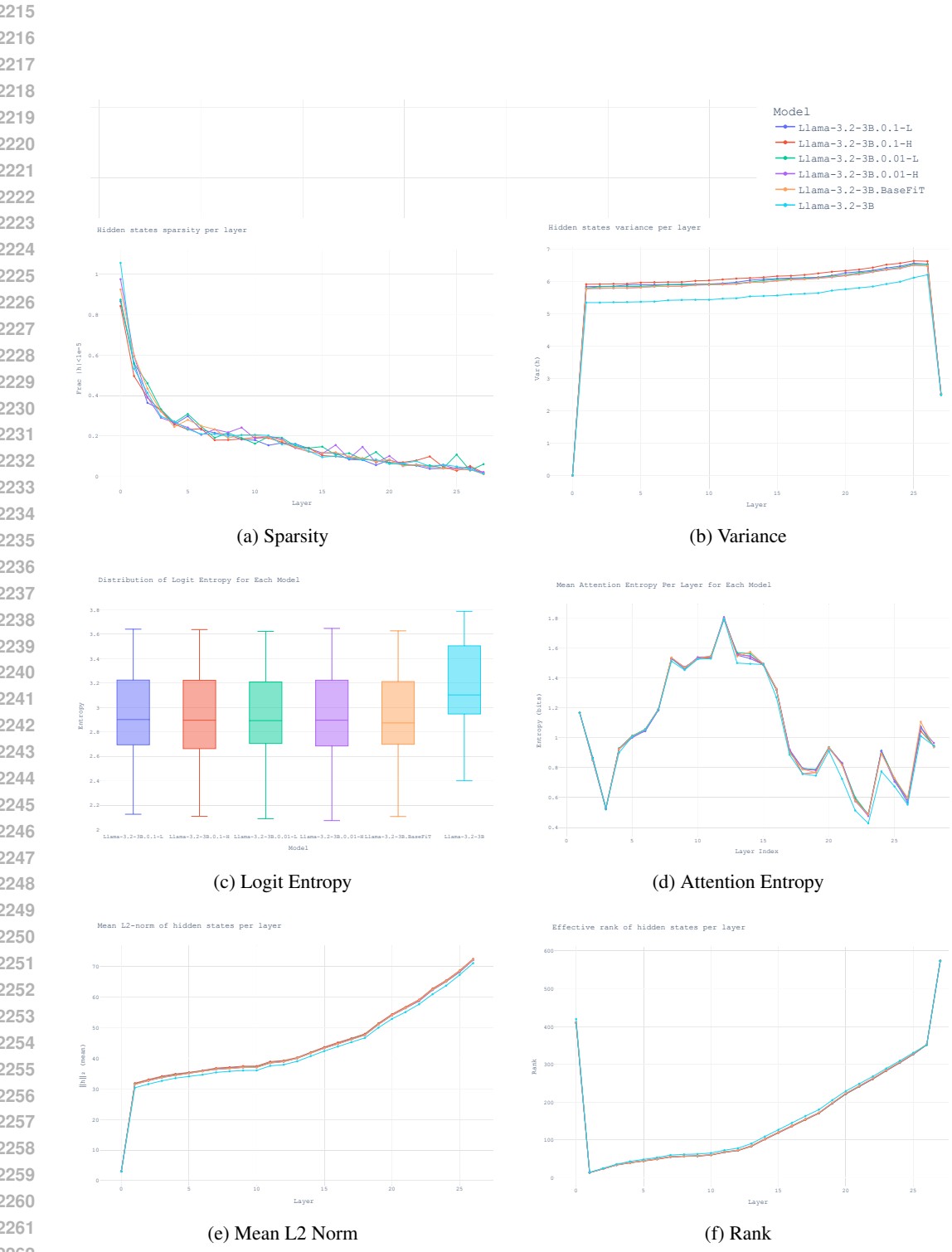

Figure G.7: Layerwise metrics for LLaMA-3.2-3B

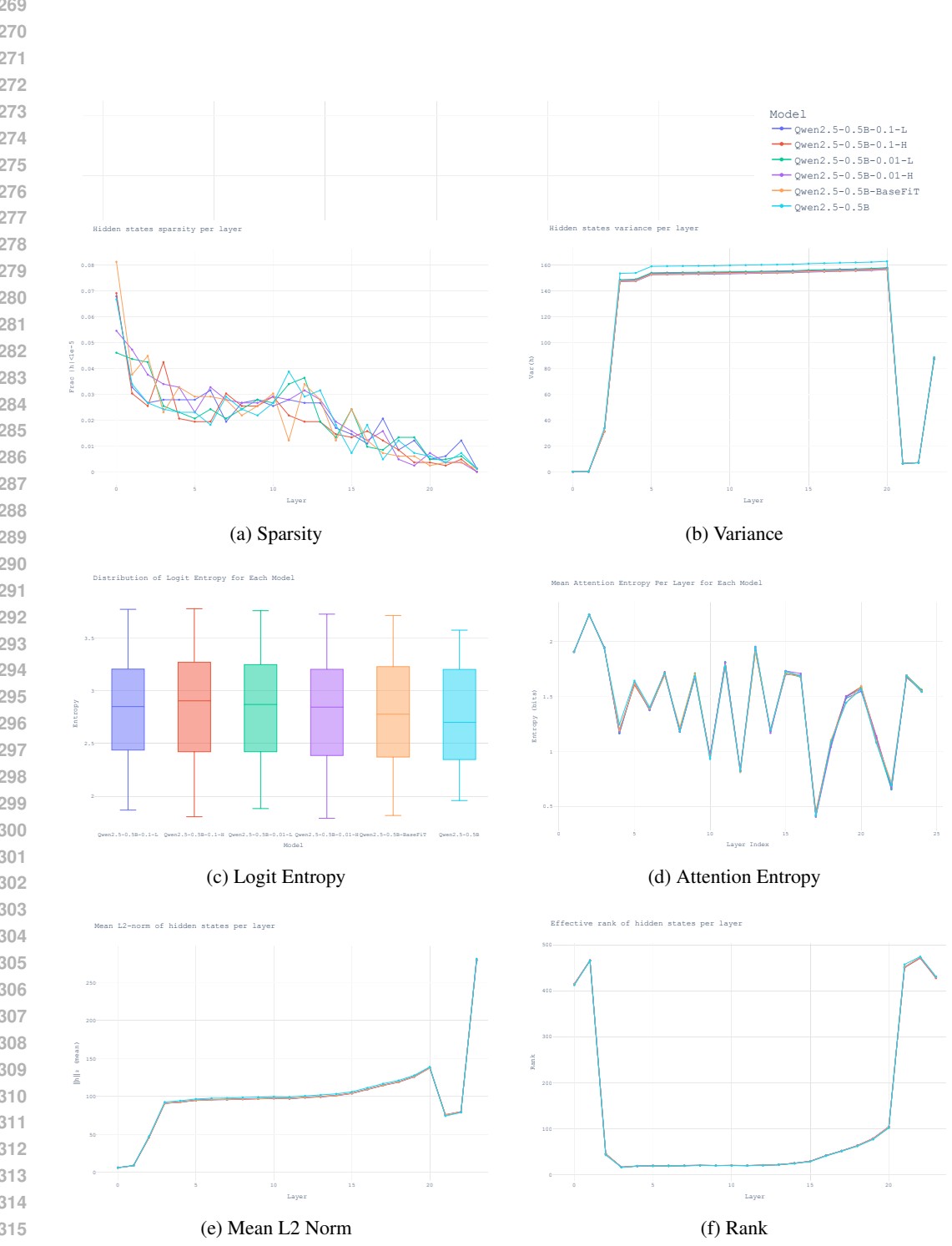

Figure G.8: Layerwise metrics for Qwen2.5-0.5B:

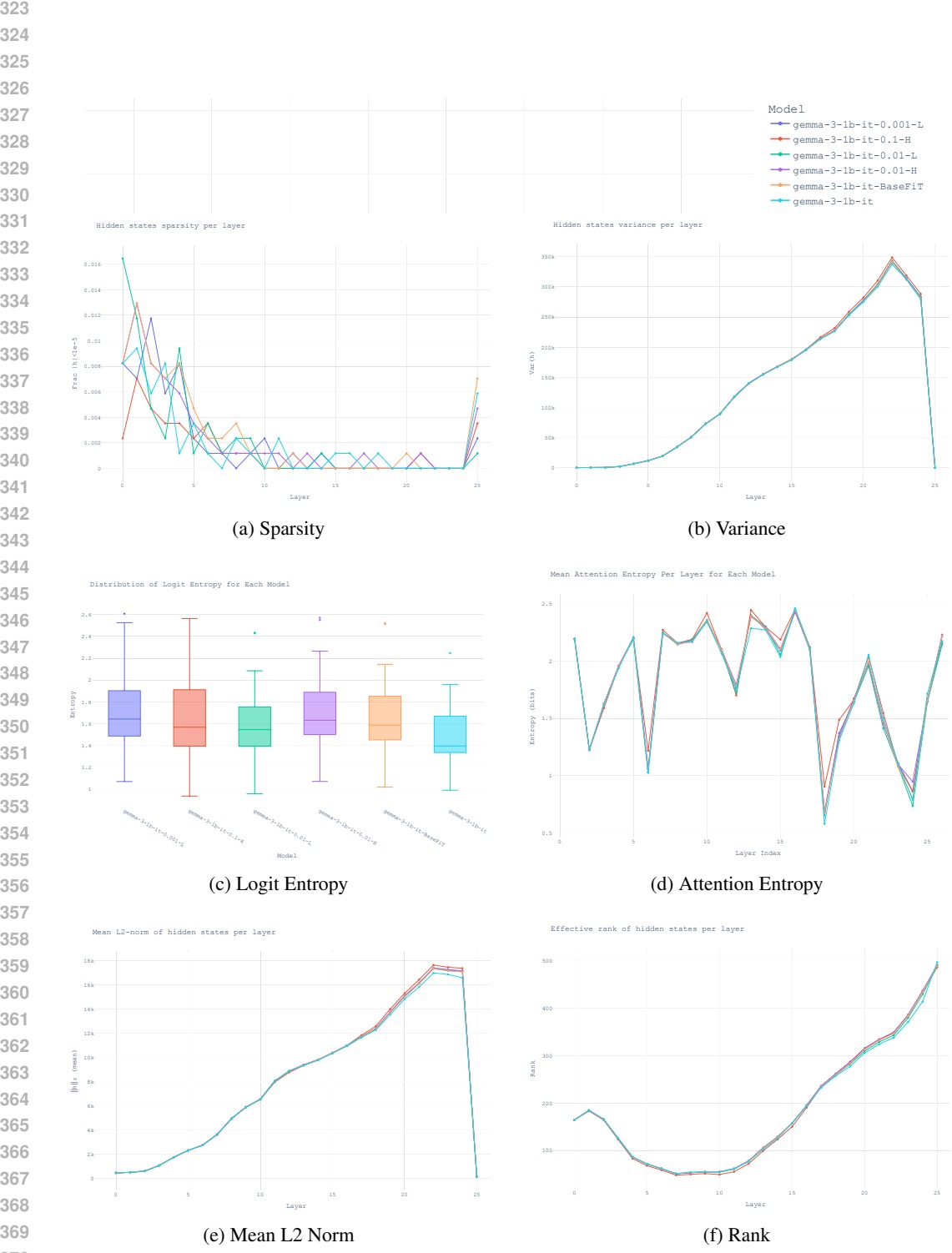

(a) Sparsity

(b) Variance

(c) Logit Entropy

(d) Attention Entropy

(e) Mean L2 Norm

(f) Rank

Figure G.9: Layerwise metrics for Gemma-3-1b-it:

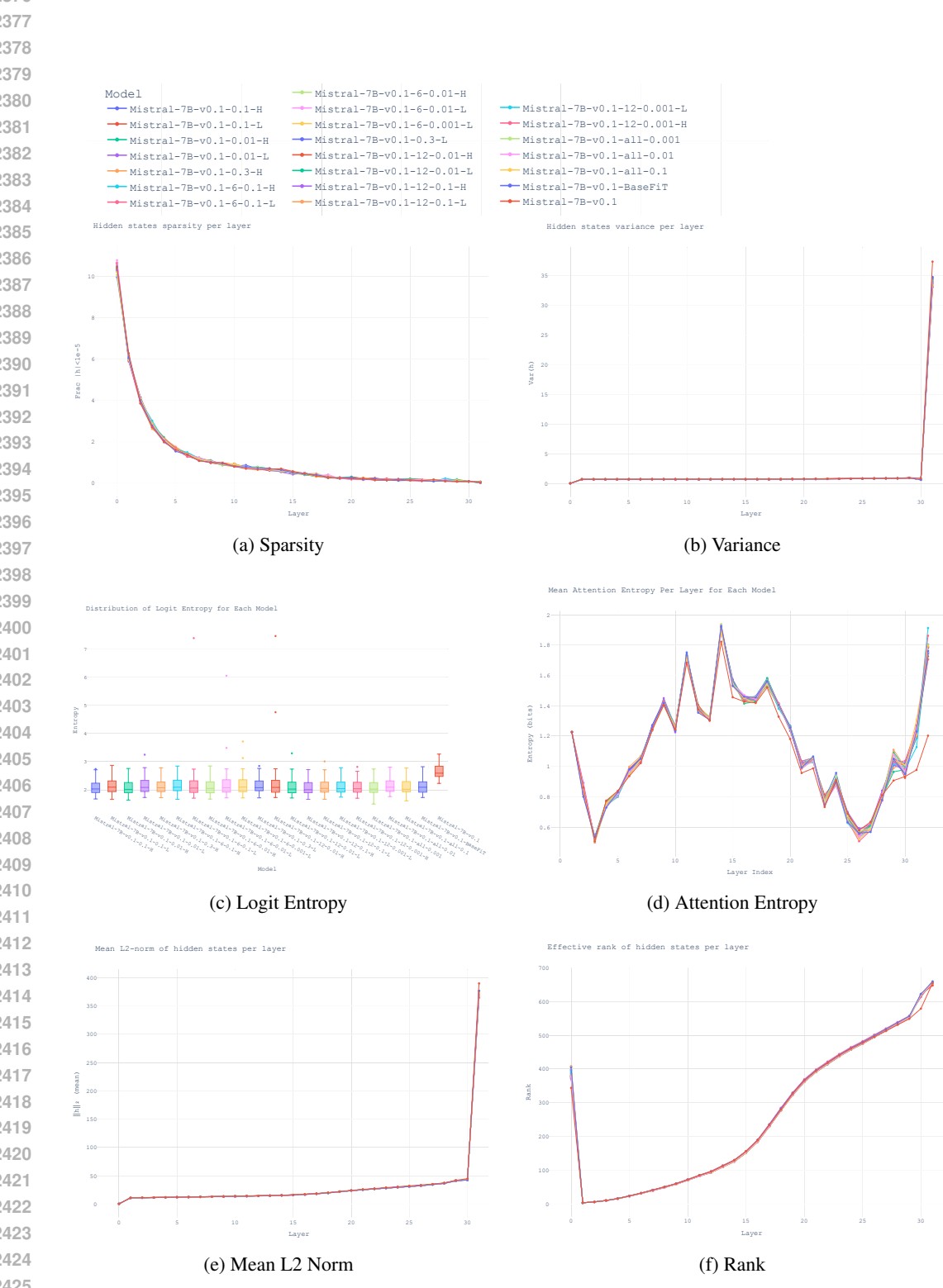

Figure G.10: Layerwise metrics for Mistral-7B-v0.1:

