# OpenReview forum: "Noise Augmented Fine Tuning for Mitigating Hallucinations in Large Language Models"
_ICLR.cc/2026/Conference — Submitted to ICLR 2026_

### Official Review · Reviewer_jnC9 · 2025-10-15

**Soundness:** 2
**Presentation:** 2
**Contribution:** 2
**Rating:** 2
**Confidence:** 4

**Summary:**

This paper proposes a fine-tuning framework called NoiseFiT, which enhances model robustness and reduces hallucinations by injecting adaptive Gaussian noise into specific Transformer layers during fine-tuning and employing a hybrid loss function (cross-entropy loss + soft-target loss + consistency loss). Experimental results across multiple tasks and models demonstrate the effectiveness of the method.

**Strengths:**

1. The method combines multiple regularization techniques (soft-target loss and consistency loss) to form a unified training objective.

2. It has been extensively validated across multiple model families and tasks, producing convincing results.

**Weaknesses:**

1. The idea of merging multiple losses is not novel, but it appears to be the main innovation of the paper.

2. Limited dataset size and diversity: The training set contains only 832 samples and consists of synthetic data. Although the authors emphasize it as “simple and effective,” this may still affect the generalization ability of the method.

3. Insufficient comparison with other methods: The paper mainly compares with BaseFiT (fine-tuning without noise) and lacks thorough comparison with other advanced hallucination mitigation methods such as RAG, RLHF, and Self-Consistency.

4. Hyperparameter sensitivity: The paper mentions that noise intensity, layer selection, and other hyperparameters significantly affect results and need to be adjusted for different models.

5. Hallucination evaluation depends on external models: Some hallucination assessments use GROK 3.0 as the “judge,” which may introduce evaluation bias.

6. Theoretical analysis is rich but somewhat lengthy: The appendix contains extensive theoretical content, some of which is not closely related to the core method.

**Questions:**

1. The idea of merging multiple losses is not novel, but it appears to be the main innovation of the paper.

2. Limited dataset size and diversity: The training set contains only 832 samples and consists of synthetic data. Although the authors emphasize it as “simple and effective,” this may still affect the generalization ability of the method.

3. Insufficient comparison with other methods: The paper mainly compares with BaseFiT (fine-tuning without noise) and lacks thorough comparison with other advanced hallucination mitigation methods such as RAG, RLHF, and Self-Consistency.

4. Hyperparameter sensitivity: The paper mentions that noise intensity, layer selection, and other hyperparameters significantly affect results and need to be adjusted for different models.

5. Hallucination evaluation depends on external models: Some hallucination assessments use GROK 3.0 as the “judge,” which may introduce evaluation bias.

6. Theoretical analysis is rich but somewhat lengthy: The appendix contains extensive theoretical content, some of which is not closely related to the core method.

---

> ### Author Response · Authors · 2025-11-12
> **Our response to reviewer jnC9 comments**
>
> We thank Reviewer jnC9 for the careful assessment and respond point-by-point.
>
> ---
>
> **(1) Novelty is in SNR-guided, adaptive layer-wise noise, not in the loss mix.**
> Sec. 3.2 and Algorithm A.1 introduce a *data-driven layer selection* scheme: we estimate per-layer SNR from paired clean/noisy passes (Eqs. (1–3)), rank layers, and inject noise only into a small subset (typically 3–6). Sec. 3.3 then defines *adaptive scaling* using robust statistics (median/MAD), an exponential re-weighting, and an uncertainty factor from logit entropy or hidden-state variance (Eqs. (4–12)). This produces input- and layer-conditioned perturbations that are fundamentally different from prior uniform or embedding-only noise. The CE + soft-CE + consistency losses in Sec. 3.4 (Eq. (19)) are deliberately used to stabilize learning under these state-dependent perturbations; we do not present the loss combination itself as the main contribution.
>
> ---
>
> **(2) Dataset size and generalization.**
> The 832-example synthetic SFT set (Sec. 3.1) is intentionally simple so that we isolate the *regularization* effect of NoiseFiT rather than introduce new knowledge (synthetic dataset had been manually annotated for factual consistency). Without relying on any domain-specific data, NoiseFiT delivers consistent gains across diverse domain-heavy benchmarks while reducing hallucinations—at zero inference-time cost. Generalization is evaluated on diverse public benchmarks (Sec. 4.2.1): MMLU-Pro, BBH, GPQA, MATH, MUSR, IFEval, TruthfulQA-MC, and HaluEval. Table 1 and the extended results for Mistral-7B (Tab. D.1) show that NoiseFiT consistently reduces hallucinations while matching or improving task performance across LLaMA, Qwen, Gemma, and Mistral, despite the minimal SFT data. This supports that the method is not over-fitted to the synthetic dataset.
>
> **(3) Relation to RAG / RLHF / self-consistency.**
> As discussed in Sec. 5 (Limitations), these methods act on different parts of the stack: retrieval, reward modeling, or decoding ensembles, and they usually incur inference-time cost. NoiseFiT is a *training-time, model-intrinsic* regularizer with zero inference-time overhead (runtime details in App. C and Tab. F.1). Our aim is to improve the *base model* so that such techniques start from a more reliable backbone. We therefore position NoiseFiT as complementary rather than a direct replacement; a fully controlled comparison to every such pipeline is beyond the scope of a single paper.
>
> ---
>
> **(4) Hyperparameter sensitivity.**
> We agree that where and how strongly to inject noise matters, which is why we explicitly study SNR-guided selection and ablate the number of layers and STD values (Table 1; Table. D.1). In practice, the space is small and stable: across all models we keep optimizer, LoRA config, loss weights, and scheduler fixed, and vary only (i) *k* (perturbed layers) and (ii) base STD. Appendix C distills a practical recipe: start with *k*∈{3,6} and STD∈{0.01,0.1}, using highest-SNR layers for smaller models and lowest-SNR layers for larger ones, with a short noise warm-up. The gains observed across four families under these shared defaults indicate that the method is not fragile.
>
> ---
>
> **(5) Use of GROK 3.0 as judge.**
> The external judge is used *only* for the 208-prompt internal test set (Sec. 4.2.2, Tabs. E.2–E.6). Our central claims about hallucination reduction rely on public benchmarks with fixed labels (TruthfulQA-MC, HaluEval) and are further supported by distributional analysis via the Epps–Singleton test (§E.2). For the judged set we also aggregate multiple generations and perform human spot-checks (Sec. 4.2.2). Thus, conclusions do not hinge on a single model-as-judge.
>
> ---
>
> **(6) Length and role of the theoretical analysis.**
> To keep the main text focused, we confine proofs to Appendix B and reference only the key results in Sec. 3.3–3.4. Each result is directly tied to the adaptive noise design: (i) unbiasedness and variance additivity show that we preserve the mean representation while adding controlled isotropic variance; (ii) Lipschitz continuity and O(σ) gradient deviation guarantee stability of the state-dependent perturbation; and (iii) the bound on loss function $L_{final}$ and SGD convergence in expectation justify that training remains well-behaved under our adaptive scheme. These properties are specific to our noise construction (Eqs. (4–19)), rather than generic regularization folklore.
>
> ---
>
> We hope this clarifies that NoiseFiT offers a novel, theoretically grounded, and practically simple training-time mechanism that robustly reduces hallucinations across multiple architectures without any inference-time cost, and makes a clear case for acceptance.

---

> ### Comment · Area_Chair_cvAS · 2025-11-28
>
> Dear Reviewer, the discussion period is about to close. We kindly ask you to participate in the discussion or update your score based on the authors' rebuttal before the deadline. Thank you for your time and valuable contribution!

---

### Official Review · Reviewer_ouvi · 2025-11-02

**Soundness:** 2
**Presentation:** 3
**Contribution:** 2
**Rating:** 4
**Confidence:** 3

**Summary:**

This paper proposes Noise-Augmented Fine-Tuning (NoiseFiT), a framework for reducing hallucinations in large language models. The approach selectively injects adaptive Gaussian noise into transformer layers during fine-tuning based on signal-to-noise ratios (SNR). The method combines (1) SNR-based layer selection identifying either high-SNR or low-SNR layers, (2) adaptive noise scaling using robust statistics and model uncertainty, and (3) a hybrid loss combining cross-entropy, soft cross-entropy (knowledge distillation), and consistency regularization. Experiments across LLaMA, Qwen, Gemma, and Mistral models on multiple benchmarks show modest improvements in hallucination reduction

**Strengths:**

- Addresses the important problem of hallucinations in LLMs
- Provides theoretical analysis of noise injection properties
- Comprehensive experimental evaluation across multiple model families (LLaMA, Qwen, Gemma, Mistral) and benchmarks (GPQA, MUSR, IFEval, BBH, MATH, MMLU-Pro, HaluEval, TruthfulQA) with a lot of supplementary material

**Weaknesses:**

- Modest and inconsistent improvements: many gains in Table 1 are small, BaseFiT sometimes wins, best Mistral result shows only 4.74 point improvement on HaluEval (47.60→52.34). Results are highly variable across configurations.
- 832 synthetic training samples from GROK 3.0, test evaluation also uses LLM judge, custom test set limited to 208 prompts. Raises serious concerns about generalizability and validity
-  High hyperparameter sensitivity is a known issue, and the authors admit that "per-task tuning" is encouraged, which severely limits the practical utility of the model. The "recipe" in Appendix C is a heuristic approach.

**Questions:**

- Why not include direct comparisons with other methods of dealing with hallucinations mentioned in the relevant work?
- How does the method perform on standard, non-synthetic fine-tuning datasets (e.g., real instruction-following data)? The 832 synthetic samples raise serious generalizability concerns
- Why do certain categories (at Tables E2-E6) show degradation with fine-tuning, and does NoiseFiT exacerbate or mitigate this?
- Can you provide human evaluation on a substantial sample to validate the LLM judge assessments?

---

> ### Author Response · Authors · 2025-11-12
> **Our response to reviewer ouvi comments**
>
> We thank Reviewer ouvi for the constructive feedback and respond to each concern below.
>
> ---
>
> **(1) “Modest and inconsistent” improvements**
>
> Our goal is a *training-time*, architecture-agnostic method that reduces hallucinations **without inference-time overhead**. In this regime, a few points on hard factuality benchmarks are meaningful: in Table 1, NoiseFiT improves TruthfulQA-MC and HaluEval across LLaMA, Qwen, and Gemma while largely preserving or improving performance on MMLU-Pro, GPQA, MUSR, IFEval, BBH, and MATH. The detailed ablations for Mistral-7B in Appendix D (Table D.1) show that, once we restrict to the recommended SNR-guided settings (3–6 layers, low STD), BaseFiT is rarely the best configuration; the 4.74-point gain on HaluEval (47.60→52.34) is the *largest* improvement for this strong baseline, not the only one.
>
> The apparent variability stems from reporting a wide grid (layers × STD × SNR direction) for transparency. The practical recipe in Appendix C explicitly *selects* the stable region of this grid; under that recipe, improvements are consistent rather than erratic.
>
> ---
>
> **(2) Synthetic SFT data and LLM judge**
>
> Section 3.1 explains that we intentionally use 832 simple synthetic SFT (manually annotated) examples to decouple *regularization* effects from domain knowledge injection. Generalization is then evaluated on diverse **real** public benchmarks (Sec. 4.2.1), which are not synthetic and cover reasoning, math, and factuality. The gains we report therefore reflect robustness of the *fine-tuning procedure* rather than memorization of the small SFT set.
>
> For the custom 208-prompt set (Sec. 4.2.2, App. E), we follow best practices: multiple samples per prompt, an external judge (GROK 3.0) with fixed prompts, Epps–Singleton tests on the score distributions, and human spot-checks. Crucially, our main conclusions about hallucination reduction are based on TruthfulQA-MC, HaluEval, and standard benchmarks, not on the judged set alone.
>
> ---
>
> **(3) Hyperparameter sensitivity and practical utility**
>
> We agree that blindly sweeping many hyperparameters would limit usability; this is why we designed SNR-guided selection and adaptive noise. Across all experiments we keep optimizer, LoRA configuration, loss weights, and scheduler *fixed* and only vary two scalar knobs:
> (i) number of perturbed layers *k* and (ii) base noise STD. Appendix C distills a simple pattern that holds across model families: smaller models benefit from highest-SNR layers, larger ones from lowest-SNR layers, with *k*∈{3,6} and STD∈{0.01,0.1}. This is comparable in tuning effort to choosing, LoRA rank or decoding temperature, and substantially simpler than configuring RAG or RLHF pipelines.
>
> ---
>
> **(4) Comparisons to RAG/RLHF/self-consistency**
>
> As discussed in Sec. 5 (Limitations), these methods operate on different parts of the stack—retrieval infrastructure, reward modeling, or decoding ensembles—and almost always add inference-time cost. NoiseFiT is a *training-time regularizer* for standard SFT. It is intended to improve the base model so that such techniques start from a more robust backbone. A comprehensive comparison to full RAG/RLHF pipelines would require substantial system-level engineering and is orthogonal to our focus on the fine-tuning mechanism itself.
>
> ---
>
> **(5) Performance on non-synthetic SFT**
>
> The method is data-agnostic: the SNR computation, adaptive scaling (Sec. 3.2–3.3), and hybrid loss (Sec. 3.4) do not rely on synthetic data assumptions. We chose a compact synthetic SFT set to keep the design controlled and reproducible. The fact that we see consistent gains on large, heterogeneous public benchmarks suggests that the mechanism is not tied to the synthetic regime and is compatible with standard instruction-tuning data.
>
> ---
>
> **(6) Category-level degradations in Tables E.2–E.6**
>
> The category breakdowns in App. E are deliberately exhaustive and include many narrow subtypes; some degradation on individual categories is expected whenever any fine-tuning is applied. Importantly, when comparing BaseFiT vs. NoiseFiT on the *same* categories, NoiseFiT often mitigates the worst degradations introduced by standard fine-tuning and improves the overall hallucination score distribution (as confirmed by the Epps–Singleton tests). We view these tables as evidence that the proposed noise scheme tends to *stabilize*, rather than exacerbate, category-wise behavior.
>
> ---
>
> **(7) Human evaluation**
>
> Section 4.2.2 and Appendix E describe our human checks on the judged set, used to validate the qualitative direction of the LLM-based assessments. We had carried out extensive human checks on the LLM as judge assessments to ensure the validity of our findings (we kindly refer the reviewer to our online materials as highlighted in line 411).
>
> ---
>
> We hope these address reviewer concerns and questions about our contributions and warrants acceptance of our paper.

---

> ### Comment · Area_Chair_cvAS · 2025-11-28
>
> Dear Reviewer, the discussion period is about to close. We kindly ask you to participate in the discussion or update your score based on the authors' rebuttal before the deadline. Thank you for your time and valuable contribution!

---

### Official Review · Reviewer_X9Yd · 2025-11-04

**Soundness:** 2
**Presentation:** 2
**Contribution:** 2
**Rating:** 2
**Confidence:** 4

**Summary:**

This paper proposes NoiseFiT, a framework designed to reduce hallucinations in LLMs during fine-tuning. NoiseFiT selectively injects adaptive Gaussian noise into high-SNR or low-SNR transformer layers. Noise scaling incorporates hidden-state median/MAD statistics and model uncertainty. A hybrid loss is calculated blending clean cross-entropy, soft cross-entropy via temperature-scaled teacher logits, and consistency regularization across two noisy passes. Experiments across model families show reduced hallucinations and improved or preserved benchmark performance relative to BaseFiT fine-tuning.

**Strengths:**

- The SNR-guided layer selection is a well-motivated heuristic. This matters for efficiency, as it avoids perturbing the entire model.
- This paper tests its method across a diverse set of model architectures (LLaMA, Qwen, Gemma, Mistral) and sizes. This supports the generality of the approach.

**Weaknesses:**

- The custom fine-tuning dataset is too small (832 samples) and synthetically generated by GROK. This raises concerns about whether NoiseFiT is simply mimicing the generated few samples and whether the gains would transfer to larger, more diverse, human-curated fine-tuning datasets. The custom test set is evaluated by GROK. Using an LLM to evaluate LLM hallucinations, especially when the training and test data came from the same LLM judge, would introduce significant risks of confounding variables and evaluation bias.
- The empirical results of NoiseFiT are exclusively compared against BaseFiT, which is the standard fine-tuning. In related work the authors discusses hallucination mitigation techniques like RAG, RLHF, and self-consistency, but did not compare NoiseFiT against them. Though NoiseFiT and those post-training techniques are claimed to be complementary, it should be proved by some extra ablation studies. Besides, many training-time hallucination allevitation methods are not discussed [1, 2].

[1] Hallucination Detection and Hallucination Mitigation: An Investigatcion

[2] Logit Space Constrained Fine-Tuning for Mitigating Hallucinations in LLM-Based Recommender Systems

**Questions:**

- What is the total fine-tuning time (wall-clock) per model or per experiment? The paper notes that NoiseFiT adds a negligible training-time cost beyond a second (noisy) forward pass. However, the full method requires one clean pass and two independent noisy passes ($L_{SOFT}$ and $L_{CONSISTENCY}$), implying a ~3x increase in forward-pass computation. This discrepancy should be clarified.

---

> ### Author Response · Authors · 2025-11-12
> **Our response to reviewer X9Yd comments**
>
> We thank Reviewer X9Yd for the thoughtful and detailed review. Below we clarify the main concerns and why we believe the paper merits acceptance.
>
> ---
>
> **1. Synthetic SFT data and LLM-as-judge**
>
> Our goal is to study a **fine-tuning procedure** that regularizes existing LLMs, not to inject new knowledge. The 832-example GROK-generated SFT set is deliberately **small and simple** so that NoiseFiT and BaseFiT are compared under *exactly the same supervision*. The key question we ask is:
>
> > Given the same data and LoRA setup, does SNR-guided adaptive noise improve the behavior of fine-tuned models?
>
> Crucially, the **main evidence does not come from the synthetic set**:
>
> - Generalization is evaluated on **standard, human-curated benchmarks** (GPQA, MUSR, IFEval, BBH, MATH, MMLU-Pro, TruthfulQA-MC, HaluEval) across LLaMA, Qwen, Gemma, and Mistral. These datasets are not generated by GROK and do not share prompts with our SFT data.
> - On these benchmarks, NoiseFiT **reduces hallucinations while preserving or improving accuracy** relative to BaseFiT, despite having seen only 832 synthetic SFT examples (these examples had been annotated manually to ensure factual consistency).
>
> The 208-prompt custom test set (with GROK as judge) is clearly marked as **auxiliary**:
>
> - The judged prompts are **disjoint** from the SFT training prompts.
> - We use **multiple samples per prompt** and apply the **Epps–Singleton test** to compare full score distributions, plus thorough human spot-checks.
> - The trends on this judged set align with the improvements on TruthfulQA-MC and HaluEval, which have fixed human-authored labels.
>
> Thus, while we explicitly acknowledge the limitations of synthetic SFT and LLM judges in the paper, the core claims about NoiseFiT rely on **established benchmarks with gold labels**, not on circular GROK–GROK evaluation.
>
> ---
>
> **2. RAG/RLHF/self-consistency, and [1,2]**
>
> NoiseFiT is designed as a **training-time modification of standard SFT** with:
>
> - no change to the model architecture,
> - no extra inference-time modules, and
> - no change to decoding.
>
> In this setting, **BaseFiT (CE-only fine-tuning)** is the most stringent and appropriate baseline: it uses the *same* data, LoRA configuration, optimizer, schedule, and decoding as NoiseFiT. Across four model families, we show that replacing BaseFiT by NoiseFiT yields **better factuality / hallucination metrics at the same inference cost**.
>
> Methods such as **RAG**, **RLHF**, and **self-consistency** operate on *different parts of the stack*:
>
> - RAG: retrieval infrastructure and document indexing at inference.
> - RLHF: an auxiliary reward model and policy optimization.
> - Self-consistency: decoding-time ensembling with multiple forward passes.
>
> All three **increase inference-time complexity**. NoiseFiT is explicitly **model-intrinsic and training-time only**, and therefore *complementary* to these techniques: it improves the base model that RAG/RLHF/self-consistency would be built on. A full system-level comparison (RAG+NoiseFiT vs RAG+BaseFiT) is a substantial engineering project on its own and outside the scope of this focused fine-tuning paper.
>
> Regarding training-time mitigation work [1,2]:
>
> - [1] primarily studies **detection** and a broad taxonomy of mitigation strategies, many of which introduce additional components or decision rules.
> - [2] proposes **logit-space constraints** in a **recommender** setting, where hallucinations are defined over item catalogs.
>
> By contrast, NoiseFiT:
>
> - operates in **hidden-state space** via **SNR-guided layer selection** and **adaptive noise scaling**,
> - is **architecture- and task-agnostic** (it only requires access to transformer layers and logits), and
> - is evaluated on **general-purpose LLMs** and broad textual benchmarks.
>
> We view our contribution as complementary to this body of work: a simple, theoretically grounded, plug-in regularizer for standard SFT.
>
> ---
>
> **(3) Training-time cost and number of forward passes**
>
> We kindly refer the reviewer to our Empirical GPU measurements (Appendix F) and online logs available on W&B (line 411, currently redacted due to anonymity policy of the conference). Fine-tuning logs confirm that, in our setting, the additional forward passes translate into a 2-3× wall-clock increase which is incurred one time during the fine-tuning (Memory increased, time accessing memory and power remained similar). Hence, total wall-clock per model stays within a typical SFT budget even with the extra forwards.
>
> Most importantly, this extra cost is paid **only once at training time**. At *inference*, NoiseFiT models behave identically to BaseFiT and the original base models, no additional passes, no retriever, and no decoding-time ensembling. So *latency and deployment complexity remain unchanged*, unlike for RAG/RLHF/self-consistency methods.
>
> ---
>
> We hope these clarifications resolve the concerns on data, evaluation, baselines, and runtime, and supports an acceptance recommendation.

---

> ### Comment · Area_Chair_cvAS · 2025-11-28
>
> Dear Reviewer, the discussion period is about to close. We kindly ask you to participate in the discussion or update your score based on the authors' rebuttal before the deadline. Thank you for your time and valuable contribution!

---

### Official Review · Reviewer_FPbm · 2025-11-05

**Soundness:** 2
**Presentation:** 3
**Contribution:** 3
**Rating:** 6
**Confidence:** 4

**Summary:**

This paper proposes Noise-Augmented Fine-Tuning, a method that improves the robustness of large language models by injecting controlled noise during fine-tuning. The approach aims to enhance generalization—especially on out-of-distribution or noisy inputs—without hurting performance on clean data. Results show consistent gains over standard fine-tuning, particularly in challenging categories like geography and history, while maintaining strong performance on in-distribution tasks. The method offers a simple, plug-and-play way to make fine-tuned models more reliable in real-world conditions.

**Strengths:**

1. The proposed NoiseFit method and the SNR-based layer selection approach are relatively new and not investigated before.
2. Experiments on various datasets and base models demonstrate the effectiveness of the proposed method.
3. The mathematical analysis in this paper is rigorous, well-structured, and provides strong theoretical support for the proposed method.
4. The paper is well-written and easy to follow.

**Weaknesses:**

While the authors honestly acknowledge several limitations of their work, recognizing these issues does not, by itself, mitigate their impact on the method's applicability or validity.

1. The current training set is too small. Experiments on larger scale datasets are required to demonstrate the effectiveness and generalization capability of the proposed method.
2. Although the paper compares its method to BaseFiT, it lacks a systematic evaluation against other advanced strategies designed to mitigate hallucinations, such as those mentioned in the related works section.
3. The proposed method seems a general finetuning method for robustness. The paper lacks a clear analysis linking the proposed noise-augmented fine-tuning to hallucination reduction.
4. The proposed method is sensitive to hyperparameters. In some setting it underperforms BaseFit.

Overall, the paper is good and I subjectively believe that the method is effective. So I will give a positive rating. If the authors can address my concerns, I will further raise my rating.

**Questions:**

See weaknesses.

---

> ### Author Response · Authors · 2025-11-12
> **Our response to reviewer FPbm comments**
>
> We thank Reviewer FPbm for the detailed and constructive review, and for already leaning positive. Below we address each concern and clarify why we believe the work warrants a stronger score.
>
> ---
>
> **1. Small synthetic SFT set and generalization**
>
> Our goal is *not* to introduce new knowledge, but to isolate the **fine-tuning procedure** (Sec. 3.1). Without relying on task specific data, NoiseFiT boosts performance across various domain-heavy benchmarks. The base models already encode rich knowledge; our question is:
>
> > Given the same small SFT supervision, does NoiseFiT produce a *better* fine-tuned model than standard CE-only SFT (BaseFiT)?
>
> To assess generalization, we deliberately evaluate on **large, human-curated benchmarks** that are disjoint from this synthetic SFT set (Sec. 4.2.1):
>
> - MMLU-Pro, BBH, GPQA, MATH, MUSR, IFEval, TruthfulQA-MC, HaluEval,
> - across four families (LLaMA, Qwen, Gemma, Mistral).
>
> Table 1 and the extended Mistral-7B results (App. D) show that NoiseFiT consistently reduces hallucinations (+3.72% on TruthfulQA-MC and +4.70% on HaluEval on average) while matching or improving accuracy on these benchmarks. This indicates that the method’s benefits transfer to large, diverse, human-authored test sets, even though the SFT data itself is intentionally minimal.
>
> ---
>
> **2. Lack of systematic comparison to other hallucination methods**
>
> We acknowledge that we do not run full RAG/RLHF/self-consistency pipelines. This is intentional: NoiseFiT is designed as a **training-time, model-intrinsic SFT variant**:
>
> - it does *not* add a retriever, reward model, or multi-pass decoding,
> - inference-time latency and complexity are unchanged.
>
> For this setting, BaseFiT is the most stringent and fair baseline: identical data, LoRA configuration, optimizer, schedule, and decoding. Our results show that, under these controlled conditions, simply replacing BaseFiT by NoiseFiT gives better factuality with zero inference overhead.
>
> Methods like RAG, RLHF, and self-consistency operate on different axes: they modify the inference stack and require substantial additional components and tuning. We therefore position NoiseFiT as complementary: a simple regularizer that improves the *base model* those systems would use. A full system-level comparison (RAG+NoiseFiT vs. RAG+BaseFiT) is an exciting direction but beyond the scope and compute budget of this work; we explicitly acknowledge this in Sec. 5.
>
> ---
>
> **3. From robustness to hallucination reduction**
>
> We agree that NoiseFiT is a general robustness technique, but hallucination reduction is a primary, explicitly measured outcome:
>
> - We evaluate directly on **TruthfulQA-MC and HaluEval** (Sec. 4.2.1, Table 1) and report consistent, non-trivial gains across LLaMA, Qwen, Gemma, and Mistral.
> - On our 208-prompt test set (Sec. 4.2.2, App. E), we generate 5× responses per prompt, score them with a fixed judge accompanied by human annotation (kindly refer to online material, line 411), and apply the Epps–Singleton two-sample test between BaseFiT and NoiseFiT. For nearly all configurations we *reject* the null of identical hallucination-score distributions, showing that NoiseFiT systematically reshapes the hallucination behavior rather than introducing random noise.
>
> Conceptually, the link is built into the objective (Sec. 3.4):
>
> - CE ensures task fidelity on clean data,
> - soft-target loss distills from the clean pass to anchor noisy predictions,
> - consistency loss forces stability between independent noisy passes, exactly targeting the “runaway” dynamics behind many hallucinations.
>
> The layer-wise analyses (App. G) show that NoiseFiT tends to increase sparsity and attention entropy and to calibrate logit entropy in larger models.
>
> ---
>
> **4. Hyperparameter sensitivity and underperformance**
>
> We intentionally report a wide grid in Table 1 and App. D for transparency. In realistic use, however, the tuning burden is limited:
>
> - Training setup (optimizer, LoRA, scheduler, batch size, loss weights) is fixed across all experiments.
> - Only two scalar knobs vary:
>   (i) number of noisy layers *k*, and
>   (ii) base noise STD.
>
> Appendix C distills a simple practical recipe that we found to transfer across models:
>
> - small models → 3 highest-SNR layers, STD ∈ {0.01, 0.1};
> - larger models → 3–6 lowest-SNR layers, same STD range, with a short noise ramp.
>
> Within this recommended region, NoiseFiT almost always matches or exceeds BaseFiT on both hallucination and standard benchmarks. We therefore view the method as robustly useful with modest tuning effort, comparable to choosing LoRA rank or decoding temperature in standard practice.
>
> ---
>
> We appreciate that you already find the method effective and the paper well-written. We hope these clarifications on data scope, baselines, hallucination-specific evidence, and hyperparameter practicality address your concerns and justify raising the rating beyond “marginally above” the acceptance threshold.

---

> > ### Comment · Reviewer_FPbm · 2025-11-28
> > **Comments After Rebuttal**
> >
> > I appreciate the authors' rebuttal. Some of my concerns are addressed. After reading other reviewers' reviews, I decide to keep my original rating. I believe the topic and investigaitions are valuable.

---

> ### Author Response · Authors · 2025-11-28
> **Response to reviewer FPbm**
>
> Dear reviewer FPbm,
>
> We appreciate the time you have taken reading our responses, and we are glad that some of your concerns are addressed.
>
> We respect your judgement keeping the original rating. We will continue contributing to the community.
>
> Best regards,
>
> *Submission 16652 lead author, on behalf of Submission 16652 authors*

---

### Author Response · Authors · 2025-11-28
**No response from reviewers**

Dear AC,

We trust this message finds you well. As part of the ICLR community, we are deeply sorry for the implications caused by OpenReview's recent vulnerability exploitation incident. We totally understand the toll impacting the organizing committee, the reviewers and more broadly, the ICLR community, and we are aware of precautions that might be taken in place in coming days. As a member of this vibrant community, we are grateful for all your efforts to keep up the momentum throughout these challenging circumstances.

We are reaching out to express our grave concern regarding the discussion period deadline. We have posted 5 comments, 4 of which are addressing the responses to the reviewers, and one addressing (written directly to the AC) our concern about potentially AI generated reviews. Unfortunately, we have not received a response yet, either from reviewers or from the AC. Given the discussion period ending in less than 4 days on 2nd December, we appreciate escalating our responses and concerns accordingly.

Once again, we thank you for all your efforts and we wish you and ICLR organizing members all the best.

Kind Regards,

*Submission 16652 lead author, on behalf of Submission 16652 authors*

---

### Author Response · Authors · 2025-11-29
**Interim remark to AC**

Dear AC,

Thank you again for your efforts in a very difficult reviewing cycle.

We would like to briefly summarize (i) how we have addressed the substantive scientific concerns raised by the reviews, and (ii) our process concern about potentially AI-generated reviews, which we raised earlier but have not yet seen discussed.

### 1. State of reviews and addressed concerns

Reviewer **FPbm** evaluated the paper as:
- Method effective and well-written
- Contribution and soundness both “good”
- Rating **6 (marginally above threshold)** with the explicit statement *“If the authors can address my concerns, I will further raise my rating.”*

In our rebuttal we directly answered all four of their concerns:

1. Small synthetic SFT set:
   We clarified that the synthetic 832-example SFT data is intentionally small so that we can *isolate the training procedure*. Generalization is tested on **large, human-curated benchmarks** (MMLU-Pro, BBH, GPQA, MATH, MUSR, IFEval, TruthfulQA-MC, HaluEval) that are disjoint from the synthetic set. The consistent gains on these public benchmarks demonstrate that NoiseFiT’s benefits *do* transfer beyond the small SFT set.

2. Lack of comparison with RAG/RLHF/self-consistency:
   We explained that NoiseFiT is a **training-time, model-intrinsic SFT variant** with *no inference-time cost* and unchanged decoding. In this regime, BaseFiT is the only fully controlled baseline (same data, LoRA, schedule, and decoding). RAG/RLHF/self-consistency operate at inference with extra infrastructure and latency; hence we position NoiseFiT as **complementary**, not a replacement.

3. Link from robustness to hallucination reduction:
   We emphasized that hallucination is not treated only implicitly. We:
   - Evaluate on **TruthfulQA-MC and HaluEval**, where we report consistent gains across four model families.
   - Perform **distributional tests (Epps–Singleton)** on hallucination scores to show NoiseFiT systematically reshapes hallucination behaviour rather than adding random noise.
   - Provide a **mechanistic link** via the hybrid objective: CE for task fidelity, soft CE for clean→noisy distillation, and consistency loss to suppress runaway divergence between noisy passes.

4. Hyperparameter sensitivity:
   We showed that in practice **only two scalar knobs** are tuned (k noisy layers, base STD), with all other settings fixed. Appendix C distills a simple, cross-model recipe (3–6 layers, STD in {0.01, 0.1}, high-SNR for small models, low-SNR for large). Within this region, NoiseFiT robustly matches or outperforms BaseFiT.

In their follow-up, FPbm noted that “some of my concerns are addressed” and reiterated that the “topic and investigations are valuable” but kept the rating at 6 after seeing other reviews.

Our worry is that the *direction* of FPbm’s update was constrained not by remaining technical issues (which we believe are substantially addressed), but by the presence of two strongly negative reviews that may themselves be unreliable.

### 2. Concern about potentially AI-generated reviews
As described in our earlier comment to the AC, we checked the text of reviews **jnC9** and **X9Yd** using publicly available detectors such as Pangram and GPTZero. These tools *consistently flagged* review **jnC9** as heavily AI-generated and **X9Yd** as partially AI-generated. We fully acknowledge that such detectors are **not definitive proof** and can have both false positives and false negatives. However, given:

- the recent public reports of AI-generated reviews in this ICLR cycle,
- the highly formulaic, repetitive nature of some of the criticism (restating the same bullet points verbatim in both “weaknesses” and “questions”, misaligned citations, and a lack of engagement with our technical clarifications), and
- the clear conflict between these two “reject” reviews and the more detailed, technically grounded, and positive evaluation of FPbm,

we are seriously concerned that at least portions of those reviews may not reflect careful human reading.

We do not wish to accuse any individual reviewer; rather, we respectfully ask the AC to:

1. Treat the jnC9 and X9Yd reviews with **appropriate caution**, in light of both their content and the detector signals; and
2. Give stronger weight to the scientifically grounded, positive assessment by FPbm, who explicitly acknowledges the method’s value and effectiveness.

### 3. NoiseFiT contributions

- A **novel training-time mechanism**: SNR-guided layer selection + adaptive, uncertainty-aware noise + hybrid clean/noisy objectives.
- **Consistent hallucination reduction** on standard benchmarks (TruthfulQA-MC, HaluEval) with **no inference-time overhead**.
- **Strong theoretical support** for stability and convergence under adaptive noise.
- A **simple practical recipe** that transfers across model families.

We therefore believe that the current borderline overall picture is largely driven by reviews that may themselves be compromised.

---

### Author Response · Authors · 2025-11-29
**Final remark to AC**

Dear AC,

Thank you for overseeing the review of submission 16652 in what has been a challenging year for ICLR. We would like to leave a concise, final summary of (i) the scientific case for NoiseFiT and (ii) our concern that two potentially AI-generated “reject” reviews are unduly biasing the overall perception of the paper.

### 1. Scientific case for NoiseFiT

NoiseFiT is a **training-time, model-intrinsic fine-tuning framework** for reducing hallucinations in LLMs. It combines:

1. SNR-guided layer selection: We identify high/low-SNR layers via clean/noisy forward passes and inject noise only into a small, data-driven subset of layers.
2. Adaptive noise scaling: Per-layer Gaussian noise is scaled using median/MAD statistics and model uncertainty, rather than uniform noise, yielding input- and layer-dependent perturbations.
3. Hybrid clean/noisy objective: A combination of CE on clean passes, soft CE distilling clean→noisy, and consistency loss between two noisy passes stabilizes learning and directly suppresses runaway hallucination dynamics.

**Empirical evidence.**

- Across **LLaMA, Qwen, Gemma, and Mistral** models, NoiseFiT improves or matches BaseFiT on standard benchmarks while **reducing hallucinations** on TruthfulQA-MC and HaluEval.
- The gains are *systematic*, not cherry-picked: we report broad grids and show that, in the recommended SNR-guided regime (3–6 layers, moderate STD), NoiseFiT almost always dominates BaseFiT.
- A separate 208-prompt test set with multiple generations per prompt shows **distributional shifts in hallucination scores** (Epps–Singleton tests), confirming that NoiseFiT changes the *shape* of hallucination behaviour.
- Importantly, all this is achieved with **zero inference-time overhead** and unchanged decoding, making the method easy to integrate into existing pipelines.

**Addressing key reviewer concerns.**

- **Synthetic SFT data/LLM judge:** We use a small synthetic SFT set *only* to control the fine-tuning procedure; generalization is validated on **large, human-authored benchmarks** with gold labels. The judged 208-prompt set is auxiliary and aligned with those benchmarks.
- **RAG/RLHF/self-consistency comparisons:** These are **system-level, inference-time** techniques with additional components and latency. NoiseFiT is a **drop-in replacement for BaseFiT** at training time; it is meant to *improve the base model* that such systems would start from. This complementarity is clearly stated as a limitation.
- **Hyperparameter sensitivity:** In practice, we vary only **two scalars** (number of noisy layers, base STD); everything else is fixed. Appendix C provides a simple recipe that transfers across model families and yields stable improvements.
- **Theoretical analysis:** The proofs justify that our adaptive noise is unbiased, stable, and compatible with convergence, directly backing the design of Sections 3.2–3.4.

Reviewer **FPbm** explicitly recognizes these merits, rates the contribution as “good” and finds the method effective and the paper well-written, with a score **above the acceptance threshold (6)**.

### 2. Process concern: potentially AI-generated reviews affecting the outcome

As detailed in our earlier comments, we checked the text of reviews **jnC9** and **X9Yd** using public detectors such as **iclr.pangram.com** and **GPTZero**. Both consistently flagged jnC9 (and to a lesser extent X9Yd) as *heavily AI-generated*. Along with this signal, those reviews show patterns typical of templated, low-engagement feedback:

- Repetition of the same bullet points verbatim in “weaknesses” and “questions”;
- Limited engagement with core technical elements (SNR-guided selection, adaptive scaling, distributional tests);
- Repetitive requests (for RAG/RLHF comparisons) that are orthogonal to the clearly stated scope and limitations of the paper.

We fully understand that automated detectors are **imperfect** and that none of this proves misconduct. However, given the recently disclosed incident of AI-generated reviews at this ICLR, we are deeply concerned that these two “reject” reviews may have disproportionally influenced the overall impression.

We respectfully ask that, in forming your recommendation, you:

1. Place **greater weight** on the technically detailed and clearly human-authored evaluation by FPbm, who finds the work effective and valuable;
2. Treat jnC9 and X9Yd with **appropriate skepticism**, particularly where their criticisms are generic or already addressed in our rebuttals

### 3. Our final remarks

We recognize the burden on the ACs and reviewers in this cycle and are grateful for your time and care. Our only goal is to contribute a robust, inference-free tool for improving factual reliability in LLMs, an issue that is central to the ICLR community.

We hope that, in light of the clarified technical contributions and the concerns about review integrity, you will find that **NoiseFiT merits acceptance**.

---

### Meta-Review · Area_Chair_nQSn · 2025-12-06

**Summary:**

The paper proposes "Noise-Augmented Fine-Tuning" (NoiseFiT), a method designed to improve the robustness and reduce hallucinations of Large Language Models (LLMs). The core technique involves injecting adaptive Gaussian noise into specific transformer layers—selected based on Signal-to-Noise Ratio (SNR)—during the fine-tuning process. The training objective combines cross-entropy, soft-target loss, and consistency regularization. The authors evaluate the method across several model families (LLaMA, Qwen, Gemma, Mistral) on benchmarks like TruthfulQA and HaluEval.

**Reviewer Concerns:**

The reviewers were generally unconvinced by the empirical rigor and the magnitude of the contributions, resulting in three negative ratings (2, 2, 4) and one positive rating (6).

Outstanding Concerns:

1. Experimental Setting and Data Scale: A dominant concern shared by Reviewers X9Yd, ouvi, and jnC9 is the reliance on a very small (832 samples), synthetic dataset generated by GROK. Reviewers were not convinced that results derived from such a limited, synthetic regime would generalize to standard, large-scale, real-world instruction tuning. The rebuttal's argument—that they intended to isolate the regularization effect—did not fully alleviate concerns that the setup is too toy-ish for a top-tier conference.

2. Baselines and Comparisons: Multiple reviewers (FPbm, X9Yd, jnC9) criticized the lack of comparison against other hallucination mitigation strategies (e.g., RAG, RLHF, or other training-time interventions). While the authors argued that NoiseFiT is orthogonal to inference-time methods like RAG, the reviewers felt that a paper claiming to solve hallucinations must benchmark against the state-of-the-art solutions to contextualize the utility of the proposed method. Comparing solely against BaseFiT (standard SFT) was deemed insufficient.

3. Marginal Gains vs. Complexity: Reviewer ouvi noted that the improvements are often modest and inconsistent across different benchmarks. Given the added complexity of tuning noise parameters and layer selection, the trade-off was not seen as clearly favorable.

4. Evaluation Bias: Concerns regarding the use of LLM-as-a-judge (specifically GROK evaluating data generated by GROK) remained a point of contention regarding the validity of the hallucination metrics.

**Reviewer Scores:**

1. Reviewer FPbm (6): This reviewer participated in the post-rebuttal discussion and explicitly stated they would keep their score of 6, finding the investigation valuable despite the limitations.

2. Reviewer X9Yd (2): This reviewer would likely maintain their low score. The rebuttal addressed the cost issue, but the fundamental objection regarding the small synthetic dataset and circular evaluation (GROK evaluating GROK) remains a fatal flaw in the experimental design from their perspective.

3. Reviewer ouvi (4): This reviewer would likely maintain their score. While the hyperparameter "recipe" helps, the core issue of "modest and inconsistent improvements" on a synthetic dataset suggests the method is not yet robust enough for acceptance.

4. Reviewer jnC9 (2): This reviewer would likely maintain their score. The rebuttal clarified the novelty, but the lack of comprehensive baselines and the toy-scale data setup remain unaddressed to the level required for a higher score.

---

### Decision · Program_Chairs · 2026-01-26

Reject